# Learning lessons from over-crediting to ensure additionality in forest carbon credits

Tom Swinfield [1,2] ✉, Abby Williams [1,2,3], David Coomes [2,4], Michael Dales[2,5], Patrick Ferris[2,5], Alejandro Guizar-Coutiño [2,3,4], James Hartup [1,2,6], Jody Holland[1,2], Sadiq Jaffer [2,5], Julia P. G. Jones [7,8], Miranda O. K. Lam [1,2], Srinivasan Keshav [2,5], Anil Madhavapeddy [2,5], Eleanor Toye-Scott[2,5], Thales A. P. West [9,10] & Andrew Balmford [1,2]

Independent evaluations have shown substantial over-issuance of REDD+ (Reducing Emissions from Deforestation and Degradation) credits traded on the voluntary carbon market. We synthesise these evaluations to estimate the additional forest conservation achieved by first-generation REDD+ projects and to identify mechanisms underlying over-crediting. We combine six independent ex post evaluations of avoided deforestation covering 44 REDD+ projects. These evaluations show that most projects reduced deforestation, but that they claimed an aggregate of 10.7 times more avoided deforestation than is justified by independent estimates. This discrepancy is not driven by the choice of forest cover data, but by selection bias in projects' control areas and modelling approaches. Although recent initiatives that transfer assessment to unconflicted parties and restrict methodological flexibility are critical, they are insufficient. Ex post certification against credible counterfactuals is necessary if carbon markets are to represent causal reductions in deforestation.

Halting tropical deforestation is essential to limit global temperature rise to below 2 °C[1] and prevent mass extinction[2,3]. Yet tropical forests continue to be lost[4,5] and forest conservation is severely underfunded, with an estimated annual finance gap of US $216 billion[6]. There is substantial global interest in the potential for both compliance and voluntary markets to fund forest conservation despite widespread evidence that many first-generation REDD+ (Reducing Emissions from Deforestation and forest Degradation) projects have issued more credits than were justified by their impact on deforestation[7–11]. The resulting controversy[12] contributed to more than US $1.1Bn being wiped from the value of the voluntary carbon market in 2023 and a further contraction in 2024[13]. Understanding what these early projects did and did not achieve and the mechanisms that resulted in over-

crediting is key to informing efforts to improve the integrity of forest carbon credits, which is an essential step if they are to contribute to closing the forest finance gap.

A major challenge in measuring the impacts of forest conservation projects is estimating counterfactual outcomes (what would have happened to deforestation in the absence of a project)[14–18]. The first-generation of REDD+ projects could choose from several certification methodologies (Table 1), each of which used measurements of historic deforestation in expert-selected 'reference' areas in conjunction with *ex ante* modelling approaches to predict 'baseline' deforestation (see supplementary note 1 for further details)[11]. Credits were issued by comparing the *ex ante* baseline with observed deforestation in project areas. In parallel, a rich scientific literature has developed methods for

[1]Department of Zoology, University of Cambridge, Cambridge, UK. [2]Conservation Research Institute, University of Cambridge, Cambridge, UK. [3]Department of Biology, University of Oxford, Oxford, UK. [4]Department of Plant Sciences, University of Cambridge, Cambridge, UK. [5]Department of Computer Science and Technology, University of Cambridge, Cambridge, UK. [6]School of Environmental Sciences, University of Liverpool, Liverpool, UK. [7]School of Environmental and Natural Sciences, Bangor University, Bangor, UK. [8]Department of Biology, Utrecht University, Utrecht, the Netherlands. [9]Institute for Environmental Studies (IVM), Vrije Universiteit Amsterdam, Amsterdam, the Netherlands. [10]Centre for Environment, Energy and Natural Resource Governance, University of Cambridge, Cambridge, UK. ✉e-mail: tws36@cam.ac.uk

**Table 1 | Description of the certification methodologies used to evaluate first-generation REDD+ projects**

| Method | Method for estimating the counterfactual rate of deforestation | Unit size | Forest cover layer[a] | Number of projects[b] |
|---|---|---|---|---|
| VM0006 | Reference area selected with exposure to drivers of deforestation within ±10% of the range observed in project area. Counterfactual deforestation rates are predicted *ex ante* from beta regression or historical average of deforestation observed in the reference area during reference period. | Whole project | Bespoke | 11 |
| VM0007 | Reference area selected with exposure to drivers of deforestation within ±20% of the range observed in project area to measure historic deforestation. Counterfactual deforestation rates are predicted *ex ante* from linear or non-linear regression, or historical average of deforestation or population growth observed in the control rea during reference period. These are allocated to the project using a spatial risk model produced for the area surrounding the project which should have exposure to drivers of deforestation within ±30% of the range observed in project. | Whole project | Bespoke | 15 |
| VM0009 | Reference areas are selected with no defined requirement for their similarity to the project in exposure to drivers of deforestation. Counterfactual deforestation rates are predicted *ex ante* from logistic regression of deforestation observed in the reference area during the reference period. | Whole project | Bespoke | 3 |
| VM0015 | Reference areas selected with exposure to drivers of deforestation within ±10% of the range observed in project area. Counterfactual deforestation rates are predicted *ex ante* from linear or non-linear regression or historical average or simulation approaches of deforestation observed in the reference area during the reference period. | Whole project | Bespoke | 15 |

[a]Forest cover layer was locally trained 'bespoke' classifications used in certified estimates by projects.
[b]The projects included in our analysis were those for which we were able to obtain certified estimates of avoided deforestation (supplementary table 1) and were assessed by at least one quasi-experimental approach.

estimating counterfactual outcomes from observational data[19–22] and for applying them to conservation[14,15,17,18]. These quasi-experimental methods account for confounders in selecting control areas and measure outcomes *ex post* in both project and control areas (Table 2).

The prominence of REDD+ has drawn scrutiny from several independent research groups that have applied these quasi-experimental methods to evaluate the impact of first-generation REDD+ projects on deforestation[7–9,23–25]. Consistently, these studies have found evidence of over-crediting. Proposed explanations include the use of inappropriate reference areas, unrealistic *ex ante* modelling that exaggerated expected deforestation, and selective use of certification methodologies that inflated credit issuance[7,9,11,23]. However, these hypothesised mechanisms through which over-crediting occurred have not been systematically tested[10]. Some industry insiders have rejected the evidence for over-crediting altogether[26,27], arguing that (1) quasi-experimental estimates of avoided deforestation are inconsistent across studies due to methodological differences[28] and (2) over-crediting is an artefact of using of publicly available global forest cover datasets that detect less deforestation than the bespoke remote sensing classifications used by projects[29].

Carbon markets are, of course, not the only way tropical forest conservation can be funded[30,31]. There are well-known problems associated with carbon offsetting[32] and with the integrity of forest carbon credits beyond the issues of over-crediting and additionality[33,34]. Nevertheless, carbon markets are likely to play a role in bridging the forest funding gap, at least in the short to medium term[6]. It is therefore critical to learn as much as possible from the first-generation of REDD+ projects, both their successes and failures, to inform the future development of methods for certifying credits. Given the adverse implications of over-crediting for the core objective of carbon markets (reducing carbon in the atmosphere), understanding the mechanisms through which over-crediting occurred is vital if future approaches are to avoid repeating the failures of the past.

In this study, we synthesise results from six quasi-experimental evaluations to provide a comprehensive assessment of the impact of 44 first-generation REDD+ projects (representing 45% of the projects producing credits by 2020). We compare these results with certified assessments from the projects themselves. We address critiques of independent quasi-experimental evaluations (questions 1 and 2 in Fig. 1) and investigate two hypothesised mechanisms by which over-crediting could have occurred (questions 3 and 4). Our results show that most projects slowed deforestation consistently across independent evaluations, yet over-crediting remained substantial. The discrepancy between quasi-experimental and certified estimates cannot be explained by differences in the remote sensing layers used, rather, it stems from bias in reference-area selection and unrealistic modelling of future deforestation. These findings point to changes needed in crediting methodologies to ensure forest carbon credits deliver their intended climate benefits.

## Results

### Q1 How consistent are estimates of project impacts assessed using independent quasi-experimental methods?

The majority of projects (36 of 44) reduced deforestation according to assessments by quasi-experimental methods (Fig. 2). Among projects with multiple quasi-experimental estimates, 12 (29% of the 42 with calculable confidence intervals) showed statistically significant reductions in deforestation at the 95% confidence level. Eight projects (shown in red in Fig. 2) experienced more deforestation than their quasi-experimental controls, although only in one case was this significant.

Despite most projects reducing deforestation, over-crediting was widespread with certified estimates of avoided deforestation significantly exceeding mean quasi-experimental estimates (one-tailed paired Wilcoxon signed-rank test; $V = 973$, $p < 0.001$). Over-crediting

**Table 2 | Description of the studies using quasi-experimental methods to evaluate first-generation REDD+ projects**

| Method/study | Method for estimating the counterfactual rate of deforestation | Unit size | Forest cover layer[a] | Number of projects[b] |
|---|---|---|---|---|
| West et al.,[7] | Synthetic control produced by weighting untreated polygons of land titles by their similarity to projects in terms of deforestation trajectories and observable confounders. Deforestation in project and control areas is observed ex post. | Whole project | MapBiomas (reprocessed) | 11 |
| West et al.,[7] | Synthetic control developed from randomly placed circular plots weighted by similarity to projects in terms of deforestation trajectory and observable confounders. Deforestation in project and control areas is observed ex post. | Whole project | GFC | 18 |
| Guizar-Coutiño et al.,[8] | 1:1 matching with replacement selected from untreated control units based on similarity in observable confounders. Deforestation in project and control areas is observed ex post. | 0.09 ha pixel | ACC | 35 |
| Guizar-Coutiño et al.,[25] | 1:1 Propensity score matching with replacement via random forests, followed by propensity score sub-classification on the matched dataset to ensure covariate balance. The average treatment effect was estimated using multiple model specifications to assess robustness and sensitivity to unobserved confounding evaluated using the sensemakr framework[49]. Deforestation in project and control areas is observed ex post. | 7.10 ha plot | ACC | 32 |
| Tang et al.,[23] | Penalized synthetic control developed from randomly placed circular plots weighted by similarity to projects in terms of deforestation trajectory and observable confounders. Deforestation in project and control areas is observed ex post. | Whole project | GFCl | 37 |
| PACT | 1:1 matching without replacement selected from untreated control units based on similarity in observable confounders. Deforestation in project and control areas is observed ex post. Bootstrapped 100 times using random samples from candidate controls. | 0.09 ha pixel | ACC | 35 |

[a]Forest cover layers were regional or global peer-reviewed data product, specifically: MapBiomas, Global Forest Change (GFC) or the Tropical Moist Forest Annual Change Collection (ACC)[5].

[b]The projects included in our analysis were those for which we were able to obtain certified estimates of avoided deforestation (supplementary table 1) and were assessed by at least one quasi-experimental approach.

was statistically significant at the 95% confidence level for 32 projects (76% of the 42 with calculable confidence intervals; Fig. 2a). Consequently, the mean over-crediting ratio (the certified estimate divided by the mean quasi-experimental estimate) was 4.1 (95% bootstrapped CI: 2.7 to 7.0), indicating the typical project achieved about a quarter of the avoided deforestation reported by certified estimates. Across all the credits assessed, the global over-crediting ratio was 10.7 (95% bootstrapped CI: 5.4 to 26.5), implying that, on average, certified credits represented approximately one-eleventh of the avoided deforestation claimed (Fig. 2b). Over-crediting was not universal: two projects from Madagascar and one from Peru had a mean over-crediting ratio less than 1 meaning these credits achieved more avoided deforestation than claimed (Supplementary Figs. 1–2).

**Q2 Is the use of global deforestation layers the reason for the discrepancy between quasi-experimental and certified estimates?**

A high-profile critique has suggested that global layers used in quasi-experimental analyses detect less deforestation than bespoke layers used by projects and that this, rather than over-crediting, explains the observed discrepancies[29]. We tested this critique by comparing estimates of deforestation (loss of undisturbed forest) in project areas derived from global deforestation layers with those reported by projects using their bespoke forest cover classifications. We were unable to perform the comparison for reference areas because deforestation rates there were not reported in project certification documents. For project areas, we found that, rather than being lower, deforestation rates measured independently using the European Union's tropical moist forest annual change collection (ACC[5]) were actually greater than certified measurements based on bespoke forest cover layers. Among the 36 projects that reported certified measurements, the median rate measured using the ACC was 0.26%/year, compared with a median certified deforestation rate of 0.17%/year (Fig. 3a). The median pairwise discrepancy between ACC and certified rates was 0.13%/year (Fig. 3b), which was significantly different from zero (Wilcoxon paired signed-ranks test, $V = 520$, d.f. = 35, $p = 0.001$). When we ran the analysis to include pixels classified as degraded forest in the ACC-measured forest area at project start and measured deforestation only as long-term (>2.5 years) transitions to non-forest, deforestation rates were lower and not significantly different from certified values (supplementary figs. 3–6). Together these results indicate that deforestation products (such as the ACC) used in quasi-experimental analyses have not detected less deforestation than the bespoke layers used in certification. Underestimation of deforestation in control areas therefore cannot explain the observed over-crediting.

**Q3 Were reference areas similar to projects in their exposure to deforestation?**

Having established that reports of lower additionality were consistent across quasi-experimental assessments and were not driven by forest-change layers that underestimate deforestation, we next examine why certification methodologies may have overestimated project additionality. Specifically, we ask whether the reference areas used were representative of the deforestation pressures to which projects were exposed.

Reference areas were systematically different from project areas, exhibiting lower exposure to deforestation before project implementation. To illustrate this, we focus on project 958 in Peru (equivalent figures are shown for all projects in Supplementary Figs. 7–19). Project 958's reference area was located immediately outside the project (Fig. 4a). Despite this proximity, the reference area was less inaccessible (a mean of 13.8 h compared with 25.5 h) and less forested at the start of the relevant evaluation period (78% for the reference area in 2001 compared with 95% for the project in 2010

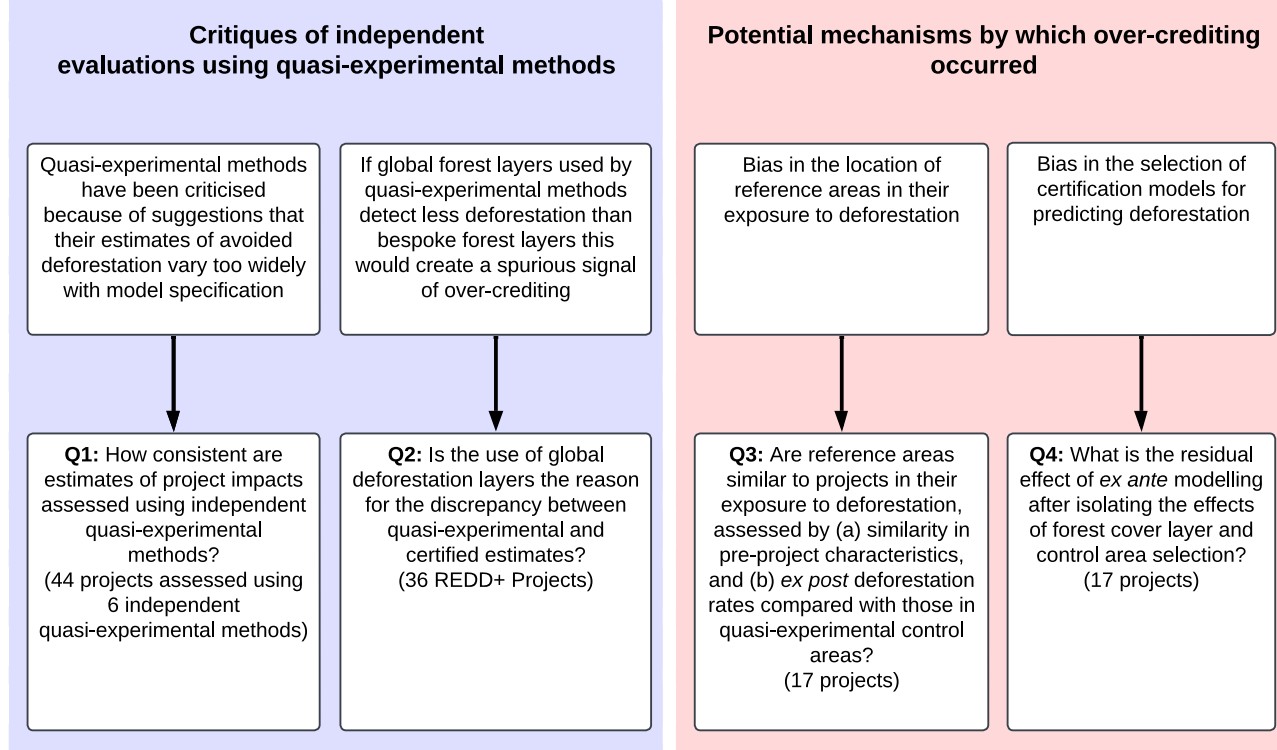

**Fig. 1 | Research questions exploring the mechanisms behind over-crediting.** The analyses test critiques of independent quasi-experimental evaluations of REDD + projects and investigates mechanisms that may have led to over-crediting. Specific research questions (Q1-4) and the number of REDD+ projects with data available to answer each question are shown. The list of projects included in each analysis is given in Supplementary Table 1.

(Fig. 4b). Standardised mean differences of −1.02 and −0.58 respectively indicate that the reference area were significantly different from the project. By contrast, the control units selected by the quasi-experimental matching approach were distributed much more widely across Peru (Fig. 4a) and were nearly identical to the project in terms of observable confounders (Fig. 4b).

The same pattern was observed across all 17 projects that provided mapped reference areas (Fig. 4c and Supplementary Figs. 7–19). Compared with project areas, reference areas were more accessible (median standardised mean difference = −0.30, $t = −2.41$, d.f. = 16, $p = 0.028$). 82% of projects used reference which had locations more accessible than the most accessible locations within projects. Similarly, reference areas were less forested than projects at their start (median standardised mean difference = −0.42, $t = −3.10$, d.f. = 16, $p = 0.007$) as well as five and ten years beforehand (median standardised mean difference = −0.48, $t = −4.04$, d.f. = 16, $p = 0.003$; and −0.48, $t = −4.28$, d.f. = 5, $p = 0.008$, respectively). Historic deforestation rates were not significantly different. In contrast, none of the quasi-experimental control areas showed meaningful differences from the project areas in any of the observed confounders (Fig. 4c).

These differences in exposure to observable confounders between reference and quasi-experimental control areas were associated with a directional difference in their deforestation rates during the project period when assessed using the ACC layer. Across the 17 projects, median deforestation rates in reference areas were 0.69%/year compared with 0.41%/year in the quasi-experimental control areas (Fig. 4d). Although this was 1.69 times more deforestation the increase was not statistically significant (one-tailed Wilcoxon paired signed-ranks test, $V = 103$, $p = 0.1123$). The same pattern was found when considering the loss of either undisturbed or degraded forest in the ACC deforestation measurements (Supplementary Figs. 20–22).

**Q4 After isolating the effects of forest cover layer and reference area selection, how much over-crediting can be explained by ex ante modelling?**

To dissect the relative contribution of the mechanisms identified in preceding analyses, we examined their effects in the subset of projects ($n = 17$) for which all required data were available. The smaller sample size meant that most effects within this subset were not statistically significant, although they were of a similar magnitude to those observed in the larger analyses of each mechanism. Certified assessments reported substantially more avoided deforestation than quasi-experimental assessments (0.95 %/year shown in pink compared to 0.18%/year shown in green in Fig. 5; Wilcoxon paired signed-ranks test, $V = 134$, $p = 0.002$). We then sequentially re-estimated avoided deforestation using different combinations of the deforestation rates drawn from the certified and quasi-experimental assessments to isolate the effect of each mechanism, with any remaining over-crediting assumed to be explained by *ex ante* modelling.

First, to account for differences between the remote sensing layers in project areas, we substituted the ACC-measured deforestation rates with the certified rates. Because certified rates were lower, this substitution increased median avoided deforestation slightly to 0.20 %/year (orange in Fig. 5), indicating that the choice of deforestation rate for the project area contributed only about 2% of the observed over-crediting.

Second, to isolate the effect of reference area selection, we substituted the quasi-experimental control areas with the certified reference areas and measured deforestation in both using the ACC. Median avoided deforestation then increased to 0.64 %/year (blue in Fig. 5), intermediate between quasi-experimental and certified assessments, suggesting that reference area choice probably accounted for 57% of the over-crediting.

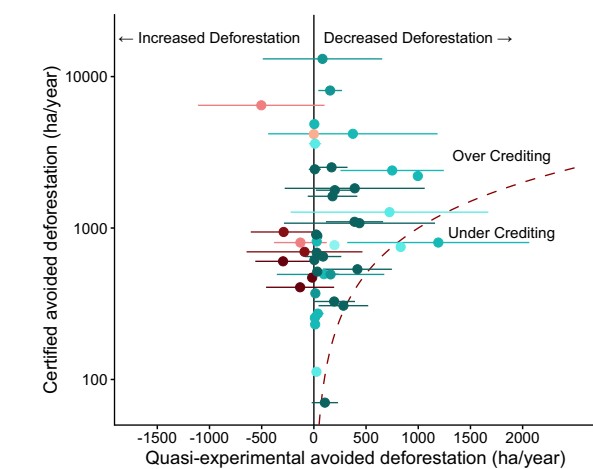

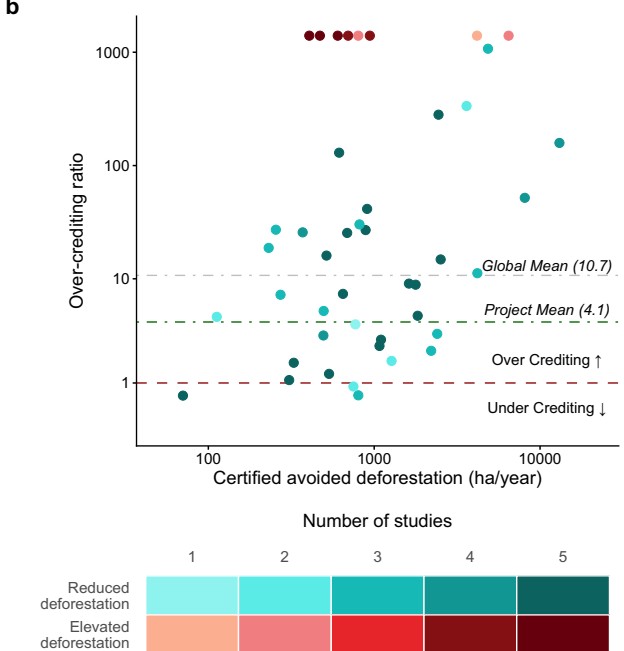

**Fig. 2 | Comparison of quasi-experimental and certified estimates of project performance. a** Quasi-experimental evaluations show consistent evidence that REDD+ projects slowed deforestation, but certified estimates were higher, indicating widespread over-crediting. Points represent mean estimates from the six quasi-experimental studies, with error bars showing 95% confidence intervals. **b** The over-crediting ratio (the certified estimate divided by the mean quasi-experimental estimate) plotted against certified avoided deforestation estimates. The dashed grey line indicates the global mean over-crediting ratio and the dashed red lines mark parity between certified and quasi-experimental estimates. Points above and below the red line represent over- and under-crediting respectively. Darker colours denote projects evaluated by a greater number of quasi-experimental approaches (up to five as there was no overlap in the projects assessed by the two West et al. studies). The over-crediting ratio is undefined for the points that experienced more deforestation than predicted by their quasi-experimental controls (shown in red). Both panels use log-scaled y-axes.

Next, we estimated the effect of remote sensing differences on reference area deforestation rates. Because project documents did not report deforestation rates in reference areas, we inferred them by multiplying the ACC-derived control area deforestation rates by 0.58, which was the median ratio of bespoke to ACC rates across the 17

projects. Substituting these inferred values reduced avoided deforestation to 0.35 %/year (grey in Fig. 5), suggesting that lower detection of deforestation by bespoke remote sensing (rather than the ACC) was not a cause of over-crediting.

Finally, having adjusted for reference area selection and differences in deforestation rate estimation, we attributed the remaining ~78% of over-crediting to the various *ex ante* modelling approaches used in certified assessments, recognising the considerable uncertainty surrounding this allocation.

## Discussion

Researchers of conservation impact evaluation, including our co-authors, differ in how strongly they view carbon markets as essential to closing the forest finance gap and tackling climate change[12,32–36]. Regardless of this, if forest conservation carbon credits are being traded, then it is crucial that one credit reflects the amount of additional deforestation avoided that is claimed. We interpret our findings with that challenge in mind.

By synthesising the results of multiple independent quasi-experimental analyses, we find that most REDD+ projects did reduce deforestation compared to credible counterfactual estimates, demonstrating a tangible contribution to forest conservation. However, the combined evidence from the six studies we have assessed highlights that these projects issued far more credits than were justified. These credits delivered fewer emission reductions than claimed; to the extent they were used for offsetting, this is likely to have undermined progress towards climate goals[32]. Nonetheless, they were not necessarily bad projects: many achieved measurable on-the-ground conservation benefits.

While some commentators have questioned if over-crediting in first-generation REDD+ projects was a problem[26,27,29], the scientific evidence is now clear. Our synthesis indicates that across the portfolio of projects examined, almost 11 times more credits were issued than was justified. However, most of this excess was driven by a small number of projects that generated the largest volumes of credits (top right in Fig. 2b). After excluding the 9 biggest issuers, over-crediting among the remaining projects was much lower, around 4.0 times, though still substantial. This pattern highlights the need for methodological advances that prevent the most extreme cases of over-crediting that disproportionately shape the overall integrity of REDD + .

One counterargument made by some commentators is that over-crediting is an artifact of lower detection of deforestation by the globally available datasets used in independent quasi-experimental evaluations compared with the bespoke layers used by REDD+ projects[29]. Several studies have found that locally trained remote sensing products can be more accurate in measuring forest cover and deforestation than global layers, such as the ACC[37–39]. However, we found no evidence that the widely used ACC layer systematically detected less deforestation. In fact, it detected similar or higher rates in project areas than certified estimates using bespoke layers. This may have occurred because the ACC is a multi-temporal approach that considers all disturbance events occurring since the start of the time series[5] whereas bespoke approaches typically detect non-forest pixels at two points in time, potentially omitting short-term disturbance and regrowth events[40]. Expanding the forest cover definition to include degraded forest and counting only long-term transitions to non-forest produced lower deforestation rates (see supplementary Figs. 3–6). Including forest classified as regrowth at the project start had almost no effect due to the rarity of this class within projects. The results of considering different forest cover definitions and different duration disturbance events are presented in the supplementary information (Supplementary Figs. 3–6 and 20–23). We could not examine these differences in reference areas, because project documents rarely report deforestation rates there[41], but classification errors are unlikely

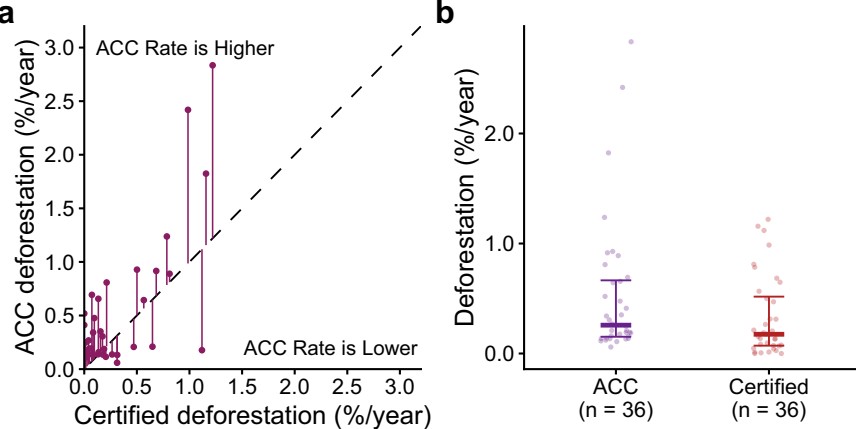

**Fig. 3 | Comparison of deforestation measured by the ACC and bespoke remote sensing. a** *Ex post* annual deforestation rates in 36 project areas measured using the European Unions' Annual Change Collection (ACC) versus the bespoke forest cover layers used in certification. The identity line indicates where estimates are equal.

**b** Median deforestation rates of these projects are significantly higher when measured by the ACC (one-tailed Wilcoxon paired signed-ranks, $n = 36$, $p = 0.001$). Error bars show the median and interquartile range. In both panels points represent individual REDD+ projects.

to differ systematically between project and reference areas unless they differ markedly in terms of species composition or fragmentation.

So why might over-crediting have occurred? One possible mechanism is that projects were situated in areas at low risk of deforestation, whereas reference areas faced greater threats[7,42]. Indeed, there is good evidence that site-based and policy interventions tend to target areas of disproportionately low risk[43,44]. Our results confirm that reference areas were more exposed to known drivers of deforestation than project areas and experienced about 1.7 times more deforestation than quasi-experimental controls.

A second, potentially complementary mechanism arises from the flexibility in model choices available to projects for making *ex ante* predictions of counterfactual deforestation. Such prediction is inherently uncertain, especially when involving spatial or non-linear components[45,46]. Projects have been shown to select methodologies that produce higher estimates of avoided deforestation from among those approved for certification[11]. We extended this analysis by contrasting multiple certified and quasi-experimental estimates (Supplementary Note 2). For most projects, the methodology used for credit issuance substantially overestimated avoided deforestation, but alternative or differently parameterised certification methodologies were available which could have produced more credible estimates. For example, across four projects, VM0006 produced estimates not significantly different from quasi-experimental results (Supplementary Table 2). We conclude that the methodological flexibility available for project proponents and certification bodies, both of whom have financial incentives to produce greater numbers of credits[47,48], provided an additional mechanism contributing to over-crediting in first generation REDD+ projects.

Our study has several limitations. Firstly, quasi-experimental methods for project impacts on deforestation are difficult to verify because the counterfactual, how much deforestation would have occurred without the project, is unobservable. Quasi-experimental methods are advancing all the time and methods exist which combine features of difference-in-differences and synthetic control, such as interactive fixed-effects, augmented synthetic controls and synthetic difference-in-differences[49–51]. However, simulated landscapes where deforestation rates are known have been used to demonstrate the reliability of the approaches used in the studies we synthesize[52,53].

Secondly, the credibility of quasi-experimental designs requires that all meaningful confounders are controlled for[15] which requires an understanding of the reasons some areas became REDD+ projects and others did not. Guizar-Coutiño et al.[25] explore the potential influence

of hidden confounders and reveal they would need to have an effect size multiple times larger than the most impactful observed covariates to undermine the evidence of over-crediting. A potentially important hidden confounder in our analysis is the economic value of land, which we proxied using accessibility (travel time to healthcare in 2019[54]). This covariate has weaknesses that could result in the selection of inappropriate controls. While healthcare is often synonymous with certain size population centres, these places are not necessarily the same as processing or transportation hubs for timber or agriculture. Equally, most projects pre-date 2019, meaning transportation patterns are partly determined by project activities.

Finally, REDD+ projects can cause spillover effects (leakage) that might influence deforestation rates in control units. While some studies have attempted to overcome spillovers driven by localised changes in markets by only matching to areas beyond a certain distance from project boundaries[23,24], this strategy may be insufficient because of the complex manner in which spillovers can occur[55].

The first-generation REDD+ methodologies evaluated here are now being replaced by jurisdictional approaches (JREDD+) for voluntary and compliance markets. JREDD+ methodologies (specifically VM0048 and ART TREES) still make *ex ante* counterfactual predictions but these are now based on the mean annual deforestation rate across the jurisdiction during a 5-6 year reference period[56,57]. By eliminating project-selected reference areas, a major source of bias has been removed. Crucially, the analysis is performed by independent data providers with no direct financial stake in the number of credits issued, which is also a substantial improvement. However, reliance on *ex ante* predictions still carries risk: it assumes deforestation drivers remain constant between the historic and project periods, which is rarely true. Factors such as forest conservation or trade policies, major infrastructure projects or extreme climatic events will vary through time with important effects on deforestation. *Ex post* measurements of deforestation in untreated jurisdictions differ by as much as 100% from *ex ante* predictions[58,59]. Random enrolment, without regard for future deforestation trends, would not result in systematic bias. However, if jurisdictional enrolment is more likely when there is an expectation that historical rates are about to fall, or if the anticipation of enrolment stimulates temporarily increased deforestation, systematic over-crediting is again a risk[60,61]. One final consideration is that VM0048 permits nested projects with baselines produced using *ex ante* spatial modelling, which we have shown to be problematic.

A potential solution is to only issue credits *ex post* based on assessments which integrate developments in quasi-experimental

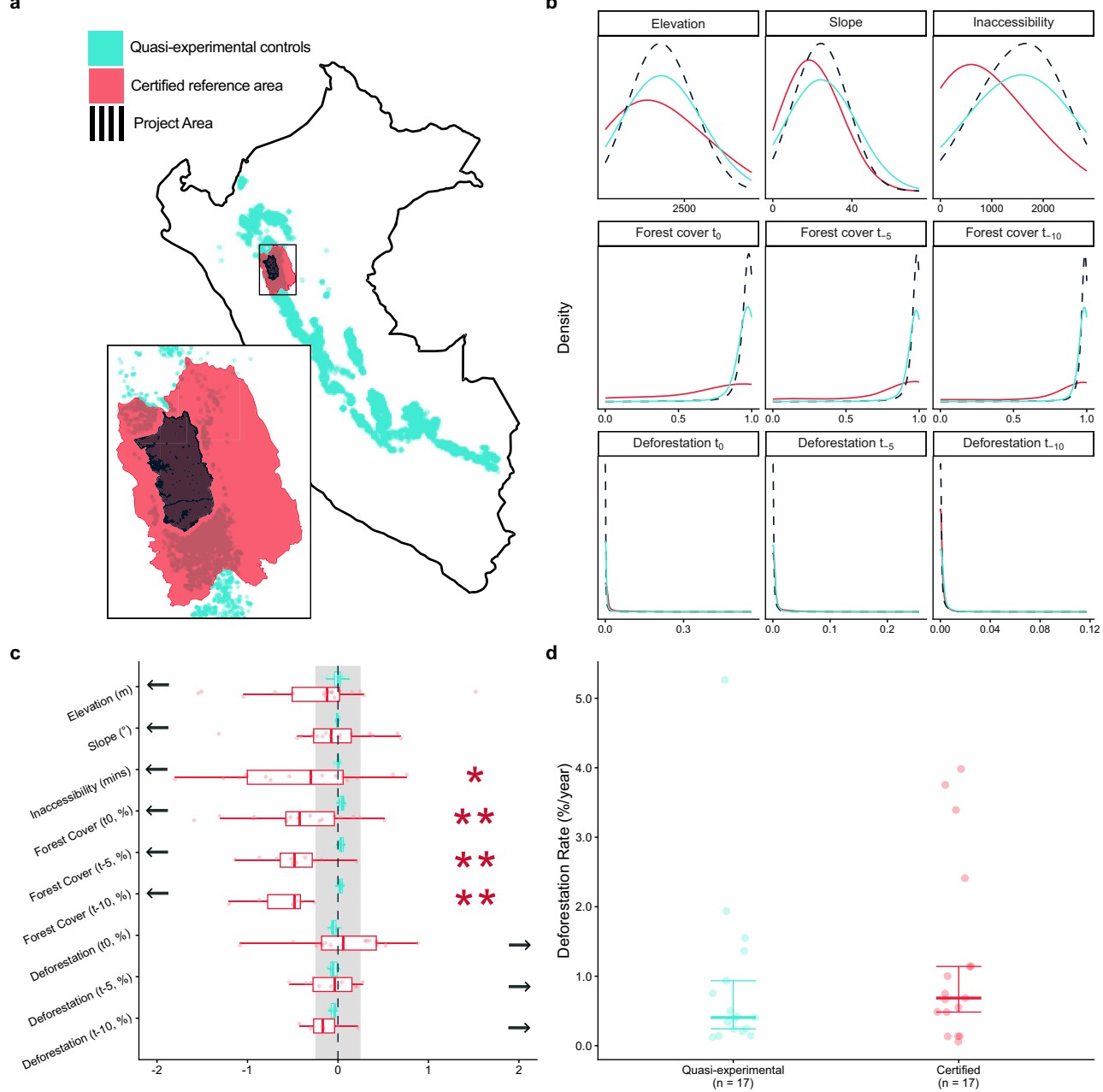

**Fig. 4 | Differences in exposure to deforestation drivers and resulting deforestation rates across project, reference and control areas.** Example from Project 958 in Peru. **a** Geographic locations of the project (black), reference area (red) and quasi-experimental controls (green). **b** Univariate frequency distributions of observable confounders which may influence deforestation, showing discrepancies between the reference area and the project in elevation, inaccessibility and forest cover. Summary for the 17 projects with mapped reference areas. **c** Standardised mean differences from the mean project values for each observable confounder across the quasi-experimental controls or certified reference areas of the 17 projects. For forest cover and deforestation at $t_{-5}$ and $t_{-10}$, only 10 and 6 projects were included, respectively, due to the lack of historic data for certified reference areas. Boxes show the median and interquartile range and whiskers show a maximum of 1.5 times the interquartile distance from the first and third quartiles. Arrows indicate the direction of change associated with increased deforestation. Project-level significance is indicated by values beyond the grey band (−0.25 to 0.25) and across-project significance is assessed by 2-sided t-tests with no adjustment made for multiple testing. Asterisks show significance at $p < 0.05$ (*) and $p < 0.01$ (**). Reference areas were significantly more accessible ($n = 17$, $p = 0.028$), less forested at $t_0$ ($n = 17$, $p = 0.007$), $t_{-5}$ ($n = 10$, $p = 0.003$) and $t_{-10}$ ($n = 6$, $p = 0.008$) than projects. **d** Median and interquartile ranges of deforestation rates of 17 projects measured by the ACC in quasi-experimental control and reference areas during the project period. In (**b**, **c**) the timing of measurements is expressed relative to the project start year ($t_0$) for quasi-experimental control areas, or the start of the reference period for reference areas.

methods[34,36]. Ideally, different quasi-experimental results from methods such as the Permanent Additional Carbon Tonne (PACT) method used in our study or advances in synthetic controls that require fewer assumptions should be considered[49,50,62]. Of course, there will continue to be uncertainties in project-level estimates, but these can be ameliorated by buying credits from multiple projects or reducing the number of credits issued to safeguard any claims being made[35,63]. Quasi-experimental crediting would not alter the timing of credit issuance, as even methods that use *ex ante* counterfactuals still rely on *ex post* measurements within projects to quantify additionality, while

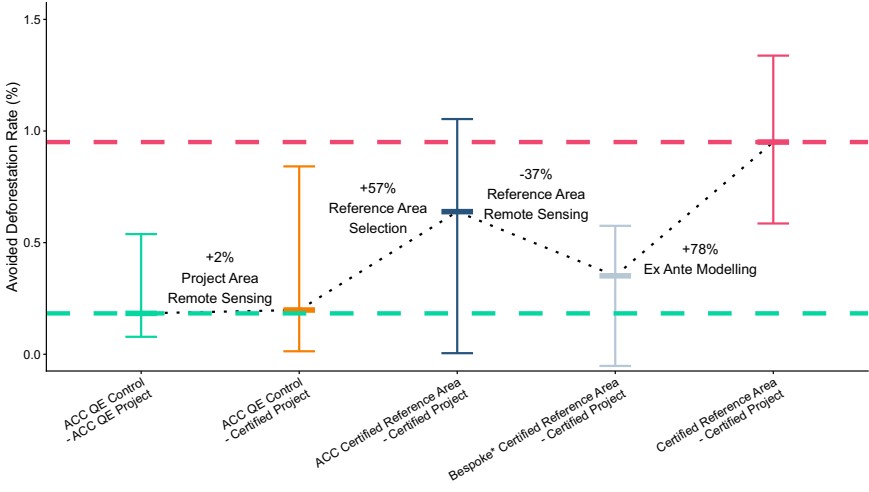

**Fig. 5 | Contributions of different mechanisms to over-crediting.** Quasi-experimental (QE) estimates of avoided deforestation (green) are significantly lower than certified assessments (pink; one-tailed Wilcoxon paired signed-ranks test, $n = 17$, $p = 0.002$) across a subset of 17 projects for which all necessary data are available. Substituting certified deforestation rates for project areas into the QE assessments produces a small increase in avoided deforestation (orange). A much larger increase occurs when certified reference areas are used instead of QE control areas, both measured by the European Union's tropical moist forest annual change collection (ACC; dark blue). Avoided deforestation is lower when deforestation rates inferred from bespoke remote sensing products are used instead of ACC values (grey). The remaining discrepancy is attributable to *ex ante* modelling. These comparisons reveal the relative contributions of each mechanism to overall over-crediting. Error bars represent medians and interquartile ranges.

greatly improving transparency[41]. These approaches are now being adopted by certification methodologies for afforestation, restoration and revegetation (ARR)[64] and improved forest management (IFM)[65]. Uncertainties in credit yields, due to the unpredictable nature of the counterfactual, along with the implications of over-crediting for existing projects would seem to be reasons such *ex post* approaches have not yet been applied despite their potential role in ensuring the future integrity of forest carbon credits[36].

While the first-generation of REDD+ projects achieved far less than claimed, many nonetheless reduced deforestation. Given the high costs and modest progress towards deploying engineered carbon capture and storage, there remains a role for forest-derived carbon credits on the path to net zero[66,67]. The challenge is to ensure that claimed impacts reflect real outcomes. Removing methodological flexibility, particularly the use of project-selected reference areas and *ex ante* modelling, is a crucial first step, but embedding quasi-experimental evaluations would ensure issued credits represent truly additional reductions in deforestation. Either way, far fewer credits would be issued to projects using these approaches, meaning prices will need to rise to pay for the genuine cost of equitable projects with real climate benefits[68]. Bridging the forest-finance gap will require abandoning expectations of low-cost carbon mitigation and paying the true cost of credible mitigation.

## Methods
### Certified estimates of avoided deforestation
Certified rates of avoided deforestation, used to generate carbon credits, were sourced from the design and monitoring documents of 44 REDD+ projects accessed through the Verra registry between April and June 2023. These include the loss of all areas that had been forest for at least 10 years prior to the establishment of the project. Reference areas were selected so that they had similar characteristics to projects in terms of forest cover and expected drivers of deforestation (see Table 1). Forest cover was classified using cloud-free satellite images and bespoke remote sensing approaches. *Ex ante* estimates of counterfactual deforestation were estimated using a variety of different modelling approaches according to the methodology used. Further detail on certified project monitoring and counterfactual estimation is presented in Supplementary Note 1. Projects included in the study were those that had sufficient publicly available data, including geospatial polygons of project areas and had reported the area of avoided deforestation in their most recent monitoring reports (Supplementary Table 1).

Certified project documents did not report deforestation consistently; however, the *ex post* observed deforestation within the project and the *ex ante* modelled deforestation under the counterfactual scenario were available. Verra projects refer to the counterfactual scenario as the 'baseline'. Importantly, deforestation was provided as a total amount in hectares over the whole evaluation period or broken down into annual amounts. Where avoided deforestation was not reported, it was calculated by subtracting project deforestation from counterfactual deforestation.

For Q1, the mean certified annual avoided deforestation in hectares was used. For Q2-Q4, for each project, the total amount of avoided deforestation claimed was disaggregated across the evaluation period (which had a mean length of 6.4 years) to produce annualised compound avoided deforestation percentage rates (see below). Taking this approach ensured that the avoided deforestation used was the same as that used to issue credits.

Projects issued credits using four different Verra methodologies (VM0006, VM0007, VM0009, VM0015) to estimate counterfactual outcomes. The different methodologies are summarised briefly in Table 1 and the process common across these methodologies is described in Supplementary Note 1. The extracted project and counterfactual deforestation rates as well as the certified avoided deforestation are presented in Supplementary Tables 3-4.

### Quasi-experimental estimates of avoided deforestation
We compiled data from six *quasi-experimental* analyses which estimated *ex post* counterfactual and project outcomes for REDD+ projects (Table 2). These included work published by West et al.[7,9] (two analyses), Guizar-Coutiño et al.[8]. and Tang et al.[23].

We also included results from a study under review by Guizar-Coutiño et al.[25]., which implemented a novel framework to reanalyse the set of projects examined in Guizar-Coutiño et al.[8]. study. In their study, 7.1-ha circular sample regions were used instead of 0.09 ha

pixels to characterise forest loss observations. Guizar-Coutiño et al. [25]. assess the causal impact of REDD+ projects on deforestation by adjusting for observed confounders and pre-treatment deforestation trends, using a two-stage matching approach, commencing with propensity score matching via random forests, followed by propensity score subclassification on the matched datasets[69]. They assess the average treatment effect of the treated using multiple model specifications with different algorithms, parameters and covariates. The avoided deforestation estimates used in our study correspond to the doubly robust model specification reported in this study.

Finally, we used the Permanent Additional Carbon Tonne (PACT) v2 method[24] (described below) to produce additional estimates and to explore the mechanisms that may have been responsible for over-crediting. In each case, we included projects for which certified estimates were also available.

Avoided deforestation estimates from quasi-experimental methods were received from the study authors as project level estimates of cumulative avoided deforestation over a defined study period. Annual avoided deforestation in hectares was calculated by dividing cumulative avoided deforestation by the total number of years of the evaluation, to compare estimates produced over different time periods. Evaluation periods were 5 years for both Guizar-Coutiño et al. studies[8,25] and a mean of 6.0, 7.6, 9.0 and 8.1 years for the West et al. [7], West et al. [9], Tang et al. [23] and PACT[24] studies, respectively.

Deforestation was assessed using publicly available remote sensing products for tracking changes in forest cover, specifically the corrected MapBiomas[70] dataset used by West et al. [7], the Global Forest Change (GFC)[71] dataset used by West et al. [9] and Tang et al, [23] and the Tropical Moist Forest Annual Change Collection (ACC)[5] used by Guizar-Coutiño et al. [8,25]. and PACT[24]. GFC reports deforestation as the year of gross forest loss.

The ACC dataset[5] classifies each 30 m Landsat pixel within the humid tropics as either undisturbed forest, water or non-forest in 1990. Then disturbance events were classified according to their duration, as either degradation ($<2.5$ years) or deforestation ($>2.5$ years). If forests subsequently recovered, after a period of at least 3 years, this was classified as regrowth. Some disturbed forests and plantations were erroneously classified as undisturbed forest due to a lack of availability of a longer time series of observations and because their spectral characteristics were similar to those of natural forests. This is particularly problematic in Africa due to limited Landsat coverage before 2000.

The Guizar-Coutiño and PACT studies considered deforestation to be degradation or deforestation of the undisturbed class, even if regrowth subsequently occurred. Degradation was included because it has been shown to result in biomass reduction of $>50\%$ within 12 months. Our analysis of above ground biomass densities for different ACC cover classes indicates that both degraded and regrowth classes had $<50\%$ of the above ground biomass of the undisturbed class (see Supplementary Table 5 and Supplementary Fig. 24). Therefore, for the mechanistic component of our study, we followed the PACT methods by defining forest cover solely as the undisturbed class within the ACC dataset. We explore the differences that arise using different definitions of forest cover in supplementary figs. 3–6 and 20–23. We chose to focus our assessment on avoided deforestation because it lies at the centre of the REDD+ controversy, while recognising that estimating carbon fluxes is an additional, important and complex topic.

## The PACT method
The PACT method used a bootstrapped one-to-one pixel (0.09 ha) matching approach to produce paired project and control units from which to estimate mean project and counterfactual outcomes[72,73]. Project areas, start dates and the date of the most recent monitoring period were obtained from the VCS registry (accessed between April and June 2023). Projects were then assessed following four main steps:

(1) compile layers for characterising project units; (2) determine the set of suitable untreated units; (3) form the control set by matching untreated and project units; and (4) measure and average outcomes across project and control sets (Supplementary Note 1).

First, geospatial raster layers were compiled to track changes in forest cover. The ACC dataset[5] was used for this purpose. We used the 2021 version, a 30 m resolution (0.09 ha) time series spanning December 1990 to December 2021, classifying pixels as one of six classes: undisturbed, degraded, deforested, restored, water and other.

Secondly, layers were compiled to select control units analogous to project units using a method informed by a causal model of land use change. This model incorporated covariates that captured the effects of local policies and regulations, economic pressures, historical changes in land cover and environmental conditions on deforestation rates and conservation efforts. We used the following layers: International country borders (OpenStreetMap[74]) to encompass national regulatory limits; ecoregions (Resolve[75]) to ensure biotic equivalence; and elevation (NASA's Shuttle Radar Topography Mission[76]) as a proxy for abiotic conditions. From the elevation layer, we calculated slope using GDALDEM v3.8.4 (default settings), an important proxy for accessibility and suitability for economic activities. This produced a slope variable consisting of integer values representing the mean slope (0-90°) in each raster cell.

Motorised accessibility to healthcare in 2019 (Malaria Atlas Project[54]) was used as a proxy for market exposure/economic pressure. This accessibility layer integrates all known travel routes, including rivers, along with terrain, land cover and road quality to assess the travel time to population centres large enough to have a healthcare facility. Finally, we computed the proportional cover of undisturbed and deforested classes from each year of the ACC time-series (1990-2021) to assess land-use change trajectories in the surrounding area through time. Proportional cover was calculated by counting the number of pixels in each class and dividing by the total number of pixels within the 1 km radius neighbourhood around each pixel.

It is important to note that accurate data regarding land value and accessibility to markets is not universally available, particular in frontier landscapes. As such we use spatial proxies, specifically proportional cover and motorised accessibility to healthcare, recognising that these leave space for unobserved confounders.

Finally, to minimise interference arising from local leakage, we excluded control units within the vicinity of any REDD+ project. This is important because, under the assumption that leakage arises through the reorganisation of local supply chains or markets, there would be concentrated interference at short distances from projects. To do this we produced a binary raster indicating the presence or absence of a REDD+ project within 5 km. REDD+ polygons were accessed from the VERRA registry and supplemented by any REDD+ project polygons shared directly with the Cambridge Centre for Carbon Credits. All layers were reprojected to the same coordinate reference system as the ACC.

Projects were sampled using a spatial grid with a density of 0.25 points per ha for projects smaller than 250,000 ha and 0.05 points per ha for larger projects (to reduce processing time). This resulted in a mean of 17,397 thousand units across the projects (min = 687, max = 44,697). The characteristics of these sample points were extracted from the layers described above. Time-varying characteristics, including the ACC pixel class and proportional cover were extracted at the start of the project ($t_0$) and at five ($t_5$) and ten ($t_{10}$) years prior.

The domain of untreated units was defined as all pixels within the same countries and ecoregions as each project, located at least 5 km from the project boundary, but no further than 2000 km away. Leakage effects caused by the displacement of production could be present but were assumed to be small, given the large domain of the untreated units and the relatively limited effects of individual projects. To reduce computation time, untreated units were filtered to include only those within the range of values observed in the project area for each known

driver of deforestation, with an added tolerance of ±200 m for elevation, ±2.5° for slope, ±10 min accessibility and ±10% points for proportional cover.

Matching proceeded by taking a random sample of 10% of the project units, which were then processed sequentially to identify control units with the smallest Mahalanobis distance across the continuous characteristics and identical values for country, ecoregion and ACC land cover class at $t_0$, $t_{-5}$ and $t_{-10}$[77]. This matching approach is referred to as 'greedy' because the algorithm sequentially finds the best pairs, which are then removed from the pool for subsequent matching.

The standardised mean difference (SMD) between the control and project sets for each of the continuous characteristics was calculated as follows:

$$\text{SMD} = \frac{\mu_c - \mu_p}{\bar{\sigma}} \quad (1)$$

Where $\mu_c$ is the mean of the control set, $\mu_p$ is the mean of the project set and $\bar{\sigma}$ is the square root of the mean of the variances. SMD values outside the range [−0.25, 0.25] for any continuous characteristic were considered statistically different and these samples were excluded from further analyses[14,77]. This process was repeated 100 times to produce matched project-control sets, each comprising 10% of the total points sampled from within the projects.

*Ex post* deforestation was measured for the project and control units by calculating changes in the undisturbed class since the start of the project. To calculate the area of undisturbed forest at each time point, we computed the proportion of all units in the undisturbed class and multiplied it by the project area. This process was repeated across all 100 matched sets to produce estimates of project and counterfactual forest cover.

Averaging across these sets, we produced annualised mean forest cover (in hectares). By subtracting counterfactual forest cover from the project forest cover, we derived annual cumulative avoided deforestation values. Total avoided deforestation during the project period was measured at the final year of the project's most recent monitoring period, as additionality figures are cumulative.

### Q1 How consistent are estimates of project impacts assessed using independent quasi-experimental methods?

For each of the 44 projects, we compiled the certified estimates of annualised avoided deforestation (in hectares) along with all available quasi-experimental estimates. For the 42 projects with more than one quasi-experimental estimate, we calculated the mean, standard deviation, coefficient of variation, standard error and 95% confidence intervals. For the remaining two projects, which had only a single quasi-experimental estimate, this value was taken to be the mean.

To test for over-crediting, we determined how many projects had mean quasi-experimental estimates below their certified estimate, which was deemed significant at the project-level if their upper 95% confidence intervals were less than their certified estimates. To test whether certified estimates were systematically greater than mean quasi-experimental estimates, we applied a one-tailed Wilcoxon signed-ranks test, due to the non-normality of the data. We also evaluated how many projects had avoided a significant quantity of deforestation by testing if the lower bound of the 95% confidence intervals were greater than zero across quasi-experimental studies.

We next assessed over-crediting as the ratio of the certified to the mean quasi-experimental estimate of avoided deforestation. We refer to this as the over-crediting ratio, which expresses how many times more credits were issued through certification than suggested by quasi-experimental estimates. The mean over-crediting ratio was calculated as the mean of the over-crediting ratios of all projects that had both issued credits and had positive mean quasi-experimental estimates; projects with negative mean quasi-experimental estimates were excluded. The

global over-crediting ratio was calculated by dividing the summed certified avoided deforestation by the summed mean quasi-experimental estimates (including negative values). The mean and global over-crediting ratios respectively represent how many hectares of certified avoided deforestation correspond to one quasi-experimentally assessed hectare for a project selected at random and across the REDD+ portfolio. Because the data were non-normal, non-parametric 95% confidence intervals were produced for the mean and global over-crediting ratios by randomly sampling projects with replacement 10,000 times.

### Determining comparable deforestation rates between certified and ACC sources

We calculated percentage annual deforestation rates to make the quantities of deforestation measured by the certified assessments and the ACC layer (e.g., those produced by PACT) directly comparable. The forest cover ($F_t$) resulting from a constant annual deforestation rate ($r$) after an interval ($t$), given the starting forest cover ($F_0$), was calculated as follows:

$$F_t = F_0(1 - r)^t \quad (2)$$

Because we had extracted the forest cover at the beginning and end of the evaluation period, we calculated the rate by rearranging the formula to:

$$r = 1 - \left(\frac{F_t}{F_0}\right)^{\frac{1}{t}} \quad (3)$$

To determine $F_0$ we took the proportion of the ACC pixels classified as the undisturbed class in the yearly layer closest to the project start date and multiplied this by the total project area derived from the project polygon. Thus, $F_0$ was the total area of undisturbed forest (in hectares) at the start of the project. $F_t$ was calculated by subtracting the area of deforestation in the certified or ACC measurements from $F_0$. The evaluation interval $t$ was the number of years between the start and end assessments. For the certified assessments, this was calculated from the number of total days between the start and end dates, whereas for the ACC data, this was the number of whole years between the ACC layers used. The calculations for each project are presented in section Supplementary Tables 6–7.

We produced certified and ACC mean annual percentage deforestation rates for 36 project areas and 17 counterfactual estimates. Between these two sets, there was an intersection of 17 projects, for which we could produce mean annual percentage avoided deforestation rates using all combinations of certified and quasi-experimental estimates for project, reference and control areas, sufficient for inclusion in Q3 and Q4.

### Q2 Is the use of global deforestation layers the reason for the discrepancy between quasi-experimental and certified estimates?

We used the annual deforestation rates to test if there was a difference in the amount of *ex post* deforestation measured in project areas. Because the data were non-normal, we tested whether the paired differences were significantly greater than zero using a one-tailed Wilcoxon signed-rank test. In the main analysis, we focused exclusively on the deforestation or degradation of undisturbed forests in the ACC measurements. In Supplementary Figs. 3–6 we broaden the forest cover definition to include undisturbed and degraded forest and measure the deforestation of either.

### Q3 Were the reference areas similar to projects in their exposure to deforestation?

We compared the exposure to the drivers of deforestation across project, reference and control areas to assess the extent to which the

selection of reference areas explained differences between certified and quasi-experimental assessments. For reference areas, we focused on the *Reference Regions for Deforestation* (RDD) used by certified methods, as these were used to measure the rate of deforestation, rather than the *Reference Regions for Location*, which were used to model how much of the deforestation was expected to occur within project areas.

Reference area polygons were not publicly available as shapefiles and were therefore digitised by georeferencing maps available in project design documents from the Verra registry. This was done either by tracing the polygons by hand in QGIS (v3.26.3) or through colour thresholding and an automated polygonisation procedure in R, equivalent to the process described by the Environmental Systems Research Institute[78]. Colour thresholding was applied when reference areas presented in project design documents were complex shapes represented by colour-coded systems within the georeferenced maps. The colours representing the reference areas were identified as the most frequent pixel colours in the maps. Binary maps were then produced from these pixels using a Boolean comparison to the reference area colours and their fidelity was checked against the original maps. Finally, the binary maps were converted to geospatial polygons using the *polygonise* function in the *Terra* package in R.

We successfully digitised reference areas for 17 projects (see Supplementary Table 1 for the projects included). We then sampled the characteristics of reference areas using the same PACT method for sampling project units (described above). Time-varying characteristics were sampled from the beginning of the historical reference period (usually 5-10 years before the start of the project), to ensure comparability with project characteristics.

To test the similarity of control and project areas in terms of exposure to deforestation risks, we examined the distributions of key characteristics prior to any measurement of forest loss. For the project areas, these covered the pre-project period up to a maximum of ten years prior. This was the same for the quasi-experimental controls. For reference areas, the period extended up to ten years prior to the start of the documented reference period.

We tested for differences in the univariate distributions of pre-project characteristics using the SMDs between the reference or control areas and the project areas. Values outside the range [−0.25, 0.25] indicated a significant project-level difference[14,77]. Across the set of projects, we tested whether the distribution of the SMDs for each characteristic differed significantly from zero using t-tests.

We also compared the observed annual deforestation rates between reference and quasi-experimental control areas using the ACC dataset. For reference areas, this was measured across the reference period; for quasi-experimental controls, it was measured across the project evaluation period. Because the data were non-normal, we applied a one-tailed Wilcoxon paired signed-rank test to assess whether reference areas were exposed to significantly more deforestation than the quasi-experimental control areas.

### Q4: What is the residual effect of ex ante modelling after isolating the effects of the forest cover layer and reference area selection?

To assess the effect of *ex ante* modelling, we used a process of isolating all possible explanations for over-crediting for the same 17 projects used in Q3. This required determining the annual avoided deforestation rate produced by five different combinations of reference, control and project area estimates.

First, we quantified over-crediting as the overall difference between avoided deforestation from quasi-experimental estimates (PACT in combination with ACC) and certified estimates. To isolate any effect of project area remote sensing within this overall difference, we examined the change in avoided deforestation resulting from substituting quasi-experimental project area deforestation estimates with their certified equivalents.

From this new combination, we then isolated the effect of reference area selection by substituting quasi-experimental control area estimates with ACC-derived estimates for the reference areas (covering the reference period) made in Q3. This third combination, therefore, represents the combined effect of the two mechanisms explored in Q2 and Q3. The residual difference between this combination and the avoided deforestation rates produced by purely certified estimates captures the influence of two remaining factors: (1) the bespoke remote sensing measurements made in reference areas; and (2) the *ex ante* modelling of reference area deforestation to predict counterfactual outcomes for project areas.

Although we could not disentangle these explanations, we examined whether differences between remotely sensed forest cover layers affected the magnitude of the reference area effect. To do this, we inferred the deforestation rates that would have been observed if bespoke measurements in control areas occurred at the same relative proportion of ACC-measured deforestation rates observed in project areas. We divided certified estimates by quasi-experimental estimates in project areas and took the median of these ratios to produce a correction coefficient. We then multiplied the ACC measurements in reference areas by the coefficient to generate corresponding 'bespoke' estimates, from which we produced our final possible avoided deforestation rate. This yielded two possible residuals covering a range of impact attributable to *ex ante* modelling.

Statistical differences between ranges of avoided deforestation were tested using a one-tailed Wilcoxon paired signed-rank test because paired differences were not normally distributed.

### Reporting summary

Further information on research design is available in the Nature Portfolio Reporting Summary linked to this article.

### Data availability

All data generated in this study have been deposited in the Zenodo database under accession code zenodo.org/records/18715093. Avoided deforestation area calculations, compound annual deforestation rates and above-ground biomass densities for all the projects included in the relevant analyses are provided in the supplementary information.

### Code availability

All analyses were undertaken in R (v4.2.1) using Terra (v1.7.65), Simple Features (v1.0.15) and Raster (v3.6.26) for geospatial processing; Vegan (v2.6.4) for ordination analysis; Tidyverse (v2.0.0) for data manipulation; and Natural Earth Data (v1.0.0) for country borders. In an effort to contribute to improved transparency, we have made the code necessary to run the PACT evaluations (github.com/quantifyearth/tmf-implementation and https://zenodo.org/records/18712812 [79]) and our analysis (github.com/quantifyearth/REDD-Over-Credit-Reasons and zenodo.org/records/18715093[80]).

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

## Acknowledgements

This study was funded with support from the Tezos Foundation (grant NRAG/719; AEW, TS, JHolland, JHartup, MOKL, PF), John Bernstein (donation; SJ) and Tarides (donation; MD). The authors thank 4 anonymous reviewers and Lynsey Stafford for their valuable comments on the manuscript. For the purpose of open access, the authors have applied a Creative Commons Attribution (CC BY) license to any Author Accepted Manuscript version arising from this submission.

## Author contributions

Conceptualisation: A.B., A.M., AEW, D.C., S.K. and T.S. Investigation: A.B., A.M., AEW, JHolland, JHartup, M.D., M.O.K.L., P.F., S.J., S.K. and T.S. Visualisation: AEW, JHolland, JHartup, S.J., and T.S. Funding acquisition: A.B., A.M., S.K., T.S. Project administration: A.B., A.M., E.T.S., D.C., S.J., S.K. and T.S. Writing—original draft: A.B., JHolland, J.P.G.J. and T.S. Writing—review and editing: A.B., AEW, A.G.C., A.M., D.C., E.T.S., JHolland, JHartup, M.D., P.F., J.P.G.J., M.O.K.L., S.J., S.K., T.A.P.W. and T.S.

## Competing interests

The Cambridge Centre for Carbon Credits (4 C) has no commercial interest in carbon credits. T.S. has an advisory position with Symbiosis, an initiative to finance carbon storage in nature. The remaining authors declare no competing interests. A.B., A.M., D.C. and S.K. are trustees and E.T.S. and T.S. are founders of Canopy PACT, an initiative to integrate science into nature-based credit markets.
