## [Transparent Peer Review file · Nature Communications]

Learning lessons from over-crediting to ensure additionality in forest carbon credits

Corresponding Author: Dr Tom Swinfield

Version 0:

Reviewer comments:

Reviewer #1

(Remarks to the Author)

This paper presents a compelling assessment of the level of additionality calculated in multiple REDD+ analyses by certification agencies compared to an assessment of the same projects using statistical methods. The purpose of this is to help assess the level of overcertification and point the way towards a more constructive process in the future. I appreciate the effort and approach, but I have a few comments that I think may help clarify the analysis and policy recommendations and also temper some of the conclusions.

Main comments:

1. Measurement error in deforestation data: I really like the general approach of trying to separate out matching quality from remote sensing error. However, I am concerned about two things related to the quality of the remote sensing data. First, the measurement error in MapBiomas and the GFC product is significant. A number of papers document serious underestimation of deforestation in GFC (Burilavova et al. 2015, Mitchard et al., 2015). I have not seen such studies for MapBiomas, but I do know that the estimations of change are sometimes revised downwards in subsequent versions of the series. This is not to criticize these products, but rather to note that it is quite possible that this error creates significant problems for statistical assessment of impacts. If the measurement error is classical, this may attenuate your point estimates. If it is non-classical, the direction of the bias is unpredictable.

Further, when estimated rates of deforestation are low, this puts a ceiling on potential impacts. You might note this point somewhere in your description of your statistical assessments.

You provide little information on the “bespoke” remote sensing products used by certifiers. It is likely that these also have significant error, but if they are created at a lower resolution than 30m, they probably more accurately assess deforestation in low income, tropical settings where clearing takes place in small patches. You resolve part of this by noting that you find certified deforestation relative to your own assessment to be higher in reference areas than in program areas. However, it feels like there need to be error bars on this assertion. The error bars that you have accommodate the error in the data used for statistical analysis. It seems like there should also be an accommodation for the error in the certification data. If standard deviations for their estimations are available, then perhaps you could use these to adjust the error bars in your figure 1b, which has error in both the numerator and the denominator.

2. Clarity on methodology:

a. It would be nice to have more clarity on the “statistical approaches” used in the paper. I am particularly interested in the covariates. The paper uses different “confounders” and approaches in every assessment. This makes it hard to figure out what is actually happening. To my mind, if you are trying to propose an alternative methodology, then you should propose something that is replicable and consistent. For example, you might want an approach that uses similar covariates across multiple projects. One can always, with refined data and endless iterations of increasingly complex statistical approaches, come up with much better estimates than are done with more coarse, globally consistent data. Further, hindsight, as they say, is 20/20. Ex post estimates will always show something different than ex ante predictions. In addition, given what you have demonstrated about the cherry picking of reference areas towards those that have higher levels of deforestation, one

wouldn't want to leave a huge number of degrees of freedom in the way that reference areas are selected. It seems unreasonable to suggest that certifying institutions should be conducting analysis with the same number of robustness checks that academic researchers do. We need to provide guidance on what might be both effective and feasible.

b. It would be useful to describe in the appendix the ways in which VERRA is estimating their counterfactual areas. Creating a deforestation risk map based upon a number of covariates and then selecting areas of similar baseline risk is quite similar matching. Your analysis of covariate similarity suggests that they may not be doing a great job on this, but it is hard to tell because we don't really know what they are doing.

Further, I have the impression from Table 1 that the VERRA estimates of deforestation are based upon predictions rather than on observed deforestation. Is that correct? In this case, it seems like a major source of the problem in the assessments could be that the prediction is not very good, not that the area of comparison is poorly selected. Perhaps you can clarify what is actually happening here.

3. Clarity of policy recommendations:

I think that a considerable amount of work could be done to make the policy recommendations of this paper more clear and helpful. Below are some suggestions.

a. Some constellation of the authors in this paper have already established that there is an issue with over certification on forest carbon markets. It would be useful to understand what the first result of this paper adds there that is new.

b. The main constructive recommendation of the paper is that ex post statistical approaches be used in carbon crediting. As I was thinking about this, I got confused about what is actually being done versus what is proposed. Are the authors suggesting withholding payments until assessment of avoided deforestation can occur ex post? This seems like an impossible proposition, both at the broader institutional level and at the level of producer participation in the event that projects involve individual land users foregoing deforestation, which they must do for there to be additionality. Recent research suggests that successfully implementing PES in low-income settings may require ex ante payments to actually induce wide-scale participation (Izquierdo-Tort et al. 2024). It therefore seems more reasonable to suggest a way to consistently apply matching or the construction of synthetic controls ex ante using data that is generally available and covariates that are known to be correlated with both selection into treatment and deforestation outcomes. I think that is what is being suggested here. Counterfactual deforestation could then be predicted for reference areas, and then once time has passed actual deforestation rates would be used in assessment. I think what I would like to see is clarification of the policy proposition.

c. Two implementation issues should be recognized in the discussion.

i. The first of these is that perfect prediction of the location of future deforestation is impossible. Deforestation is a rare event statistically, and so targeting programs such that only future deforested land is enrolled does not make sense, regardless of how much data you have with which to make the prediction. Even trying to approach this kind of targeting in a REDD+ program would require taking money away from actual incentives and funneling it into bureaucratic processes intended to support targeting, which would be wasteful. This means that you have to pay for enrollment of larger areas of land than are likely to be deforested.

ii. This does not in itself undermine the principle of this paper. However, it is worth considering that the global community often says that avoided deforestation and, more recently, forest restoration, is "cheap." The perceived cheapness of the intervention may be at least partly driven by the fact that when we calculate REDD+ transfers per hectare using current estimates of deforestation, they are very low. However, we might find that if we use current per hectare rates spread over a much lower, precisely estimated amount of avoided deforestation, we may not be able to induce any participation in these types of programs, because they will not sufficiently compensate producers for their opportunity cost. Environmental non-profit organizations and rich country governments often engage in wishful thinking regarding the true costs of nature-based solutions. It is important that the research community does not fall prey to this same mistake.

Smaller issues:

I believe that the acronym JRC TMF is not defined anywhere.

Works cited:

Burivalova, Zuzana, Martin R. Bauert, Sonja Hassold, Nandinanjakana T. Fatroandrianjafinonjasolomiovazo, and Lian Pin Koh. 2015. Relevance of global forest change data set to local conservation: Case study of forest degradation in Masoala National Park, Madagascar. *Biotropica* 47 (2): 267–74.

Izquierdo-Tort, S., Jayachandran, S. and Saavedra, S., 2024. Redesigning payments for ecosystem services to increase cost-effectiveness (No. 21022). https://seemajayachandran.com/pes_redesign_mexico.pdf

Mitchard, Edward, Karin Viergever, Veronique Morel, and Richard Tipper. 2015. Assessment of the accuracy of University of Maryland (Hansen et al.) Forest Loss Data in 2 ICF project areas: Component of a project that tested an ICF indicator methodology. Technical report, University of Edinburgh.

(Remarks on code availability)

Reviewer #2

(Remarks to the Author)

Please refer to the attached document.

(Remarks on code availability)

Reviewer #3

(Remarks to the Author)

(Remarks on code availability)

The link for the data availability did not work.

Reviewer #4

(Remarks to the Author)

The manuscript entitled “Independent statistical approaches address overcrediting problems in REDD+” compares the avoided deforestation rates from ex ante assessments using the methods applied by a major carbon crediting agency with the ex post estimates of statistical approaches for 47 REDD+ projects in Latin America and Africa. The manuscript argues that the ex ante methodology tends to overstate the avoided deforestation by the REDD+ projects due to the selection of often incomparable reference groups.

While the topic of revising REDD+ and quantifying additionality from conservation and climate change mitigation interventions is very timely and important, significant concerns remain about the manuscript regarding the statistical methods employed.

Statistical and model considerations

1. Without perusing the prior studies on which the manuscript is based, it is not clear what the ex post statistical methods employed are: For example, the manuscript mentions synthetic controls and one-to-one matching without replacement (Table 1), but does not provide key details. For example, why not synthetic difference-in-difference models that combine features of synthetic and heterogeneity-robust difference-in-difference estimators¹? These are applicable to small as well as large samples, and can dominate synthetic controls¹. Matching without replacement is used (lines 400-410) but there is no information if the resulting standard errors are clustered at the matched pair level (as they should)². It is also unclear if matching is combined with difference-in-difference techniques when the sample size permits. Further, one of the studies, from which ex post statistical estimates for the current manuscript are used, is still unpublished (line 402); without evaluating the statistical analysis, it is unclear if the estimates therein are credible. Further: Do the studies employing pixels as the unit of analysis (Table 1) use sampling approach that results in the same probability of selection for each pixel? If not, how are the different sampling probabilities accounted for in the analysis? Since the point of the manuscript is to highlight the advantages of statistical methods, details about the statistical methods are crucial, especially for studies not peer-reviewed/published yet.

2. Are the ex ante and ex post estimates statistically different? Fig. 2 and 3 suggest that there may be substantial overlap between the point estimates and the confidence intervals. That is, are the differences driven by a few outliers (e.g., Fig. 2)?

2. The annual average rates of avoided deforestation are not the total deforestation/#years (line 417-419). Instead, some compounding is necessary. Consider switching to effect sizes instead^{3,4}

3. What is the formula for the standardized mean difference used in this study (line 497-499)? There are two versions in the literature (for example, see^{5,6}).

4. The principal components analysis is not traditionally used to assess degree of covariate overlap (e.g., lines 502-510); instead, studies report the degree of overlap for each covariate used in the analysis. Why was the principal components analysis used? Can you reference prior studies that advocate for this method over comparisons of the distributions of each covariate? The approach should be backed up with citations.

5. What are the “likely confounders for the project” (line 500). The only covariates listed are those in Fig. 3d—elevation, slope, inaccessibility, baseline and lagged versions of forest cover and deforestation. Were any other covariates (e.g., ones that capture local economic processes used?) For example, the proximity to rivers is of huge importance in some locations as this is how households transport timber illegally (e.g., 7,8)—was that captured by the inaccessibility variable? No other (e.g., socio-economic) variables seem to have been included. Have the original studies assessed the sensitivity of the

results to unobserved confounders especially when no difference-in-difference models are used?

6. What is the “forest scarcity factor” (line 134). Is it possible it is driving a wedge between the ex ante and ex post estimates?
7. What are the reference sites used in the ex post statistical analysis? For example, the manuscript mentions leakage as a concern (line 53) but it is not clear how that leakage is accounted for in any of the estimations without a proper economic model of change.
9. The calculations of the overcrediting (lines 150-155) are very hard to follow. Please elaborate how you obtained the %.
10. It is unclear what the test on lines 465-471 is trying to achieve. Is there a prior study that proposes or uses this method? Why is this method better than alternatives (e.g., something that imposes an underlying distribution)?
11. It seems that most of the projects assessed are from a few countries (Table S1). Is it possible to look at heterogeneity between countries? That is, it is possible the overestimation of avoided deforestation is country-specific due to country-specific factors.

Data

1. Table 1 and lines 514-534 summarize the data sources used to quantify deforestation. It appears that the ex post and ex ante analyses use different data sources: The methodology employed by Verra uses the Bespoke data, whereas the ex post statistical analyses use publicly available remote sensing products. What remains unclear is to what extent are these datasets comparable? For example, what is the resolution of the data? Do they handle missing values in the same way? What are the error/uncertainty estimates for each—for example, the TMS dataset spans all moist forests; the Hansen data (ref 40)-the whole globe. For this reason, it is likely the two datasets have different errors than a dataset that is produced and groundtruthed locally: <https://www.globalforestwatch.org/blog/data-and-tools/how-accurate-is-accurate-enough-examining-the-glad-global-tree-cover-change-data-part-2/>.

Do the different datasets classify deforestation in the same way? The manuscript alludes that that the ex post statistical studies may not be consistent with each other. First, the Hansen and TMS datasets use different methods and classifications. The TMS dataset defines forest degradation as the temporary (less than 2.5 years) loss of forest cover⁹—is this a consistent definition employed in all of the land cover datasets used to assess avoided deforestation? The Hansen data used in the manuscript (line 129, reference 40) does not define deforestation, but rather “forest cover loss” events. The Hansen data does not define what “forest” is—prior studies suggest employing a 25% tree cover cutoff to define “forests”¹⁰ but the cut-off may be too low for the Amazon but ok for parts of Africa. Was any pre-processing of the Hansen data done in any of the statistical analyses? It also does not allow for easy comparisons before and after 2015 because of different remote sensing classification methodologies used: <https://www.globalforestwatch.org/blog/data-and-tools/tree-cover-loss-satellite-data-trend-analysis/>. Finally, some of the statistical estimates include the switch for undisturbed forest to degradation as “deforestation” (line 433, lines 425-435).

Is a consistent definition of “deforestation” used in both the ex ante and ex post analyses? For example, is the creation of roads considered deforestation? For example, clearing forests for logging roads in logging concessions may not be considered as “deforestation” in the policy world. Forest cover loss associated with traditional slash-and-burn agriculture is also not considered “deforestation” in the policy world. Both temporary clearing and road building will be picked up by the Hansen data as “forest loss events” as well as any event that results in a forest cell having less than 50% tree cover in a given year.

Details on the data used are crucial and should be inserted in the manuscript and the appendix. The credibility of the main claim of the manuscript needs to be beefed up by comparing the data used for ex ante analyses with the data used for ex post analyses.

Writing:

1. Given the heavy reliance on other, mostly peer-reviewed and published studies, that also question the credibility of REDD+ avoided deforestation estimates, it is not immediately clear what the contributions of this manuscript are. Is it a novel comparison with the methods employed by Verra? Is it a meta-analysis? Further, the manuscript mentions that VERRA has introduced new consolidated methods to replace earlier ones (Line 96). Do these new methods still rely on ex-ante forecasts or include ex-post methods like those used in the statistical approaches? Incorporating new approaches could suggest that VERRA acknowledges the weaknesses in their earlier methodology and is working on improving their assessment tools.
2. It is not clear why the additionality ratios can be smaller than 0 (lines 141-149)? Out of the 47 REDD+ projects, did any certified estimates indicate increased deforestation in the project areas? Since the additionality ratio is the statistical estimate divided by the certified estimate (444-446), a negative ratio would imply that either one of the estimates had the opposite sign. If so, why and where was that the case?
3. What is a “binary process” (line 434)?
4. While the Introduction is generally well-written and easy to follow, the rest of the text has frequent grammar and

punctuation errors. There are also very long sentences with interrupting parentheses with additional text (e.g., lines 104-107).

5. The first 3 paragraphs of the Discussion section summarize the main findings of the manuscript. It appears unnecessary: please consider shortening them and avoiding redundancies.

Work cited:

1. Arkhangelsky, D., Athey, S., Hirshberg, D. A., Imbens, G. W. & Wager, S. Synthetic Difference-in-Differences. *Am. Econ. Rev.* 111, 4088–4118 (2021).
2. Abadie, A. & Spiess, J. Robust Post-Matching Inference. *J. Am. Stat. Assoc.* 117, 983–995 (2022).
3. Börner, J. et al. Emerging Evidence on the Effectiveness of Tropical Forest Conservation. *PLOS ONE* 11, e0159152 (2016).
4. Samii, C., Lisiecki, M., Kulkarni, P., Paler, L. & Chavis, L. Effects of decentralized forest management (DFM) on deforestation and poverty in low- and middle-income countries: a systematic review. *Campbell Syst. Rev.* 10, 1–88 (2014).
5. Imbens, G. W. & Wooldridge, J. M. Recent developments in the econometrics of program evaluation. *J. Econ. Lit.* 47, 5–86 (2009).
6. Abadie, A. & Imbens, G. W. Bias-Corrected Matching Estimators for Average Treatment Effects. *J. Bus. Econ. Stat.* 29, 1–11 (2011).
7. Ferretti-Gallon, K. & Busch, J. What Drives Deforestation and What Stops It? A Meta-Analysis of Spatially Explicit Econometric Studies. 44–44 (2014).
8. Miteva, D. A., Loucks, C. & Pattanayak, S. K. Social and Environmental Impacts of Forest Management Certification in Indonesia. *PLoS ONE* 10, e0129675–e0129675 (2015).
9. Vancutsem, C. et al. Long-term (1990–2019) monitoring of forest cover changes in the humid tropics. *Sci. Adv.* 7, eabe1603 (2021).
10. Sexton, J. O. et al. Conservation policy and the measurement of forests. *Nat. Clim. Change* 6, 1–6 (2015).

(Remarks on code availability)

I was not aware there was code to review when I downloaded the files to review.

The review pertains to the intuition, theoretical justification, and interpretation of the analyses included in the main text and the appendix to the manuscript.

Reviewer #5

(Remarks to the Author)

(Remarks on code availability)

Version 1:

Reviewer comments:

Reviewer #1

(Remarks to the Author)

I have uploaded my review as a pdf below.

(Remarks on code availability)

Reviewer #2

(Remarks to the Author)

Please see the attached file.

(Remarks on code availability)

Reviewer #4

(Remarks to the Author)

The paper has the potential to make a significant contribution. However, while the manuscript has improved significantly in terms of the analysis and writing, it is still not ready for publication, in my opinion. There are 2 main remaining challenges: The quality of the writing and the lack of proper description and justification of statistical techniques.

Writing:

While the contributions of the manuscript and the description of the methods are much clearer, grammatical errors and overly complicated sentences still abound in the text.

Does deforestation in the bespoke layer capture forest loss due to road construction? Do the Verra approaches consider reforestation as a way to offset forest loss? It is important to clarify what the original certification mentions consider "deforestation" as some forest loss is not deforestation.

What is the default setting for GDALDEM for slope calculations? Is it degrees? (line 514)

I suggest including citations of the actual datasets used in the analyses (e.g. lines 503-506) and not referencing just the secondary studies that use these datasets

I don't know what the text on lines 625, 279-280, and 59-60 mean. For example, what are "areas used for spatial modeling"?

Why introduce the synthetic control method in lines 90-93? It seems out of place. Also, as outlined in my previous report, newer methods like synthetic difference-in-difference may outperform synthetic controls (Arkhangelskiy et al. 2021).

What is bootstrapped in the PACT method (Table 1 and line 494)? How is the bootstrap done?

I would clarify that Fig 1a is similar to a quantile-quantile plot for log-transformed data. Currently, that figure and its legend are very difficult to read. I would clarify in the axes labels that the outcome is log-transformed.

Reword that you are showing 95% Confidence Intervals instead of "confidence intervals show 1.96 standard errors" (line 174-175)

The presence and location of spillovers depends on the extent of markets (e.g., see Pfaff & Robalino 2017 for a review and Miteva et al. 2017 for an example of a specific mechanism of spillovers as a function of markets). The text mentions that some of the projects are large (lines 526-527). Given the lack of underlying theory of change, the manuscript cannot model where spillovers are likely to occur. However, this point should be acknowledged as a limitation in the discussion.

Reword Q4 (line 256)

I am not sure what lines 285-286 and lines 343-344 mean—a critique of a specific ex ante method? A general critique of ex ante methods?

Fig 5 is very hard to read: Why are there bars that don't correspond to X axis labels?

I am not following the point on lines 334-335. If something is not statistically significant, the coefficient is 0. Unless the point the text is trying to make is that there is a lot of heterogeneity (hence the wide confidence intervals)?

I would emphasize that the credibility of quasi-experimental designs depends on, first and foremost, the theory of change that helps us formulate hypotheses, select relevant to a specific context covariates, select a proper estimation approach, and explain observed patterns (lines 361-365). Also, the robustness of quasi-experimental designs can be assessed via sensitivity analyses (lines 363-364).

What is a "modified method" (line 461)?

Lines 494-496 & 627-635 need a citation to back up the approaches

Which correlation was used --Spearman vs. Pearson (line 566)?

Since F0 is a share, lines 595-596 do not make sense as worded.

The manuscript supports the range for, what it refers to as "standardized mean difference" by citing a recent paper in Conservation Biology (Ref. 22). Given the long history of the statistic in other disciplines, I would reference studies by statisticians or econometricians who propose that cut-off as a rule of thumb (e.g. Abadie & Imbens, 2011; Imbens et al. 2009 but also earlier studies). Also, line 551 references an article in the Guardian and not a statistics study.

Data

It is still not clear to me why the main text of the manuscript uses only "undisturbed" forests in the Vancutsem et al (2021) data and not all forest including forest regrowth (e.g., paragraph starting on line 593). Only a note in passing is made in the text that some analysis using all 4 forest categories can be found in the appendix (although I could not find that--Section S4

uses just deforestation and degradation). The main text does not clarify what the comparison indicates (lines 616-617).

Lines 447-449 mention that whenever deforestation rates were not available, they were “totaled over the evaluation period”. However, later the methods present a compound formula for calculating the average deforestation over time (Eq. 2.2)—why were 2 different methods used?

Statistics:

The credibility of counterfactual analyses depends on the underlying theory of change. Omitting key covariates (e.g., proximity to rivers, markets which may not be necessarily captured by the travel time to health centers etc) may drastically change the control units and, thus, results. For this reason, it is necessary to justify the baseline covariates used in the quasi-experimental design with some theory of change. The current version of the manuscript mentions only a handful, some of which may not refer to the baseline (period prior to the intervention) (lines 503-524).

Why was it necessary to limit the analysis to only 100 matched pairs based on 10% of the data (lines 540-545)?

Why was a t-test used to assess differences in the distributions for the certified areas (line 654), whereas other assessments were based on the standardized mean difference? The issue is that the t-statistic is a function of the sample size: Just increasing the sample size can make a t-statistic statistically significant, *ceteris paribus* (Imbens & Wooldridge 2009). Conversely, a t-test based on a small # observations is likely to have insufficient power.

References

Abadie, Alberto, and Guido W Imbens. “Bias-Corrected Matching Estimators for Average Treatment Effects.” *Journal of Business & Economic Statistics* 29, no. 1 (2011): 1–11. <https://doi.org/10.1198/jbes.2009.07333>.

Arkhangelsky, Dmitry, Susan Athey, David A. Hirshberg, Guido W. Imbens, and Stefan Wager. “Synthetic Difference-in-Differences.” *American Economic Review* 111, no. 12 (December 1, 2021): 4088–4118. <https://doi.org/10.1257/aer.20190159>.

Imbens, Guido W, and Jeffrey M Wooldridge. “Recent Developments in the Econometrics of Program Evaluation.” *Journal of Economic Literature* 47, no. 1 (2009): 5–86. <https://doi.org/10.1257/jel.47.1.5>.

Miteva, Daniela A., Randall A. Kramer, Zachary S. Brown, and Martin D. Smith. “Spatial Patterns of Market Participation and Resource Extraction: Fuelwood Collection in Northern Uganda.” *American Journal of Agricultural Economics* 99, no. 4 (2017): 1–19. <https://doi.org/10.1093/ajae/aax027>.

Pfaff, Alexander, and Juan Robalino. “Spillovers from Conservation Programs.” *Annual Review of Resource Economics* 9, no. 1 (October 5, 2017): 299–315. <https://doi.org/10.1146/annurev-resource-100516-053543>.

(Remarks on code availability)

Version 2:

Reviewer comments:

Reviewer #1

(Remarks to the Author)

I am satisfied with the responses to most of my remaining comments.

Regarding my previous statement on the extent of your contribution and your response, I believe that we simply have a disagreement of opinion. I think that most researchers believe that their work is “vital”, which is why they continue to do it. The fact that your previously results are “still not widely accepted” is not surprising, as research can take time to be integrated into policy and private sector spheres. However, a frustration that people aren't paying sufficient attention to what you have said isn't necessarily a strong justification for publishing results that are very similar to previous ones.

That said, the science of what you have done here is sound, and there is sufficient detail for replication to occur. Your conclusions follow logically from the analysis and so you have met key benchmarks for publishable research.

(Remarks on code availability)

Reviewer #2

(Remarks to the Author)

See file attached

(Remarks on code availability)

Reviewer #4

(Remarks to the Author)

While the article has improved, I still have non-negligible concerns regarding the writing, and data, specifically pertaining to the differences in the bespoke and remote sensing datasets. More details are below.

Writing

The contributions of this article are still not clearly stated in the Intro (lines 125-132). I think the text in the response documents was better.

It should be clarified in the text that the implicit underlying assumption is that markets (and property rights) are localized, so that the leakage would also be local (line 581, 595-597).

The manuscript claims that because the data by Vancutsem et al (2021) detected higher forest loss inside project areas, it must also detect higher rates in control areas (line 355-358). I am not sure if this is true: If the forests inside the control polygons are allocated differently (e.g., more scattered or different tree species), it may not be as easy to forest loss. Some more support is needed to address this claim.

How many projects had negative quasi-experimental estimates? (line 653).

Include in the text whether the forest loss differences vanished if temporary forest loss ("degraded category) and regrowth forest were included (line 695). This is statement is currently only in S4.

Insert in the text some potential explanation for why the Madagascar projects resulted in more plausible carbon credits (lines 335-340)

I would use "forest loss" and not deforestation (e.g., line 354) as it is not clear why forest was lost (some forest loss is not deforestation).

In my experience, the Discussion section focuses on interpreting the results. I am not used to seeing results (e.g., figures) referenced there (e.g., as in lines 340-370)

The main text should briefly mention what alternative methods are available (line 380)

Lines 381-384 contain significant claims that need to be substantiated with citations (e.g., is there actual evidence of "perverse incentives" and of "financial interests"? Also, whose—Verra's?

The credibility of quasi-experimental designs depends, first and foremost, on a rigorous theory of change that takes into consideration the local context. This point is acknowledged in lines 394-400. Ex post robustness checks and sensitivity analyses need to assess the credibility of quasi-experimental research designs. Nowadays there are also better methods than matching (e.g., the explosion of panel data estimators, some of which can be applied to small samples (see ref 26 in the text). These points should be clarified in the discussion in lines 385-412.

I am not sure how the 83% prediction of variance (line 391) supports robustness to unobserved confounders. Isn't a 17% margin of error large?

Given that potentially better statistical methods exist, why are they not used? (lines 442-444)

What "multiple model specifications" (line 504-505)?

I am not sure what "propensity score subclassification" (line 503) is. At the very least, a citation is needed.

Was any sensitivity analysis done for more than just the study by Guizar-Coutino et al. (2025) (line 397)? The text mentions the sensemakr framework (line 505) but no description appears in the results.

The horizontal bar with two stars on Fig. 3b is confusing.

Issues related to grammar and readability (not a comprehensive list):

- Lines 645-646 as worded are hard to understand.
- Lines 653-655 are likely missing a verb.
- Line 663 is also confusing: What deforestation rates?
- Active voice is better than passive voice (e.g., lines 715-720), but, at least, it did not obfuscate the meaning of sentences.
- The readability sentences like the ones on lines 346-349 and 276-279 can be improved (unnecessarily wordy)
- Awkward phrasing (line 355-357 and 461-462): For example, it is unclear what "this" refers to.
- Grammar issue in lines 474-476 ("however" is a conjunctive adverb that requires different punctuation)
- Some sentences contain unnecessary repetitions: For example, "Replace" is used twice in the same sentence (line 413-415)

Analysis

How was the bootstrapping done (lines 658, 162)? What is being resampled?

The first 4 bars in Fig. 5 are overlapping. The dark blue, the orange, and the red bars also overlap. As displayed, they suggest that the differences are not statistically significant. For this reason, I am not convinced by the decomposition of the differences in the quasi-experimental vs. ex ante models (lines 281-301). The horizontal bar with stars also seems out of place.

Data

What do the bespoke layers consider as forest loss? Do logging roads and log landing pads count? It is still not clear to me from the text to what extent the bespoke layers and the remote sensing datasets are consistent in how they quantify forest loss.

The ACC layer (Vancutsem et al. 2021) uses “degraded” as a label for forest that is temporarily lost (i.e., less than 2.5 years). By excluding this short-term forest loss, the analysis predictably underestimates the forest loss calculated from remote sensing data. However, S4 mentions that including the “degraded” forest decreased the rates of forest loss. This does not make sense to me: If only the numerator is increasing (e.g., #cells with forest loss)/total forest cells in a unit, the ratio should increase.

Further, the study focuses on the loss of “undisturbed” forest and does not allow for forest regrowth or the loss of forest that has been previously restored (lines 573-579, line 619, 672). It is not clear from the write-up if the exclusion of regrowth is another reason there are differences between the remote sensing and bespoke datasets (e.g., lines 350-354). S4 mentions that including regrowth lowers the deforestation rates. However, I am not clear if S4 also includes the loss of forest that has been previously regrown?

The manuscript argues that using a 2019 layer for the travel time to health centers is an appropriate proxy for economic pressure/“market exposure” (lines 567-571). I am still not convinced: First, whether the variable proxies for market access depends on the drivers of deforestation in a given location: Commercial ports may not have a health center; timber is often transported down a river, where there may not be health centers. Logging roads may or may not be accessible for motorized vehicles throughout the year. Second, the endogeneity concern is significant: It is possible that roads and health centers appeared after deforestation took place or after the projects were implemented. Land cover, one of the components in the calculation, is also endogenous. For this reason, it is not clear to me why the distances to ports and rivers (both readily available) are not used, at least, to supplement the list of covariates. At minimum, some sensitivity analysis is needed for the endogenous travel time covariate.

(Remarks on code availability)

Version 3:

Reviewer comments:

Reviewer #1

(Remarks to the Author)

I appreciate the effort that you made to reframe your results. My own opinion is that this dissection of sources of error is more interesting for a field journal than for a general interest outlet like Nature Communications. That said, assessments of fit are vague and subjective, and so I leave this up to the editors of this journal to decide if they feel that is the case.

(Remarks on code availability)

Reviewer #4

(Remarks to the Author)

The manuscript has improved substantially in terms of writing. Two remaining issues:

1. It is still not clear to what extent the bespoke and ACC layers are comparable (this is an issue I have been raising for a while now). The bespoke layers seem to account for the loss of regrown forests provided they had been in place for at least 10 years (lines 471-472). In contrast, the ACC layers used in the quasi-experimental designs exclude reforestation. Further, the ACC layers classify forest disturbance, regrowth, or loss based on the preceding 3 years of data (Vancutsem et al. 2021). They use a different methodology for the last three years of data. The ACCs layers do not have good data prior to 2000 for Africa and may not do a good job excluding tree plantations for some locations (see the description of the caveats in the Vancutsem et al. 2021 paper). The manuscripts reports that ACC data from 1990 to 2021 (last year of available data in ACC, I think)were used (lines 561-563).

To improve the paper, I think the loss of long-term reforestation (>10 years) should be included in the deforestation calculations to match the bespoke layers. The Vancutsem et al. paper reports estimates for long-term reforestation (10+ years) in their paper, so the data by the same authors exist. The data/methods section needs to insert more details about the time frames of the ACC data used as well as factors like tree plantations or projects in Africa could be a reason for the difference in the two data sources.

2. While the text has been substantially improved, I am still struggling to understand some paragraphs. For example, what is the difference between the over-crediting ratio (lines 663-668) and the global over-crediting ratio (lines 670-675)? Just the exclusion of the negative quasi-experimental estimates? Given that there are 17 projects with good enough data for comparisons, I am not following why “Projects were randomly sampled with replacement 10,000 times” (lines 673-675)

Minor points

- I would insert the original sample size in line 616
- Line 525 claims the annual avoided deforestation is just the cumulative estimate/#years whereas the methods describe compounding (lines 680-695)
- In many disciplines “mechanisms” connotes the process through which something happens. Here you are using “mechanism” instead of “explanation”. Consider revising the text.
- Spillovers are not “unpredictable” (line 416) but rather require modeling.

- You may want to add the need for transparency (e.g., in determining reference areas) in generating credits to lines 370-375.
- I am not sure what “pooling credits at the scheme-level” means (line 443)
- If possible, it will be interesting to insert a short description of how the reference areas are selected for ex-ante modeling: Is it just the exposure to the drivers of deforestation within a pre-defined cutoff (as reported in Table 1) ? Is there a reason to believe the ex ante modeling assumes a different mechanism of change than the ex post quasi-experimental analysis?
- Why are there two types of shapes (diamonds and circles) in Fig. 2b?
- Was the accessibility in the reference areas within 10-20% of the treated areas as suggested by Table 1? (lines 259-266)

References

Vancutsem, C., F. Achard, J.-F. Pekel, et al. 2021. “Long-Term (1990–2019) Monitoring of Forest Cover Changes in the Humid Tropics.” *Science Advances* 7 (10): eabe1603. <https://doi.org/10.1126/sciadv.abe1603>.

(Remarks on code availability)

I have not reviewed the code.

Version 4:

Reviewer comments:

Reviewer #4

(Remarks to the Author)

The manuscript has improved enough to warrant publication. However, there are still issues that would need to be addressed:

1. Citations are needed when you make statements based on previous work. For example, lines 545-552 are based on Vancutsem et al. 2021 (the authors of the ACC layer); the points summarized in these lines are featured in the original paper.
2. Fine to keep "mechanism" instead of "explanation", but add a sentence to define what you mean in the text.
3. Please proofread for statements like "In fact, the most accessible parts of reference areas were more accessible than the most accessible areas within projects (+/- 20%) 82% of the time." (lines 251-253)--apart from the needless repetitions, the sentence takes a lot of time to process.

(Remarks on code availability)

Response to reviewers

NCOMMS-24-45991-T: Understanding the reasons for over-crediting in REDD+

We would like to thank the reviewers for their extremely helpful and valued comments. Their suggestions helped us make substantial changes to our manuscript which we believe have greatly strengthened the analysis and improved our exposition of its necessity and relevance.

Specifically, we have:

- 1) Changed the title to better reflect the evidence we present.
- 2) Substantially rewritten the whole piece to more clearly draw out the novelty. Decisions and announcements at the recent COP29 climate conference make it clearer than ever that there are vitally important lessons to be learnt from the 1st generation of REDD+ methodologies as these have implications for new approaches being promoted in what is being called Voluntary Carbon Market 2.0.
- 3) Rewritten the introduction to explicitly draw out criticisms levelled at independent studies using quasi-experimental methods and explanations why certification methods produced less valid estimates of counterfactual deforestation used to assess REDD+ impacts. This is made clear through the addition of figure 1, which sets out the questions we ask.
- 4) Provided a description of the PACT method.
- 5) Reorganised the results section to clearly answer the questions posed in the introduction.
- 6) Reanalysed our data using improved statistical approaches and more consistency in terms of the projects included in each analysis. We now present:
 - a) Confidence intervals from standard errors instead of ranges (Fig. 2;
 - b) Assessments of differences in the deforestation detected by global forest cover layers and the bespoke layers used in remote sensing (Fig. 3 and S4). We calculate compounded annual deforestation rate as the amount of forest loss required each year to result in the forest loss at the end of the evaluation (Figs. 3-5) and test the sensitivities of these results to different forest cover definitions (S4 and S7)
 - c) Formal statistical tests for significant differences in observable confounders across 17 projects (Fig. 4)
 - d) An attempt to isolate the relative contributions of causes of over-crediting across the same 17 projects (Fig. 5)

- e) Moved what was previously figure 2, considering the selection of ex ante modelling approaches, to the supplementary information (S6).
- 7) Rewritten the discussion so that it clearly explains that inappropriate assessments permitted by flexible methods and perverse incentives led to over-crediting. We then explain that while these findings suggest the new jurisdictional REDD+ methods being promoted in Voluntary Carbon Market 2.0 are an improvement over 1st generation REDD+ methods, moving to ex post verification of credits using quasi-experimental approaches is needed to restore confidence in the carbon credits from avoided deforestation. We propose the relevance of these results beyond REDD+ to other nature-credit markets.

Below we lay out a detailed point-by-point summary of the changes we have made.

Reviewer #1 (Remarks to the Author):

This paper presents a compelling assessment of the level of additionality calculated in multiple REDD+ analyses by certification agencies compared to an assessment of the same projects using statistical methods. The purpose of this is to help assess the level of overcertification and point the way towards a more constructive process in the future. I appreciate the effort and approach, but I have a few comments that I think may help clarify the analysis and policy recommendations and also temper some of the conclusions.

Main comments:

1. Measurement error in deforestation data: I really like the general approach of trying to separate out matching quality from remote sensing error. However, I am concerned about two things related to the quality of the remote sensing data. First, the measurement error in MapBiomas and the GFC product is significant. A number of papers document serious underestimation of deforestation in GFC (Burilavova et al. 2015, Mitchard et al., 2015). I have not seen such studies for MapBiomas, but I do know that the estimations of change are sometimes revised downwards in subsequent versions of the series. This is not to criticize these products, but rather to note that it is quite possible that this error creates significant problems for statistical assessment of impacts. If the measurement error is classical, this may attenuate your point estimates. If it is non-classical, the direction of the bias is unpredictable.

Response: We agree with the reviewer that remote sensing error is a substantial unknown in our analysis and was previously only considered indirectly. We have therefore made a significant effort to improve our analysis to give a clearer insight into the effect of differences in remote sensing. We were not able to test the accuracy of remote sensing classifications in certified control areas because neither the numbers or the remote sensing layers are made available in the project documents. Similarly, we could not test the accuracy of the ACC because the project's ground truthing data are never made publicly available. However, we were able to assess the differences in deforestation detected by global forest cover layers and the bespoke layers used in remote sensing within the project areas, which should be the same. We find that the ACC measures at least as much as the bespoke layers, which is presented in the newly added (Fig. 3 & S4). We have now added the following statement to discussion:

"We explored the critique that independent studies only found over-crediting because the global forest cover layers they used measured less deforestation. This could mean that deforestation events were omitted from quasi-experimental control areas, leading to underestimated counterfactual and avoided deforestation¹⁸. Several studies have found that locally trained remote sensing products demonstrated greater accuracy in measuring forest cover and deforestation than global layers, such as the ACC, especially in some regions and contexts²⁸⁻³⁰. However, the ACC's overall accuracy for detecting disturbances was 91.4% (9.4% omissions and 8.1% false detections)⁵. We were unable to directly test for differences in deforestation measured in control areas because certified documents rarely report these data, but we were able to compare ACC-derived and certified rates within project areas.

Here, the ACC global layer detected similar or higher deforestation (Fig. 3 and S4). If this result was repeated in certified control areas, the ACC measurements would be equivalent to or exceed certified control deforestation. Therefore, we found little support for this critique and suggest that within some projects, certified remote sensing is a possible cause of over-crediting. However, as an isolated effect this explained only around 2% of total over-crediting (Fig. 5)."

We believe this makes clear there can be substantial errors with global forest cover datasets but that slightly higher *ex post* measurements in project areas suggests this is not the reason for over-crediting.

Further, when estimated rates of deforestation are low, this puts a ceiling on potential impacts. You might note this point somewhere in your description of your statistical assessments.

Response: We agree that if remote sensing approaches are less sensitive at detecting deforestation this will reduce the deforestation rates that can be detected and have accordingly added the following statement to the introduction:

"Criticism levied at the independent studies is based on ... the globally available, remotely sensed forest cover datasets they rely on detect less deforestation than bespoke layers¹⁷."

You provide little information on the "bespoke" remote sensing products used by certifiers. It is likely that these also have significant error, but if they are created at a lower resolution than 30m, they probably more accurately assess deforestation in low income, tropical settings where clearing takes place in small patches. You resolve part of this by noting that you find certified deforestation relative to your own assessment to be higher in reference areas than in program areas. However, it feels like there need to be error bars on this assertion. The error bars that you have accommodate the error in the data used for statistical analysis. It seems like there should also be an accommodation for the error in the certification data. If standard deviations for their estimations are available, then perhaps you could use these to adjust the error bars in your figure 1b, which has error in both the numerator and the denominator.

Response: We thank the reviewer for this comment, which caused us to restructure the manuscript to better pick apart the different causes of over-crediting. The new structure makes clear the different explanations for over-crediting. Certified assessments typically used classification of Landsat or other remote sensing imagery (that was not higher in spatial resolution or contained fewer spectral bands than Landsat) using an approach summarised in a new section of the supplementary information (S1). As above, we agree with the reviewer that it is possible that certified classifications were more accurate than the ACC. However, our comparison of *ex post* measurements of deforestation in project areas, which should be the same for both the ACC and the bespoke layers, demonstrates there was no significant difference and the ACC actually detected more deforestation. This is now presented clearly in the new figure 3. We also now make clear throughout the manuscript that remote sensing constitutes only one source of error in the certified estimates. Other errors come from the selection of the control areas and the choice of modelling approach. These effects are isolated in new figure 5. Unfortunately, certified estimates do not typically, if ever, present statistical errors that properly account for all sources of error. This precludes the possibility of us adjusting the error bars to what was figure 1b (now figure 2b).

2. Clarity on methodology:

a. It would be nice to have more clarity on the “statistical approaches” used in the paper. I am particularly interested in the covariates. The paper uses different “confounders” and approaches in every assessment. This makes it hard to figure out what is actually happening. To my mind, if you are trying to propose an alternative methodology, then you should propose something that is replicable and consistent. For example, you might want an approach that uses similar covariates across multiple projects. One can always, with refined data and endless iterations of increasingly complex statistical approaches, come up with much better estimates than are done with more coarse, globally consistent data. Further, hindsight, as they say, is 20/20. Ex post estimates will always show something different than ex ante predictions. In addition, given what you have demonstrated about the cherry picking of reference areas towards those that have higher levels of deforestation, one wouldn't want to leave a huge number of degrees of freedom in the way that reference areas are selected. It seems unreasonable to suggest that certifying institutions should be conducting analysis with the same number of robustness checks that academic researchers do. We need to provide guidance on what might be both effective and feasible.

Response: We agree with the reviewer that our manuscript previously did not properly explain the independent studies. We have now addressed this in several ways. Firstly we refer to ‘quasi-experimental methods’ used by independent studies, which we think is a more specific term than ‘statistical approaches’. Secondly, we have added a new section to the methods to explain ‘The PACT method’, which summarises all the variables it used to control for known confounders across all projects. Thirdly we have added a section to the supplementary information (S1) which summarises certified and quasi-experimental approaches.

We have improved our communication of the primary aim of the paper: not to propose a single methodology but to directly compare the certified estimates of avoided deforestation with quasi-experimental methods to understand the reasons for over-crediting. We find that very different quasi-experimental methods draw broadly consistent conclusions about the scale of over-crediting, and that remote sensing, control area selection and ex ante modelling all contributed to over-crediting.

We have made this clear in the Introduction:

“Unfortunately, over-crediting by REDD+ projects is still not widely accepted. As recently as 2024, a high-profile report and study by industry-insiders dismissed concerns around over-crediting from the first generation of certified REDD+ methods^{15,16}. Criticism levied at the independent studies is based on two key arguments: (1) their results lack consistency due to variations in model specification¹⁷, and (2) the globally available, remotely sensed forest cover datasets they rely on detect less deforestation than bespoke layers¹⁸. These arguments require investigation, however reluctance to accept over-crediting has prevented a consolidated effort to identify its causes and develop the safeguards vital to protecting the integrity of recovering credit markets¹⁹.”

In other work we have argued for the use of ex post approaches to issue credits over ex ante approaches because they take into account all pertinent information from similar but

untreated locations, up to the point of credit issuance (Swinfield et al. 2024). However we recognize that currently the VCM and methodologies being developed for Article 6 compliance, all use *ex ante* approaches. We believe that PACT, which uses a single set of consistent and well considered covariates for estimating counterfactual rates of deforestation, is completely transparent and is available as open-source software could be widely used. We have made this clear in the discussion:

“Unfortunately, the movement to JREDD+ does not entirely remove the risk of perverse incentives leading to over-crediting. The methods for JREDD+ assume that the drivers of deforestation remain constant between the historic period and the project period. This assumption will often not hold as forest conservation or trade policies, major infrastructure projects³⁸ or extreme climatic events³⁹ will all vary over time and can impact deforestation. In fact, ex post measures of deforestation in untreated jurisdictions have been shown to differ by as much as 100% from ex ante predictions based on historic averages^{33,40}. If projects were to enrol in JREDD+ schemes randomly without regard for the likely change in future deforestation trends, this may not result in systematic bias. However, evidence tells us that proponents will opt into crediting schemes when it is to their advantage^{41,42}, with projects being preferentially developed in jurisdictions with declining deforestation pressures. A potential solution is to move to only issuing credits ex post based on assessments which integrate new developments in quasi-experimental methods. Ideally, different quasi-experimental results should be considered, from methods such as the Permanent Additional Carbon Tonne (PACT) method used in our study⁴³ or advances in synthetic controls that require fewer assumptions²⁶. Although, quasi-experimental methods are still susceptible to unobservable, time-varying confounders, the scale of these effects can be tested and reported³¹. Therefore, until there is clear consensus on how best to measure impacts, a precautionary approach should be taken, with conservative estimates of avoided deforestation produced to safeguard the claims being made¹⁹.”

b. It would be useful to describe in the appendix the ways in which VERRA is estimating their counterfactual areas. Creating a deforestation risk map based upon a number of covariates and then selecting areas of similar baseline risk is quite similar to matching. Your analysis of covariate similarity suggests that they may not be doing a great job on this, but it is hard to tell because we don't really know what they are doing.

Response: To aid understanding we have added more detail on the selection of control areas under the different certification methods in Table 1 and for the interested reader cited West, Bomfim and Haya (2024), where this topic is also covered. We have also added a section to the supplementary information (S1) which summarises certified and quasi-experimental approaches:

“The first generation of REDD+ projects could choose from several certification methods (Table 1) to measure historic deforestation during a reference period in expert-selected control areas (or ‘reference areas’) and model ex ante (or ‘baseline’) predictions of counterfactual deforestation¹⁰ (summarised in S1).”

Further, I have the impression from Table 1 that the VERRA estimates of deforestation are based upon predictions rather than on observed deforestation. Is that correct? In this case, it

seems like a major source of the problem in the assessments could be that the prediction is not very good, not that the area of comparison is poorly selected. Perhaps you can clarify what is actually happening here.

Response: We agree with the reviewer that prediction is a better word than forecast and have changed this throughout the text. We also agree that prediction error is a major cause of over-crediting. This is now demonstrated in new figure 5, which shows that between 41% and 78% of over-crediting can be attributed to either ex ante modelling or remote sensing of deforestation in control areas. We have made this point in the discussion:

“Predicting an ex ante deforestation counterfactual scenario is difficult. However, flexibility in the modelling approaches used to estimate this counterfactual likely played a major role in over-crediting. Isolating this effect, though likewise difficult due to insufficient reporting, suggests it contributed to between 41% and 78% of over-crediting (Fig. 5). This range of possible contributions resulted from different considerations in assessing the effect of control area selection and remote sensing of deforestation (Fig. 5). West, Bomfim and Haya¹⁰ showed that projects routinely selected methods resulting in high estimates of avoided deforestation from among the pool of possible certification methods. Further, they found that the certification methods selected produced estimates that were higher than quasi-experimental findings. We extended this assessment and contrasted their possible certified estimates against distributions of multiple quasi-experimental results. For most projects analysed, the certification methods chosen predicted avoided deforestation levels that constituted significant over-crediting, despite credible alternative methods being available (S6). Our finding supports the argument that the ability to choose ex ante modelling approaches within the first generation of REDD+ projects was a critical vulnerability, allowing parties with perverse incentives to exploit this flexibility to favour their financial interests³² and ultimately over-credit projects.”

3. Clarity of policy recommendations:

I think that a considerable amount of work could be done to make the policy recommendations of this paper more clear and helpful. Below are some suggestions.

a. Some constellation of the authors in this paper have already established that there is an issue with over certification on forest carbon markets. It would be useful to understand what the first result of this paper adds there that is new.

Response: As the reviewer says, a series of papers indicated over-crediting but there has been substantial push-back arguing that quasi-experimental methods are unreliable because they generate too varied estimates and use remote sensing layers that detect too little deforestation. For the first time, this paper examines these claims in detail by applying several different quasi-experimental methods to a large sample of projects and finds that:

1. While quasi-experimental estimates do indeed vary, they do so far less than certified estimates
2. The global remote sensing layers used by quasi-experimental methods do not detect less deforestation than bespoke layers used in certification

3. There is evidence that certified control areas were exposed to greater threat of deforestation than projects
4. There is also evidence to suggest flexibility in *ex ante* modelling caused elevated claims of avoided deforestation
5. These issues together resulted in over-crediting, underscoring the fundamental unsuitability of current certification methods compared with quasi-experimental methods.

We have made this much clearer in our revision through the introduction of figure 1, which summarises these questions. The manuscript is then structured so that the questions are posed one-by-one and answered in the results.

b. The main constructive recommendation of the paper is that *ex post* statistical approaches be used in carbon crediting. As I was thinking about this, I got confused about what is actually being done versus what is proposed. Are the authors suggesting withholding payments until assessment of avoided deforestation can occur *ex post*? This seems like an impossible proposition, both at the broader institutional level and at the level of producer participation in the event that projects involve individual land users foregoing deforestation, which they must do for there to be additionality. Recent research suggests that successfully implementing PES in low-income settings may require *ex ante* payments to actually induce wide-scale participation (Izquierdo-Tort et al. 2024). It therefore seems more reasonable to suggest a way to consistently apply matching or the construction of synthetic controls *ex ante* using data that is generally available and covariates that are known to be correlated with both selection into treatment and deforestation outcomes. I think that is what is being suggested here. Counterfactual deforestation could then be predicted for reference areas, and then once time has passed actual deforestation rates would be used in assessment. I think what I would like to see is clarification of the policy proposition.

Response: We appreciate the reviewer's confusion here. Current practice is that the *ex ante* counterfactual scenario is produced before the project is implemented but credits are issued once an *ex post* assessment of project stock has been completed. The problem that we are raising is that the *ex post* assessment is limited to the project deforestation, while the counterfactual is not revised to accommodate the most up-to-date information. The reviewer is of course right that these projects have to raise investment in order to finance the project, prior to the sale of credits, but we have argued in Swinfield *et al.* (2024) that the credits should not be issued without also revising the counterfactual in the *ex post* assessment. We now make this clear at several points.

In the introduction:

"These ex ante counterfactual estimates were compared with measurements of deforestation in project areas to determine a certified metric of avoided deforestation with which to issue credits."

In the discussion:

"A potential solution is to move to only issuing credits ex post based on assessments which integrate new developments in quasi-experimental methods."

AND

“For example, nature-based restoration projects typically assume that all increases in carbon or biodiversity above a pre-project level are additional⁴⁴. However, this is a strong assumption which should be validated by comparing project stocks with the average stocks in control areas selected using quasi-experimental methods⁴⁵.”

c. Two implementation issues should be recognized in the discussion.

i. The first of these is that perfect prediction of the location of future deforestation is impossible. Deforestation is a rare event statistically, and so targeting programs such that only future deforested land is enrolled does not make sense, regardless of how much data you have with which to make the prediction. Even trying to approach this kind of targeting in a REDD+ program would require taking money away from actual incentives and funneling it into bureaucratic processes intended to support targeting, which would be wasteful. This means that you have to pay for enrollment of larger areas of land than are likely to be deforested.

Response: We agree with the reviewer that predicting the exact location of deforestation is very difficult. There are inevitable trade-offs between costs of targeting and additionality. For that reason we think that moves towards jurisdictional programmes will be positive but the inclusion of quasi-experimental approaches would be a further improvement. We now say this in the discussion:

“Under JREDD+ methods such as VM0048 and ART TREES, counterfactual deforestation rates are set as the mean annual deforestation rate over a recent historic period (usually 5-6 years)¹². Our results suggest that this more straightforward and inflexible approach to estimating a counterfactual could significantly reduce over-crediting risks (Fig. 5). Further JREDD+ is designed to enable integration into Article 6 of the Paris agreement allowing jurisdictions to raise finance through the trade of International Transfers of Mitigation Outcomes³⁷. Under these methodologies, independent data providers, with no direct financial stake in the number of credits issued produce the assessment of deforestation over the historic period. This is a positive step which should reduce the perverse incentives which resulted in over-crediting in earlier project-level REDD+ methodologies.

Unfortunately, the movement to JREDD+ does not entirely remove the risk of perverse incentives leading to over-crediting. The methods for JREDD+ assume that the drivers of deforestation remain constant between the historic period and the project period. This assumption will often not hold as forest conservation or trade policies, major infrastructure projects³⁸ or extreme climatic events³⁹ will all vary over time and can impact deforestation. In fact, ex post measures of deforestation in untreated jurisdictions have been shown to differ by as much as 100% from ex ante predictions based on historic averages^{33,40}. If projects were to enrol in JREDD+ schemes randomly without regard for the likely change in future deforestation trends, this may not result in systematic bias. However, evidence tells us that proponents will opt into crediting schemes when it is to their advantage^{41,42}, with projects being preferentially developed in jurisdictions with declining deforestation pressures. A potential solution is to move to only issuing credits ex post based on assessments which integrate new developments in quasi-experimental methods. Ideally, different

quasi-experimental results should be considered, from methods such as the Permanent Additional Carbon Tonne (PACT) method used in our study⁴³ or advances in synthetic controls that require fewer assumptions²⁶. Although, quasi-experimental methods are still susceptible to unobservable, time-varying confounders, the scale of these effects can be tested and reported³¹. Therefore, until there is clear consensus on how best to measure impacts, a precautionary approach should be taken, with conservative estimates of avoided deforestation produced to safeguard the claims being made¹⁹.”

ii. This does not in itself undermine the principle of this paper. However, it is worth considering that the global community often says that avoided deforestation and, more recently, forest restoration, is “cheap.” The perceived cheapness of the intervention may be at least partly driven by the fact that when we calculate REDD+ transfers per hectare using current estimates of deforestation, they are very low. However, we might find that if we use current per hectare rates spread over a much lower, precisely estimated amount of avoided deforestation, we may not be able to induce any participation in these types of programs, because they will not sufficiently compensate producers for their opportunity cost. Environmental non-profit organizations and rich country governments often engage in wishful thinking regarding the true costs of nature-based solutions. It is important that the research community does not fall prey to this same mistake.

Response: We agree with the reviewer that there is a widespread belief that REDD+ credits are the cheap option. We have now made clear in the discussion that our finding reveals that conserving nature is likely to cost more than expected:

“Indeed, despite inflated supply obscuring their real value, nature-based credits remain a critical tool in the fight against climate collapse and mass biodiversity loss.”

Smaller issues:

I believe that the acronym JRC TMF is not defined anywhere.

Response: We thank the review for identifying the use of the acronym JRC TMF when we should have used ACC as defined in the introduction. All instances have now been changed.

Works cited:

Burivalova, Zuzana, Martin R. Bauert, Sonja Hassold, Nandinanjakana T. Fatroandrianjafinonjasolomiovazo, and Lian Pin Koh. 2015. Relevance of global forest change data set to local conservation: Case study of forest degradation in Masoala National Park, Madagascar. *Biotropica* 47 (2): 267–74.

Izquierdo-Tort, S., Jayachandran, S. and Saavedra, S., 2024. Redesigning payments for ecosystem services to increase cost-effectiveness (No. 21022). https://seemajayachandran.com/pes_redesign_mexico.pdf

Mitchard, Edward, Karin Viergever, Veronique Morel, and Richard Tipper. 2015. Assessment of the accuracy of University of Maryland (Hansen et al.) Forest Loss Data in 2 ICF project areas: Component of a project that tested an ICF indicator methodology. Technical report, University of Edinburgh.

Reviewer #2 (Remarks to the Author):

Summary

Thank you for the opportunity to read this paper. I very much enjoyed reading it.

This work investigates the Carbon Credit certification methods for avoiding deforestation from REDD+ (Reducing Emissions from Deforestation and Degradation) projects. Carbon credits are an essential pillar of the agenda to reduce the impacts of deforestation on carbon emissions and climate change. The author's work highlights the flaws of current methodologies in creating reliable estimates of avoided deforestation from REDD+ projects. Crediting institutions mostly estimated high levels of avoided deforestation. In contrast, using reliable assessment methods, the paper shows that REDD+ projects had much lower levels of avoided deforestation. As a result, REDD+ projects have been the base for issuing overly high carbon credits. The authors highlight that over-crediting stems (a) from a failure to create credible counterfactual scenarios and (b) from systematically choosing methods that generate higher levels of avoided deforestation. This research is especially important as upcoming reforms aim to expand REDD+ projects and the monetary compensation to jurisdictional entities. Maintaining the credibility of the certification system will be essential to secure the future funding of these market-based forest conservation instruments.

I appreciate the analysis of the overestimation of avoided deforestation in REDD+ projects. This is an excellent depiction of the potential flaws in estimating overly high levels of avoided deforestation. My primary concerns are about the novelty of the results, the author's empirical reasoning, and the use of the empirical terminology. If the authors pursue these improvements and others they receive elsewhere, I believe the paper will make a very valuable contribution to the literature.

Major comments:

1. Novelty: The database for this paper is collected from different sources. The authors combine data from the crediting institution (i.e., VERRA) and from previous publications evaluating REDD+ projects using quasi-experimental methods published by West et al. (2020), West et al. (2024), Guizar-Coutiño et al. (2022), Guizar-Coutiño et al. (2024), and Balmford et al. (2023) (henceforth WGGB). The latter, Balmford et al. (2023), is an unpublished document describing the PACT method. Based on the manuscript, it is very difficult to understand if PACT or Balmford et al. (2023) are a source of data or if the authors are referring to a methodology. Is the paper conducting new impact evaluations or just using the results from Balmford et al. (2023) or even from (cf. Swinfield et al., 2024) – a recent publication of the lead authors? If so, what is the genuine contribution of the paper? Has the paper “only” pooled estimates and re-used the information to display the results in a more comprehensive way?¹.

¹ Figure 1 is based on all 47 REDD+ projects as it collects data from VERRA, West et al. (2020), West et al. (2024), Guizar-Coutiño et al. (2022), Guizar-Coutiño et al. (2024), and Balmford et al. (2023) (henceforth WWGGB). Figure 2 is limited to 4 projects because West et al. (2024) as data on it. Figure 3 is restricted to 16 projects where boundaries of VERRA's

The authors should highlight their essential contribution better. It seems the essence lies in the two messages i) There is over-crediting, ii) Over-crediting can stem from a) selecting dissimilar control, b) mismeasuring deforestation in control areas, and c) choosing evaluation methods that rely on strong assumptions (see comment XXX). To avoid confusion, I suggest sticking with the PACT method. First, show Figure 3, then Figure 1, and finally, Figure 4. Figure 2 should be an Appendix table. It lacks power and is merely based on 4 REDD+ projects analyzed by West et al. (2024).²

Response: We thank the reviewer for their comment and agree that the novel contribution was unclear in the previous version of the manuscript. We have made a substantial effort in our revised version to ensure this is properly demonstrated. A summary is provided in our response to Reviewer 1 above, under item (3). Where we make clear we have incorporated nearly all of the reviewer 2's suggestions.

We agree it is necessary to properly present details of the PACT method in this paper as it is unpublished and we rely so heavily on its results. For this reason these details are now presented in the methods under a new section entitled 'The PACT method'. The focus on the PACT method required reordering the results, which we have done in line with the reviewers' recommendations:

1. There is consistent over-crediting (Fig. 2)
2. Over crediting is not caused by the remote sensing layers used to measure deforestation (Fig. 3)
3. Certified control areas were exposed to greater threat of deforestation (Fig. 4)
4. Evaluation methods with strong and unsubstantiated assumptions were chosen (Fig. 5)

We have also moved old figure 2 to supplementary information (S6). The errors associated with model selection are dealt with in our new figure 5 where we attempt to measure the effects of different explanations for over-crediting .

2. Empirical terminology: The manuscript is filled with confusing terminology. I suggest carefully reviewing the manuscript using an econometric lens. Examples are

- I. line 61: "The key problem in quantifying additionality is to identify the counterfactual outcome". It is impossible to "identify" the counterfactual. The counterfactual is never observable. One can only estimate the counterfactual. By estimating the counterfactual, one can identify the causal effect and "infer" causality.

Response: We agree with the reviewer that the language we used was inappropriate. We have rewritten the paper and now emphasise throughout that counterfactuals are "estimated" and we are explicit about the assumptions on which the key quasi-experimental methods for estimating avoided deforestation rest. For example, in the introduction we now state:

"A major cause of over-crediting stems from problems associated with estimating counterfactual outcomes (what would have happened to deforestation in the absence of projects)⁸. The first generation of REDD+ projects could choose from several certification

control/ reference area are available but then again uses PACT/Balmford et al. (2023) to assess the projects.

² Figure 2 shows min/max bounds instead of confidence intervals of each estimate. I wonder if VERRA's estimates lie within the confidence intervals of the quasi-experimental estimates.

methods (Table 1) to measure historic deforestation during a reference period in expert-selected control areas (or ‘reference areas’) and model ex ante (or ‘baseline’) predictions of counterfactual deforestation¹⁰ (summarised in S1). These ex ante counterfactual estimates were compared with measurements of deforestation in project areas to determine a certified metric of avoided deforestation with which to issue credits. All methods require that control areas were similar to projects in terms of key characteristics but the lack of a formal approach to account for confounders (factors which influence both which areas are selected to be REDD+ projects, and the deforestation rate)^{20,21} risks biasing control area selection towards places exposed to greater deforestation than the project. The flexibility afforded in ex ante modelling could also introduce bias if models were selected for their tendency to predict higher estimates of counterfactual deforestation¹⁰.

Independent studies used quasi-experimental methods which explicitly and transparently account for known confounders to produce more robust estimates of counterfactual deforestation^{6–8}. Quasi-experimental methods explicitly account for confounders in the selection of control areas and measure deforestation for both control and project areas in parallel during the project period ex post. One common design is to match units (pixels or polygons) exposed to the REDD+ project with control units which are as similar as possible in exposure to known confounders before the project began^{7,22,23}. Assuming there are no unobserved or time-varying confounders, observing ex post outcomes in the control area serves as a good estimate of deforestation in the counterfactual scenario²¹. The Synthetic Control Method is an advance which can replicate pre-project trends through the selection and weighting of untreated units and, at least in theory, account for time varying, unobserved confounders^{8,24–26}.”

- II. Similarly, all figures use the label “counterfactual” whereas it should be labeled control group, control area, comparison group, etc.

Response: We agree and now refer to control areas used to estimate the counterfactual throughout. With this in mind figures 4 and 5 now refer to ‘Certified control’ and ‘Quasi-experimental control’.

- III. The methods used by WWGGB are labeled “statistical methods”. This is a wrongful description because VERRA also uses a “statistical method”. It is just that WWGGB use quasi-experimental methods.
- IV.

Response: We agree and have changed the terminology to quasi-experimental methods throughout.

3. Empirical reasoning:

- I. In general, the paper misses a careful discussion about empirical concepts and methods. I agree that WWGGB’s and the presented quasi-experimental methods (QEM) are superior, but the authors’ advocacy is flawed. The authors argue that QEMs presented are better because they create a “better” comparison group. This judgement is based on improved covariate balance after matching. Nonetheless, the empirical assumption of QEMs (they vary in detail) is that unrealized counterfactual outcomes can be projected by observable pre-treatment characteristics (and

outcomes). Unobservable characteristics might still exist that drive the selection of both treatment and outcome simultaneously. This would then lead to an estimation bias (direction unclear) of any of the quasi-experimental methods used in WWGGB. VERRA's methods, in essence, rely on similar assumptions. That is why VERRA could always argue that they have 'context knowledge' - a characteristic that is not controlled for in WWGGB. Therefore, the essence of the critique of VERRA cannot hinge on a "better" covariate balance among observable characteristics³. The argument must be made on the details of the underlying assumptions of VERRA's methods and the quasi-experimental methods. Clearly, VERRA uses very strong assumptions: no leakage, no unobserved time-varying confounders (linear projection), no cross-sectional time-constant confounders, population as the main driver of deforestation, etc. In comparison, matching and panel regression use less stringed empirical assumptions: Selection on observable, no unit-and time-varying confounders, i.e., conditional parallel trends, etc.

Response: We agree with this comment strongly and have reworked the text to be more explicit about the assumptions required for the quasi-experimental and certification methods. This includes the new paragraph in the introduction explaining the assumptions inherent to the quasi-experimental methods to estimate counterfactual deforestation (see above) and explicit acknowledgement in the discussion that none of the quasi experimental methods include testing for the sensitivity of results to hidden confounders:

"One important caveat is that projects may have used expert knowledge of projects to take account of local drivers of deforestation in selecting controls (Table 1). These local drivers would not be accounted for by quasi-experimental methods, and could act as hidden confounders in the analysis³¹. However, it seems unlikely that hidden confounders alone would explain the consistency and scale of over-crediting across the majority of projects."

And providing a better justification for the use of quasi-experimental methods in jurisdictional REDD+ in the discussion:

"Unfortunately, the movement to JREDD+ does not entirely remove the risk of perverse incentives leading to over-crediting. The methods for JREDD+ assume that the drivers of deforestation remain constant between the historic period and the project period. This assumption will often not hold as forest conservation or trade policies, major infrastructure projects³⁸ or extreme climatic events³⁹ will all vary over time and can impact deforestation. In fact, ex post measures of deforestation in untreated jurisdictions have been shown to differ by as much as 100% from ex ante predictions based on historic averages^{33,40}. If projects were to enrol in JREDD+ schemes randomly without regard for the likely change in future deforestation trends, this may not result in systematic bias. However, evidence tells us that proponents will opt into crediting schemes when it is to their advantage^{41,42}, with projects being preferentially developed in jurisdictions with declining deforestation pressures. A potential solution is to move to only issuing credits ex post based on assessments which integrate new developments in quasi-experimental methods. Ideally, different

³ It is easy to counter-argue that the list of covariates used in PACT is incomplete and does not reflect the regional socio-economic differences and deforestation pressures in, e.g., Peru (Figure 3).

quasi-experimental results should be considered, from methods such as the Permanent Additional Carbon Tonne (PACT) method used in our study⁴³ or advances in synthetic controls that require fewer assumptions²⁶. Although, quasi-experimental methods are still susceptible to unobservable, time-varying confounders, the scale of these effects can be tested and reported³¹. Therefore, until there is clear consensus on how best to measure impacts, a precautionary approach should be taken, with conservative estimates of avoided deforestation produced to safeguard the claims being made¹⁹.”

- II. Furthermore, I suggest adding a few sentences on the rapidly developing quasi experimental methods literature. WWGGB’s methods could be deemed outdated. For example, synthetic control methods are evolving, and new estimators rely on fewer assumptions while producing better covariate balances (cf. Hazlett and Xu, 2018). Nonetheless, I don’t think the authors should implement any of the newer estimators.

Response: We agree that a brief discussion of the evolving field of quasi-experimental would be valuable. In the discussion, we refer to these advances.

“A potential solution is to move to only issuing credits ex post based on assessments which integrate new developments in quasi-experimental methods. Ideally, different quasi-experimental results should be considered, from methods such as the Permanent Additional Carbon Tonne (PACT) method used in our study⁴³ or advances in synthetic controls that require fewer assumptions²⁶. Although, quasi-experimental methods are still susceptible to unobservable, time-varying confounders, the scale of these effects can be tested and reported³¹. Therefore, until there is clear consensus on how best to measure impacts, a precautionary approach should be taken, with conservative estimates of avoided deforestation produced to safeguard the claims being made¹⁹.”

Minor comments

1. 1. Figure 1b shows there ratio (Avoided deforestation credited by VERRA / Avoided deforestation using quasi-experimental methods). For overestimation, the ratio is < 1, and for underestimation, the ratio is > 1. I suggest flipping the definition of the ratio so that values > 1 reflect overestimation.

Response: We agree with the reviewer that this is confusing however because quasi-experimental methods tend to estimate values close to, at or even below zero, whereas certifying methods are all positive, the proposed approach would make it impossible to present all the projects as they would have very large or infinite value. Logging the data is also problematic due to the negative values. Therefore we have opted to retain the same presentation. However, we now presented over-crediting as suggested in the results:

“The mean additionality ratio (calculated as the mean quasi-experimental estimate of avoided deforestation / certified estimate, averaged across the 44 projects claiming to have avoided deforestation) of 0.27 (Fig. 2b) was significantly > 0 (one-tailed Mann-Whitney U

against a group of 0s, $U = 1628$, $p < 0.001$). This suggests that on average projects were additional but had over-credited by a factor of 3.8.”

2. Instead of using min/max values for the lines in Figure 1, I suggest using 90%/95% confidence intervals. For points based on multiple estimates, you can calculate the joint standard error and plot the joint confidence interval. This way, it would be easier to assess if the quasi-experimental methods significantly differ from VERRA's estimates. Alternatively, I would plot all the estimates separately instead of making a joint Plot of all estimates of WWGGB or only focus on estimates from PACT to remain consistent with Figures 3 and 4.

Response: We have now presented 1.96 times standard errors instead of ranges in figure 1.

3. Again, I suggest excluding the whole section on the “difference in variation” of the estimates (cf. Figure 2). On reason: Variance depends on the number of point estimates. WWGGB has more point estimates than VERRA, so the variance of this group tends to be lower. Further, the depiction of the probability in line 196 is a bit far-fetched⁴. The whole thing lacks power – only 4 projects.

Response: We agree with the reviewer that the previous presentation of the analysis was inappropriate due to the comparison of the variance of the certification and quasi-experimental results. As a result of this feedback, we now use figure 5 to demonstrate the effect of modelling choice on over-crediting. The results from what was figure 2 now constitutes S6 in the supplementary information, being referenced as such in the discussion:

“Predicting an ex ante deforestation counterfactual scenario is difficult. However, flexibility in the modelling approaches used to estimate this counterfactual likely played a major role in over-crediting. Isolating this effect, though likewise difficult due to insufficient reporting, suggests it contributed to between 41% and 78% of over-crediting (Fig. 5). This range of possible contributions resulted from different considerations in assessing the effect of control area selection and remote sensing of deforestation (Fig. 5). West, Bomfim and Haya¹⁰ showed that projects routinely selected methods resulting in high estimates of avoided deforestation from among the pool of possible certification methods. Further, they found that the certification methods selected produced estimates that were higher than quasi-experimental findings. We extended this assessment and contrasted their possible certified estimates against distributions of multiple quasi-experimental results. For most projects analysed, the certification methods chosen predicted avoided deforestation levels that constituted significant over-crediting, despite credible alternative methods being available (S6). Our finding supports the argument that the ability to choose ex ante modelling approaches within the first generation of REDD+ projects was a critical vulnerability, allowing parties with perverse incentives to exploit this flexibility to favour their financial interests³² and ultimately over-credit projects.”

⁴ Also, 0.009 should be compared to 0.5 (the probability of a random selection).

4. line 53: I don't understand the point in "Second, the carbon ..."

Response: This text has now been removed.

5. line 547: The link to download the data does not work.

Response: The repository was mistakenly left private at the time of submission. This has now been changed to public.

References

- Balmford, Andrew, David Coomes, James Hartup et al. 2023. "PACT Tropical Moist Forest Accreditation Methodology." 10.33774/coe-2023-g584d-v2.
- Guizar-Coutiño, Alejandro, David Coomes, Tom Swinfield, and Julia P G Jones. 2024. "Sensitivity of estimates of the effectiveness of REDD+ projects to matching specifications and moving from pixels to polygons as the unit of analysis." bioRxiv. 10.1101/2024.05.22.595326.
- Guizar-Coutiño, Alejandro, Julia P. G. Jones, Andrew Balmford, Rachel Carmenta, and David A. Coomes. 2022. "A global evaluation of the effectiveness of voluntary REDD+ projects at reducing deforestation and degradation in the moist tropics." *Conservation Biology* 36 (6): . 10.1111/cobi.13970.
- Hazlett, Chad, and Yiqing Xu. 2018. "Trajectory Balancing: A General Reweighting Approach to Causal Inference With Time-Series Cross-Sectional Data." *SSRN Electronic Journal*. 10.2139/ssrn.3214231.
- Swinfield, Tom, Siddarth Shrikanth, Joseph W. Bull, Anil Madhavapeddy, and Sophus O. S. E. zu Ermgassen. 2024. "Nature-based credit markets at a crossroads." *Nature Sustainability*. 10.1038/s41893-024-01403-w.
- West, Thales A. P., Jan Börner, Erin O. Sills, and Andreas Kontoleon. 2020. "Overstated carbon emission reductions from voluntary REDD+ projects in the Brazilian Amazon." *Proceedings of the National Academy of Sciences* 117 (39): 24188–24194. 10.1073/pnas.2004334117.
- West, Thales A.P., Barbara Bomfim, and Barbara K. Haya. 2024. "Methodological issues with deforestation baselines compromise the integrity of carbon offsets from REDD+." *Global Environmental Change* 87 102863. 10.1016/j.gloenvcha.2024.102863.

Reviewer #3 (Remarks to the Author)

Reviewer #3 (Remarks on code availability):

The link for the data availability did not work.

Response: The repository was mistakenly left private at the time of submission. This has now been changed to public.

Reviewer #4 (Remarks to the Author):

The manuscript entitled “Independent statistical approaches address overcrediting problems in REDD+” compares the avoided deforestation rates from ex ante assessments using the methods applied by a major carbon crediting agency with the ex post estimates of statistical approaches for 47 REDD+ projects in Latin America and Africa. The manuscript argues that the ex ante methodology tends to overstate the avoided deforestation by the REDD+ projects due to the selection of often incomparable reference groups.

While the topic of revising REDD+ and quantifying additionality from conservation and climate change mitigation interventions is very timely and important, significant concerns remain about the manuscript regarding the statistical methods employed.

Statistical and model considerations

1. Without perusing the prior studies on which the manuscript is based, it is not clear what the ex post statistical methods employed are: For example, the manuscript mentions synthetic controls and one-to-one matching without replacement (Table 1), but does not provide key details. For example, why not synthetic difference-in-difference models that combine features of synthetic and heterogeneity-robust difference-in-difference estimators¹? These are applicable to small as well as large samples, and can dominate synthetic controls¹. Matching without replacement is used (lines 400-410) but there is no information if the resulting standard errors are clustered at the matched pair level (as they should)². It is also unclear if matching is combined with difference-in-difference techniques when the sample size permits. Further, one of the studies, from which ex post statistical estimates for the current manuscript are used, is still unpublished (line 402); without evaluating the statistical analysis, it is unclear if the estimates therein are credible. Further: Do the studies employing pixels as the unit of analysis (Table 1) use sampling approach that results in the same probability of selection for each pixel? If not, how are the different sampling probabilities accounted for in the analysis? Since the point of the manuscript is to highlight the advantages of statistical methods, details about the statistical methods are crucial, especially for studies not peer-reviewed/published yet.

Response: We thank the reviewer for pointing out the absence of important details pertaining to the quasi-experimental methods. We agree that more details are necessary to understand our analysis and interpret our results. For this reason we have provided a full description of ‘The PACT method’ in the Methods section. Hopefully it is now clear that pixel-pairs are used to constitute control areas, from which counterfactual outcomes are estimated. We have not presented standard errors from any one quasi-experimental study, instead we assess the standard errors across quasi-experimental studies. Credibility of counterfactual outcomes is assumed from the similarity in terms of observed known confounders and parallel trends in pre-project deforestation.

2. Are the ex ante and ex post estimates statistically different? Fig. 2 and 3 suggest that there may be substantial overlap between the point estimates and the confidence intervals. That is, are the differences driven by a few outliers (e.g., Fig. 2)?

Response: We thank the reviewer for asking this question. It encouraged us to move old figure 2 to the supplementary information (S6) and to conduct appropriate statistical tests on old figure 3 (now figure 4) and elsewhere throughout the manuscript. As stated in our response to reviewer 2 (minor comment 3). The important finding is that:

“Predicting an ex ante deforestation counterfactual scenario is difficult. However, flexibility in the modelling approaches used to estimate this counterfactual likely played a major role in over-crediting. Isolating this effect, though likewise difficult due to insufficient reporting, suggests it contributed to between 41% and 78% of over-crediting (Fig. 5). This range of possible contributions resulted from different considerations in assessing the effect of control area selection and remote sensing of deforestation (Fig. 5). West, Bomfim and Haya¹⁰ showed that projects routinely selected methods resulting in high estimates of avoided deforestation from among the pool of possible certification methods. Further, they found that the certification methods selected produced estimates that were higher than quasi-experimental findings. We extended this assessment and contrasted their possible certified estimates against distributions of multiple quasi-experimental results. For most projects analysed, the certification methods chosen predicted avoided deforestation levels that constituted significant over-crediting, despite credible alternative methods being available (S6). Our finding supports the argument that the ability to choose ex ante modelling approaches within the first generation of REDD+ projects was a critical vulnerability, allowing parties with perverse incentives to exploit this flexibility to favour their financial interests³² and ultimately over-credit projects.”

In old figure 3 (now figure 4) we previously had not conducted significance tests of the differences in the standardised mean differences of the observable known confounders. In the revised version, we use t-tests to show that certified control areas were significantly different from project areas in terms of several known confounders, including inaccessibility, and forest cover over the preceding 10 years. This is not driven by outliers.

“Compared with project areas, mean inaccessibility in certified control areas was lower (median std. mean diff. = -0.30, $t = -2.41$, $d.f. = 16$, $p = 0.0283$), as was forest cover at the start of the project (median std. mean diff = -0.42, $t = -3.10$, $d.f. = 16$, $p = 0.0068$) as well as five and ten years beforehand (median std. mean diff = -0.48, $t = -4.04$, $d.f. = 16$, $p = 0.0029$ and median std. mean diff = -0.48, $t = -4.28$, $d.f. = 5$, $p = 0.0078$, respectively) prior. Historic deforestation rates were not significantly different. In contrast none of the control areas used in the quasi-experimental method differed from the project areas in any of the observed confounders (Fig. 4c).”

2. The annual average rates of avoided deforestation are not the total deforestation/#years (line 417-419). Instead, some compounding is necessary. Consider switching to effect sizes instead^{3,4}

Response: We have updated the calculation of the deforestation rates in new figures 3-5 so that they are compounded annual deforestation rate in terms of the amount of forest loss required each year to result in the forest loss at the end of the evaluation period. This is described in a new section entitled 'Determining comparable deforestation rates between Certified and ACC sources' in the Methods:

"We used annual deforestation rates to compare the certified and ACC deforestation rates, including those produced by PACT. The future forest cover F_t resulting from a constant annual deforestation rate r after an interval t , given the starting forest cover F_0 is calculated as follows:

$$F_t = F_0(1 - r)^t \quad \text{Eq. 2.1}$$

As we had extracted the forest cover at the beginning and end of the evaluation period, we calculated the rate by rearranging the formula to:

$$r = 1 - \left(\frac{F_t}{F_0}\right)^{\frac{1}{t}} \quad \text{Eq. 2.2}$$

We determined F_0 as the proportion of the ACC pixels that were in the undisturbed class for the yearly layer closest to the project start date and multiplied this by the area of the project, measured from the project polygon. F_t was calculated by subtracting the area of deforestation in the certified or ACC measurements from F_0 . The evaluation interval t was the number of years between the start and end assessments. For the certified assessments this was calculated using the number of total days between the start and end dates, whereas for the ACC this was the number of whole years between the layers used. The calculations for each project are presented in section S10 of the supplementary information."

3. What is the formula for the standardized mean difference used in this study (line 497-499)? There are two versions in the literature (for example, see 5,6).

Response: We thank the reviewer for pointing this out we now specify the formula used in the Methods as follows:

$$SMD = \frac{\mu_c - \mu_p}{\bar{\sigma}} \quad \text{Eq. 1}$$

Where μ_c is the mean of the control set, μ_t is the mean of the project set and $\bar{\sigma}$ is the square root of the mean of the variances."

4. The principal components analysis is not traditionally used to assess degree of covariate overlap (e.g., lines 502-510); instead, studies report the degree of overlap for each covariate used in the analysis. Why was the principal components analysis used? Can you reference prior studies that advocate for this method over comparisons of the distributions of each covariate? The approach should be backed up with citations.

Response: We agree with the reviewer that principle components analysis (PCA) is not typically the standard approach for assessing covariate balance. As such we have removed this analysis from the manuscript.

5. What are the “likely confounders for the project” (line 500). The only covariates listed are those in Fig. 3d—elevation, slope, inaccessibility, baseline and lagged versions of forest cover and deforestation. Were any other covariates (e.g., ones that capture local economic processes used?) For example, the proximity to rivers is of huge importance in some locations as this is how households transport timber illegally (e.g., 7,8)—was that captured by the inaccessibility variable? No other (e.g., socio-economic) variables seem to have been included. Have the original studies assessed the sensitivity of the results to unobserved confounders especially when no difference-in-difference models are used?

Response: The covariates listed are the only covariates used in the PACT method, which is now made clear in the Methods under ‘*The PACT Method*’:

“First, a set of geospatial raster layers were compiled to track changes in forest cover and encode observable characteristics known to drive deforestation. The Annual Change Collection for Tropical Moist Forests (ACC) produced by the Joint Research Council of the European Commission was the dataset used to track the changes in forest cover. We used the 2021 version, which is a 30 m resolution (0.09 ha) time series from December 1990 to December 2021, classifying pixels as one of six classes: undisturbed, degraded, deforested, restored, water and other. To ensure control units were similar in terms of local policies and regulations, economic pressures, historical changes in land cover, and environmental conditions, we used the following layers: international country borders produced by OpenStreetMap⁵⁰, Ecoregions provided by Resolve⁵¹, elevation provided by NASA’s Shuttle Radar Topography Mission⁵², and motorised accessibility to healthcare in 2019 provided by the Malaria Atlas Project⁵³. From the elevation layer we calculated slope using GDALDEM (v3.8.4) with default settings. We also computed the proportional cover of undisturbed and deforested classes from each year of the ACC time-series (1990-2021). Proportional cover was calculated by counting the number of pixels in each class and dividing by the total number of pixels within the 1 km radius neighbourhood around each pixel. Finally, to minimise interference of effects from the treatment, we did not allow control units to be selected from within the vicinity of any REDD+ project. To do this we produced a binary raster indicating the presence or absence of a REDD+ project within 5km. REDD+ polygons were accessed from the VERRA registry and supplemented by any REDD+ project polygons shared directly with the Cambridge Centre for Carbon Credits. All layers were reprojected to the same coordinate reference system as the ACC.”

Slope, historic deforestation rate and accessibility represent a mixture of proxies for local economic processes that have been shown to be important predictors of deforestation (Vieilledent et al. 2023). We used the inaccessibility layer from Weiss et al. (2020) which shows the quickest path across all forms of transport to healthcare from any location. It is measured using travel times from open street map, including roads and rivers, as well as walking speeds in different vegetation types and topographies. Distance to healthcare is likely a better proxy of distance to nearby settlements of a certain size than is distance to nearest road. Of course there is always a trade off between a generalised approach which can be used across multiple sites, and one which makes the best possible use of site-based information, We have updated the text to be more explicit about the assumption of causal

inference from such a study design that there are no unobserved confounders, in the introduction:

“One common design is to match units (pixels or polygons) exposed to the REDD+ project with control units which are as similar as possible in exposure to known confounders before the project began^{7,22,23}. Assuming there are no unobserved or time-varying confounders, observing ex post outcomes in the control area serves as a good estimate of deforestation in the counterfactual scenario²¹. The Synthetic Control Method is an advance which can replicate pre-project trends through the selection and weighting of untreated units and, at least in theory, account for time varying, unobserved confounders^{8,24–26}.”

And in the discussion:

“One important caveat is that projects may have used expert knowledge of projects to take account of local drivers of deforestation in selecting controls (Table 1). These local drivers would not be accounted for by quasi-experimental methods, and could act as hidden confounders in the analysis³¹. However, it seems unlikely that hidden confounders alone would explain the consistency and scale of over-crediting across the majority of projects.”

Guizar-Coutiño et al. (2024) has assessed the sensitivities of the additionality estimates to different covariates but these remain at a pre-publication stage and are thus not presented in our manuscript .

6. What is the “forest scarcity factor” (line 134). Is it possible it is driving a wedge between the ex ante and ex post estimates?

Response: The forest scarcity factor is an exponent in the mathematical formula used to estimate counterfactual deforestation term in VM0006. The purpose of the term is to reduce estimated deforestation rates when forest cover approaches zero. West, Bomfim and Haya (2024) find it has relatively little effect on the estimated counterfactual deforestation rates.

7. What are the reference sites used in the ex post statistical analysis? For example, the manuscript mentions leakage as a concern (line 53) but it is not clear how that leakage is accounted for in any of the estimations without a proper economic model of change.

Response: The control areas produced by PACT are described in ‘The PACT method’ section. We accept that leakage could increase the apparent signal of the treatment effect via more deforestation of untreated units. In this way the ex post estimates of counterfactual deforestation would be higher than appropriate and meaning that over-crediting would be underestimated. We do not think this is likely to be a substantial problem but have added the following text to the Methods:

“The domain of untreated units was defined as all pixels within the same country and ecoregions as the project, at least 5 km from the project, and no further than 2,000 km away. Leakage effects caused by the displacement of production could be present⁴⁶ but were assumed to be small, given the large domain of the untreated units and the relatively small effects of projects.”

9. The calculations of the overcrediting (lines 150-155) are very hard to follow. Please elaborate how you obtained the %.

Response: We appreciate that the original text was hard to follow. Over-crediting is now first defined in the introduction:

“... evidence from several independent studies⁶⁻¹⁰ suggests that the first generation of REDD+ projects overestimated their impact, meaning that certified claims far exceeded the additional progress achieved. This over-crediting contributed to a sharp decline in market confidence with US\$1Bn of value wiped from the voluntary carbon market in 2023¹¹.”

With this definition in mind, in the Results we now specify that over-crediting occurs when quasi-experimental estimates are lower than certified estimates. To avoid confusion from only referring to additionality ratios we also now talk about the factor by which over-crediting occurred, which we hope is easier to understand:

“While there was a correlation between mean estimates of avoided deforestation from quasi-experimental and certified methods ($r = 0.43$, $df = 45$, $p = 0.0025$), certified estimates were consistently higher (one-tailed Wilcoxon signed-rank, $V = 950$, $p < 0.001$) suggesting systematic over-crediting is a reality (Fig. 2 and labelled by project in S3). The mean additionality ratio (calculated as the mean quasi-experimental estimate of avoided deforestation / certified estimate, averaged across the 44 projects claiming to have avoided deforestation) of 0.27 (Fig. 2b) was significantly > 0 (one-tailed Mann-Whitney U against a group of 0s, $U = 1628$, $p < 0.001$). This suggests that on average projects were additional but had over-credited by a factor of 3.8.”

10. It is unclear what the test on lines 465-471 is trying to achieve. Is there a prior study that proposes or uses this method? Why is this method better than alternatives (e.g., something that imposes an underlying distribution)?

Response: We agree with the reviewer that the original text did not make it clear enough what was being assessed. We have now employed a new statistical analysis as summarised in response to this reviewer’s comment 2.

11. It seems that most of the projects assessed are from a few countries (Table S1). Is it possible to look at heterogeneity between countries? That is, it is possible the overestimation of avoided deforestation is country-specific due to country-specific factors.

Response: We thank the reviewer for this suggestion. We have added a new section in the supplementary information (S8) which shows figure 2 coloured by country. We find no evidence of differences between countries except for Madagascar. In the discussion we now say:

“...quasi-experimental methods consistently find lower deforestation than was certified, suggesting substantial and consistent over-crediting in the voluntary carbon market. This effect was repeated across countries, except in Madagascar where all three projects we assessed were found to have protected more forest than claimed (S8)”

Data

1. Table 1 and lines 514-534 summarize the data sources used to quantify deforestation. It appears that the ex post and ex ante analyses use different data sources: The methodology

employed by Verra uses the Bespoke data, whereas the ex post statistical analyses use publicly available remote sensing products. What remains unclear is to what extent are these datasets comparable? For example, what is the resolution of the data? Do they handle missing values in the same way? What are the error/uncertainty estimates for each—for example, the TMS dataset spans all moist forests; the Hansen data (ref 40)-the whole globe. For this reason, it is likely the two datasets have different errors than a dataset that is produced and groundtruthed locally:

<https://www.globalforestwatch.org/blog/data-and-tools/how-accurate-is-accurate-enough-examining-the-glad-global-tree-cover-change-data-part-2/>.

Do the different datasets classify deforestation in the same way? The manuscript alludes that the ex post statistical studies may not be consistent with each other. First, the Hansen and TMS datasets use different methods and classifications. The TMS dataset defines forest degradation as the temporary (less than 2.5 years) loss of forest cover⁹—is this a consistent definition employed in all of the land cover datasets used to assess avoided deforestation? The Hansen data used in the manuscript (line 129, reference 40) does not define deforestation, but rather “forest cover loss” events. The Hansen data does not define what “forest” is—prior studies suggest employing a 25% tree cover cutoff to define “forests”¹⁰ but the cut-off may be too low for the Amazon but ok for parts of Africa. Was any pre-processing of the Hansen data done in any of the statistical analyses? It also does not allow for easy comparisons before and after 2015 because of different remote sensing classification methodologies used:

<https://www.globalforestwatch.org/blog/data-and-tools/tree-cover-loss-satellite-data-trend-analysis/>. Finally, some of the statistical estimates include the switch for undisturbed forest to degradation as “deforestation” (line 433, lines 425-435).

Is a consistent definition of “deforestation” used in both the ex ante and ex post analyses? For example, is the creation of roads considered deforestation? For example, clearing forests for logging roads in logging concessions may not be considered as “deforestation” in the policy world. Forest cover loss associated with traditional slash-and-burn agriculture is also not considered “deforestation” in the policy world. Both temporary clearing and road building will be picked up by the Hansen data as “forest loss events” as well as any event that results in a forest cell having less than 50% tree cover in a given year.

Details on the data used are crucial and should be inserted in the manuscript and the appendix. The credibility of the main claim of the manuscript needs to be beefed up by comparing the data used for ex ante analyses with the data used for ex post analyses.

Response: We thank the reviewer for the comment and agree that the definition of deforestation is important and likely has consequences for the analysis. The independent studies all used different remote sensing layers to measure deforestation. However, if the studies used the same forest cover layers it is likely we would find even more similar results than we have presented in. To understand the differences between the bespoke layers and different ways of using the ACC, we now include new analyses to assess the effect of differences in the quasi-experimental results caused by remote sensing (Fig. 3, S4, Fig. 5).

First we consider the ex post deforestation observed in project areas by the bespoke layers used in certified assessments and ACC used by PACT (Fig. 3). Here we find that:

Ex post deforestation rates in project areas, measured independently using a global forest cover layer (ACC⁵), were higher than certified measurements produced using bespoke forest cover layers. For the subset of 36 projects that provided certified measurements from project areas, the median certified rate was 0.17%/year compared with 0.26%/year measured as the loss of undisturbed forest by the ACC (Fig. 3a). These discrepancies resulted in a median pairwise difference of 0.13%/year (Fig. 3b) that was significantly different from zero (Wilcoxon paired signed-ranks, $V = 520$, $d.f. = 35$, $p = 0.003$)”.

This could be because we considered transitions to the degraded class as deforestation. As such, we explored the effect of instead excluding the degraded class from measures of deforestation. In this way deforestation was only measured by transition to the deforested class. When we did this, median annual deforestation rates dropped from 0.26%/year to 0.09%/year and were now lower than the certified rate of 0.16%/year (S4). Clearly, definitions of deforestation matter. Degradation events (as measured by the ACC) were nearly twice as common in REDD+ projects as deforestation events. As the reviewer states, the ACC defines degradation as a non-forest classification of pixels over a period of 2.5 years or less. This has been shown to result in a median biomass loss of more than 50% within 12 months, assessed using space-borne lidar (Holcomb et al. 2024). For this reason, we believe REDD+ impacts must be measured including short duration disturbances in both project and control areas. Our finding, that certified assessments measured less deforestation in project areas than the ACC, suggests REDD+ projects have under-reported emissions within project areas (Fig. 5).

Turning to differences in deforestation in control areas, we now state in the discussion:

“We were unable to directly test for differences in deforestation measured in control areas because certified documents rarely report these data, but we were able to compare ACC-derived and certified rates within project areas. Here, the ACC global layer detected similar or higher deforestation (Fig. 3 and S4). If this result was repeated in certified control areas, the ACC measurements would be equivalent to or exceed certified control deforestation. Therefore, we found little support for this critique and suggest that within some projects, certified remote sensing is a possible cause of over-crediting. However, as an isolated effect this explained only around 2% of total over-crediting (Fig. 5).”

We also utilise the results from figure 3 to construct a correction coefficient to approximate a conversion between ACC and certified bespoke measurements of deforestation, assuming that the difference in project areas would be repeated in certified control areas. This is used in figure 5. For both ACC measurements and this new approximated bespoke metric, control area deforestation rates are lower than modelled counterfactual deforestation rates reported by projects (Fig. 5). This meant that the residual difference was not due to detecting less deforestation was likely explained by *ex ante* modelling.

Writing:

1. Given the heavy reliance on other, mostly peer-reviewed and published studies, that also question the credibility of REDD+ avoided deforestation estimates, it is not immediately clear what the contributions of this manuscript are. Is it a novel comparison with the methods

employed by Verra? Is it a meta-analysis? Further, the manuscript mentions that VERRA has introduced new consolidated methods to replace earlier ones (Line 96). Do these new methods still rely on ex-ante forecasts or include ex-post methods like those used in the statistical approaches? Incorporating new approaches could suggest that VERRA acknowledges the weaknesses in their earlier methodology and is working on improving their assessment tools.

Further, the manuscript mentions that VERRA has introduced new consolidated methods to replace earlier ones (Line 96). Do these new methods still rely on ex-ante forecasts or include ex-post methods like those used in the statistical approaches?

Response: We agree with the reviewer that we had not made the novelty of our study clear enough. We have made substantial efforts to make clear the purpose of our study and its novel contributions in a nearly complete rewrite. First we have set out the continuing reluctance to accept over-crediting in the introduction:

“Unfortunately, over-crediting by REDD+ projects is still not widely accepted. As recently as 2024, a high-profile report and study by industry-insiders dismissed concerns around over-crediting from the first generation of certified REDD+ methods^{15,16}. Criticism levied at the independent studies is based on two key arguments: (1) their results lack consistency due to variations in model specification¹⁷, and (2) the globally available, remotely sensed forest cover datasets they rely on detect less deforestation than bespoke layers¹⁸. These arguments require investigation, but reluctance to accept over-crediting has prevented a consolidated effort to identify its causes and develop the safeguards vital to protecting the integrity of recovering credit markets¹⁹.”

Then we introduce new figure 1, which sets out the questions we pose and answer in the manuscript. This is summarised with a new final paragraph in the introduction:

“In this study (Fig. 1) we provide the most comprehensive assessment of REDD+ project performance to date by bringing together certified assessments of 46 projects (48% of the projects producing credits by 2020)²⁷ and comparing these with results from five independent quasi-experimental studies (Table 2). We start by tackling the critiques of independent quasi-experimental studies and then explore two potential mechanisms by which over-crediting could have occurred. For the critique claiming inconsistent evidence of over-crediting, we ask how consistently five independent quasi-experimental methods indicate that the deforestation avoided by projects is substantially less than certified estimates (Q1). For the critique claiming global forest layers are insufficient, we assess whether estimates of deforestation rates in project areas differed between the remotely sensed forest classifications used in certified and quasi-experimental methods (Q2). To understand the mechanisms that could explain over-crediting, for projects with sufficient data (listed in S2), we ask whether control areas used in certified and quasi-experimental assessments differ from project areas in their exposure to observable confounders associated with deforestation, and if there are differences in deforestation rates between the certified and quasi-experimental control areas (Q3). Finally, to understand the effect of improbably high prediction of deforestation from ex ante modelling, we ask how much

over-crediting remained after isolating the effects of the remotely sensed forest classification used and control area selection (Q4).”

In the discussion, we put our findings in the context of the evolving systems for producing JREDD+ credits and nature-credits more broadly:

“There will be those who argue that further investigation of the impact of REDD+ projects certified with 1st generation methodologies is redundant because the Voluntary Carbon Market is moving towards jurisdictional approaches (JREDD+)”³⁶. Under JREDD+ methods such as VM0048 and ART TREES, counterfactual deforestation rates are set as the mean annual deforestation rate over a recent historic period (usually 5-6 years)¹². Our results suggest that this more straightforward and inflexible approach to estimating a counterfactual could significantly reduce over-crediting risks (Fig. 5). Further JREDD+ is designed to enable integration into Article 6 of the Paris agreement allowing jurisdictions to raise finance through the trade of International Transfers of Mitigation Outcomes³⁷. Under these methodologies, independent data providers, with no direct financial stake in the number of credits issued produce the assessment of deforestation over the historic period. This is a positive step which should reduce the perverse incentives which resulted in over-crediting in earlier project-level REDD+ methodologies.”

“Unfortunately, the movement to JREDD+ does not entirely remove the risk of perverse incentives leading to over-crediting. The methods for JREDD+ assume that the drivers of deforestation remain constant between the historic period and the project period. This assumption will often not hold as forest conservation or trade policies, major infrastructure projects³⁸ or extreme climatic events³⁹ will all vary over time and can impact deforestation. In fact, ex post measures of deforestation in untreated jurisdictions have been shown to differ by as much as 100% from ex ante predictions based on historic averages^{33,40}. If projects were to enrol in JREDD+ schemes randomly without regard for the likely change in future deforestation trends, this may not result in systematic bias. However, evidence tells us that proponents will opt into crediting schemes when it is to their advantage^{41,42}, with projects being preferentially developed in jurisdictions with declining deforestation pressures. A potential solution is to move to only issuing credits ex post based on assessments which integrate new developments in quasi-experimental methods. Ideally, different quasi-experimental results should be considered, from methods such as the Permanent Additional Carbon Tonne (PACT) method used in our study⁴³ or advances in synthetic controls that require fewer assumptions²⁶. Although, quasi-experimental methods are still susceptible to unobservable, time-varying confounders, the scale of these effects can be tested and reported³¹. Therefore, until there is clear consensus on how best to measure impacts, a precautionary approach should be taken, with conservative estimates of avoided deforestation produced to safeguard the claims being made¹⁹.”

“The problems arising from the combination of methodological flexibility and perverse incentives extend beyond REDD+ carbon credits to other nature-based schemes for generating carbon and biodiversity credits⁹. For example, nature-based restoration projects typically assume that all increases in carbon or biodiversity above a pre-project level are additional⁴⁴. However, this is a strong assumption which should be validated by comparing project stocks with the average stocks in control areas selected using quasi-experimental methods⁴⁵. Leakage and impermanence are also rarely considered significant issues in

certified assessments, possibly because they are less well understood than additionality, but their impacts are likely to be underestimated, in part for similar reasons (around flexibility and incentives) to those associated with overestimation of the additionality of REDD+ projects⁴⁶⁻⁴⁸.

2. It is not clear why the additionality ratios can be smaller than 0 (lines 141-149)? Out of the 47 REDD+ projects, did any certified estimates indicate increased deforestation in the project areas? Since the additionality ratio is the statistical estimate divided by the certified estimate (444-446), a negative ratio would imply that either one of the estimates had the opposite sign. If so, why and where was that the case?

Response: We agree with the reviewer that this was unclear and have now added the following sentence to the results to clarify:

“Negative additionality ratios occur when the quasi-experimental assessments find more deforestation in a project than its counterfactual.”

3. What is a “binary process” (line 434)?

Response: We agree that the original text wasn’t clear and have now provided the new text:

“The ACC dataset reports degradation and deforestation separately and the three analyses using it which we include here assessed deforestation differently. The Guizar-Coutiño studies considered deforestation as the transition from undisturbed or degraded classes to the deforested class. PACT considered degradation of the undisturbed class, alongside deforestation, because this has been shown to result in biomass reduction of >50% within 12 months period⁵².”

4. While the Introduction is generally well-written and easy to follow, the rest of the text has frequent grammar and punctuation errors. There are also very long sentences with interrupting parentheses with additional text (e.g., lines 104-107).

Response: We have taken on board the reviewer’s comment and have completely rewritten most of the manuscript, shortening long sentences and simplifying the text throughout in order to make our central message clearer.

5. The first 3 paragraphs of the Discussion section summarize the main findings of the manuscript. It appears unnecessary: please consider shortening them and avoiding redundancies.

Response: We agree with the reviewer and have restructured the discussion appropriately so that rather than summarising the results, the first part of the discussion is now used to unpick what the results mean for how we interpret the reasons why over-crediting happened.

Work cited:

1. Arkhangelsky, D., Athey, S., Hirshberg, D. A., Imbens, G. W. & Wager, S. Synthetic Difference-in-Differences. *Am. Econ. Rev.* 111, 4088–4118 (2021).
2. Abadie, A. & Spiess, J. Robust Post-Matching Inference. *J. Am. Stat. Assoc.* 117,

983–995 (2022).

3. Börner, J. et al. Emerging Evidence on the Effectiveness of Tropical Forest Conservation. *PLOS ONE* 11, e0159152 (2016).

4. Samii, C., Lisiecki, M., Kulkarni, P., Paler, L. & Chavis, L. Effects of decentralized forest management (DFM) on deforestation and poverty in low- and middle-income countries: a systematic review. *Campbell Syst. Rev.* 10, 1–88 (2014).

5. Imbens, G. W. & Wooldridge, J. M. Recent developments in the econometrics of program evaluation. *J. Econ. Lit.* 47, 5–86 (2009).

6. Abadie, A. & Imbens, G. W. Bias-Corrected Matching Estimators for Average Treatment Effects. *J. Bus. Econ. Stat.* 29, 1–11 (2011).

7. Ferretti-Gallon, K. & Busch, J. What Drives Deforestation and What Stops It? A Meta-Analysis of Spatially Explicit Econometric Studies. 44–44 (2014).

8. Miteva, D. A., Loucks, C. & Pattanayak, S. K. Social and Environmental Impacts of Forest Management Certification in Indonesia. *PLoS ONE* 10, e0129675–e0129675 (2015).

9. Vancutsem, C. et al. Long-term (1990–2019) monitoring of forest cover changes in the humid tropics. *Sci. Adv.* 7, eabe1603 (2021).

10. Sexton, J. O. et al. Conservation policy and the measurement of forests. *Nat. Clim. Change* 6, 1–6 (2015).

Reviewer #4 (Remarks on code availability):

I was not aware there was code to review when I downloaded the files to review.

The review pertains to the intuition, theoretical justification, and interpretation of the analyses included in the main text and the appendix to the manuscript.

Reviewer #5 (Remarks to the Author):

** See Nature Portfolio's author and referees' website at www.nature.com/authors for information about policies, services and author benefits.

This email has been sent through the Springer Nature Tracking System NY-610A-NPG&MTS

Confidentiality Statement:

This e-mail is confidential and subject to copyright. Any unauthorised use or disclosure of its contents is prohibited. If you have received this email in error please notify our Manuscript Tracking System Helpdesk team at <http://platformsupport.nature.com>.

Details of the confidentiality and pre-publicity policy may be found here <http://www.nature.com/authors/policies/confidentiality.html>

Privacy Policy | Update Profile

Bibliography

- Borcard, Daniel, Francois Gillet, and Pierre Legendre. 2011. *Numerical Ecology with R*. PDF. 2011th ed. Use R! New York, NY: Springer.
- Guizar-Coutiño, Alejandro, David Coomes, Tom Swinfield, and Julia P. G. Jones. 2024. "Sensitivity of Estimates of the Effectiveness of REDD+ Projects to Matching Specifications and Moving from Pixels to Polygons as the Unit of Analysis." *bioRxiv*. <https://doi.org/10.1101/2024.05.22.595326>.
- Swinfield, Tom, Siddarth Shrikanth, Joseph W. Bull, Anil Madhavapeddy, and Sophus O. S. E. zu Ermgassen. 2024. "Nature-Based Credit Markets at a Crossroads." *Nature Sustainability* 7 (10): 1217–20.
- Vieilledent, Ghislain, Christelle Vancutsem, Clément Bourgoïn, Pierre Ploton, Philippe Verley, and Frédéric Achard. 2023. "Spatial Scenario of Tropical Deforestation and Carbon Emissions for the 21st Century." *bioRxiv*. <https://doi.org/10.1101/2022.03.22.485306>.
- Weiss, D., A. Nelson, C. Vargas-Ruiz, Kristina Gligoric, S. Bavadekar, E. Gabrilovich, A. Bertozzi-Villa, et al. 2020. "Global Maps of Travel Time to Healthcare Facilities." *News@nature.com* 26 (September):1835–38.
- Zhang, Hengtao, Guosheng Yin, and Donald B. Rubin. 2024. "PCA Rerandomization." *Revue Canadienne de Statistique [The Canadian Journal of Statistics]* 52 (1): 5–25.

Response to reviewers

NCOMMS-24-45991-T: Understanding the mechanisms for over-crediting in REDD+

Reviewer 1 (Remarks to the Author):	2
Comment 1	2
Comment 2	3
Comment 3	4
Comment 4	6
Reviewer 2 (Remarks to the Author):	7
Comment 1	7
Comment 2	7
Comment 3	12
Comment 4	12
Comment 5	12
Comment 6	12
Comment 7	13
Reviewer #3 (Remarks to the Author):	13
Reviewer #4 (Remarks to the Author):	13
Comment 1	13
Comment 2	14
Comment 3	14
Comment 4	14
Comment 5	15
Comment 6	16
Comment 7	16
Comment 8	17
Comment 9	17
Comment 10	17
Comment 11	18
Comment 12	18
Comment 13	19
Comment 14	19
Comment 15	20
Comment 16	21
Comment 17	21
Comment 18	22
Comment 19	22
Comment 20	22
Comment 21	22
Comment 22	23

Comment 23	24
Comment 24	25
Comment 25	26

We would like to thank the reviewers for their extremely helpful and valued comments. Their suggestions helped us make substantial changes to our manuscript which we believe have greatly strengthened the analysis and improved our exposition of its necessity and relevance.

We have made the following edits to the manuscript:

Reviewer 1 (Remarks to the Author):

Thank you for your thorough responses to my comments. I feel quite satisfied with the main methodological responses and I think the paper has been improved by restructuring. I am not quite at ease with the changes made in response to my comments on policy framing. I will present these one by one. My new comments are *italicized*.

Comment 1

Original Comment:

Some constellation of the authors in this paper have already established that there is an issue with over certification on forest carbon markets. **It would be useful to understand what the first result of this paper adds there that is new.**

Original Response:

As the reviewer says, a series of papers indicated over-crediting but there has been substantial push-back arguing that quasi-experimental methods are unreliable because they generate too varied estimates and use remote sensing layers that detect too little deforestation. For the first time, this paper examines these claims in detail by applying several different quasi-experimental methods to a large sample of projects and finds that:

1. While quasi-experimental estimates do indeed vary, they do so far less than certified estimates
2. The global remote sensing layers used by quasi-experimental methods do not detect less deforestation than bespoke layers used in certification
3. There is evidence that certified control areas were exposed to greater threat of deforestation than projects
4. There is also evidence to suggest flexibility in *ex ante* modelling caused elevated claims of avoided deforestation
5. These issues together resulted in over-crediting, underscoring the fundamental unsuitability of current certification methods compared with quasi-experimental methods.

We have made this much clearer in our revision through the introduction of figure 1, which summarises these questions. The manuscript is then structured so that the questions are posed one-by-one and answered in the results.

New Comment:

In some sense, many of these points appear to be refinements of work that you have already published. It seems to me that a dissection of these mechanisms is a more field-relevant focus, rather than an analysis that brings us information that should be widely distributed to the readership of Nature Communications.

New Response:

We disagree with this perspective. We believe our work is vital for two reasons. Firstly, the fact that there was substantial over-crediting in REDD+ credits is still not widely accepted by the carbon crediting industry. This was clear throughout London Climate Action Week (June 2025) in panel presentations and by the questions co-authors of this paper got when on panels. It is difficult for the necessary reform of the voluntary carbon market to happen until this is accepted. Our paper goes way beyond existing published papers in the level of synthesis offered to confirm, beyond doubt, that over-crediting is a real problem. However, even more significantly our paper lifts the lid to explore why over-crediting occurred and what can be done to prevent its recurrence. In case we were not clear enough about why our study continues to be important despite the studies already published. We have now added the following to the start of paragraph 2:

“Without recognition of the scale of over-crediting, and an understanding of the underlying mechanisms, there is a risk that the same mistakes will be repeated in new generations of credits, seriously undermining the impact of nature-finance.” (lines 57 - 59)

Comment 2

Original comment:

The main constructive recommendation of the paper is that ex post statistical approaches be used in carbon crediting. As I was thinking about this, I got confused about what is actually being done versus what is proposed. Are the authors suggesting withholding payments until assessment of avoided deforestation can occur ex post? This seems like an impossible proposition, both at the broader institutional level and at the level of producer participation in the event that projects involve individual land users foregoing deforestation, which they must do for there to be additionality. Recent research suggests that successfully implementing PES in low-income settings may require ex ante payments to actually induce wide-scale participation (Izquierdo-Tort et al. 2024). It therefore seems more reasonable to suggest a way to consistently apply matching or the construction of synthetic controls ex ante using data that is generally available and covariates that are known to be correlated with both selection into treatment and deforestation outcomes. I think that is what is being suggested here. Counterfactual deforestation could then be predicted for reference areas, and then once time has passed actual deforestation rates would be used in assessment. I think what I would like to see is clarification of the policy proposition.

Original response:

We appreciate the reviewer’s confusion here. Current practice is that the *ex ante* counterfactual scenario is produced before the project is implemented but credits are issued once an *ex post* assessment of project stock has been completed. The problem that we are raising is that the ex post assessment is limited to the project deforestation, while the

counterfactual is not revised to accommodate the most up-to-date information. The reviewer is of course right that these projects have to raise investment in order to finance the project, prior to the sale of credits, but we have argued in Swinfield *et al.* (2024) that the credits should not be issued without also revising the counterfactual in the *ex post* assessment. We now make this clear at several points.

In the introduction:

These ex ante counterfactual estimates were compared with measurements of deforestation in project areas to determine a certified metric of avoided deforestation with which to issue credits.

In the discussion:

A potential solution is to move to only issuing credits ex post based on assessments which integrate new developments in quasi-experimental methods.

AND

For example, nature-based restoration projects typically assume that all increases in carbon or biodiversity above a pre-project level are additional⁴⁴. However, this is a strong assumption which should be validated by comparing project stocks with the average stocks in control areas selected using quasi-experimental methods⁴⁵.

New comment:

I think that you might add a sentence after your first discussion sentence that says something like, "This does not change the timing of credit issuance, but rather mandates that the counterfactual be taken into account."

New response:

We thank the reviewers for this suggestion, and have added the following sentence in the discussion:

Quasi-experimental crediting would not alter the timing of credit issuance, as even current ex ante counterfactual methods rely on ex post project measurements for quantifying additionality, but it does introduce uncertainty in credit yields via the unpredictable nature of the counterfactual. (lines 437 - 438)

Comment 3

Original comment:

Two implementation issues should be recognized in the discussion. i. The first of these is that perfect prediction of the location of future deforestation is impossible.

Deforestation is a rare event statistically, and so targeting programs such that only future deforested land is enrolled does not make sense, regardless of how much data you have with which to make the prediction. Even trying to approach this kind of targeting in a REDD+ program would require taking money away from actual incentives and funneling it into bureaucratic processes intended to support targeting, which would be wasteful. This means that you have to pay for enrollment of larger areas of land than are likely to be deforested.

Original response:

We agree with the reviewer that predicting the exact location of deforestation is very difficult. There are inevitable trade-offs between costs of targeting and additionality. For that reason we think that moves towards jurisdictional programmes will be positive but the inclusion of quasi-experimental approaches would be a further improvement. We now say this in the discussion:

Under JREDD+ methods such as VM0048 and ART TREES, counterfactual deforestation rates are set as the mean annual deforestation rate over a recent historic period (usually 5-6 years)¹². Our results suggest that this more straightforward and inflexible approach to estimating a counterfactual could significantly reduce over-crediting risks (Fig. 5). Further JREDD+ is designed to enable integration into Article 6 of the Paris agreement allowing jurisdictions to raise finance through the trade of International Transfers of Mitigation Outcomes³⁷. Under these methodologies, independent data providers, with no direct financial stake in the number of credits issued produce the assessment of deforestation over the historic period. This is a positive step which should reduce the perverse incentives which resulted in over-crediting in earlier project-level REDD+ methodologies.]

Unfortunately, the movement to JREDD+ does not entirely remove the risk of perverse incentives leading to over-crediting. The methods for JREDD+ assume that the drivers of deforestation remain constant between the historic period and the project period. This assumption will often not hold as forest conservation or trade policies, major infrastructure projects³⁸ or extreme climatic events³⁹ will all vary over time and can impact deforestation. In fact, ex post measures of deforestation in untreated jurisdictions have been shown to differ by as much as 100% from ex ante predictions based on historic averages^{33,40}. If projects were to enrol in JREDD+ schemes randomly without regard for the likely change in future deforestation trends, this may not result in systematic bias.

However, evidence tells us that proponents will opt into crediting schemes when it is to their advantage^{41,42}, with projects being preferentially developed in jurisdictions with declining deforestation pressures. A potential solution is to move to only issuing credits ex post based on assessments which integrate new developments in quasi-experimental methods. Ideally, different quasi-experimental results should be considered, from methods such as the Permanent Additional Carbon Tonne (PACT) method used in our study⁴³ or advances in synthetic controls that require fewer assumptions²⁶. Although quasi-experimental methods are still susceptible to unobservable, time-varying confounders, the scale of these effects can be tested and reported³¹. Therefore, until there is clear consensus on how best to measure impacts, a precautionary approach should be taken, with conservative estimates of avoided deforestation produced to safeguard the claims being made¹⁹.

New comment:

I am happy with this discussion, but this suggests that a very policy-relevant analysis would try to evaluate the improvement in assessment by comparing the JREDD counterfactual to your more precise approach.

New response:

We agree with the reviewer that this is a very policy relevant analysis and have therefore added the following statement:

“This can be tested by comparing quasi-experimental estimates of counterfactual outcomes with those from JREDD+ methods.” (lines 427 - 428)

Comment 4

Original comment:

This does not in itself undermine the principle of this paper. However, it is worth considering that the global community often says that avoided deforestation and, more recently, forest restoration, is “cheap.” The perceived cheapness of the intervention may be at least partly driven by the fact that when we calculate REDD+ transfers per hectare using current estimates of deforestation, they are very low. However, we might find that if we use current per hectare rates spread over a much lower, precisely estimated amount of avoided deforestation, we may not be able to induce any participation in these types of programs, because they will not sufficiently compensate producers for their opportunity cost. Environmental non-profit organizations and rich country governments often engage in wishful thinking regarding the true costs of nature-based solutions. It is important that the research community does not fall prey to this same mistake.

Original response:

We agree with the reviewer that there is a widespread belief that REDD+ credits are the cheap option. We have now made clear in the discussion that our finding reveals that conserving nature is likely to cost more than expected:

Indeed, despite inflated supply obscuring their real value, nature-based credits remain a critical tool in the fight against climate collapse and mass biodiversity loss.

New comment:

I think you could say this more clearly. There is a real risk that if someone decided to issue credits based upon your methodology, there would not be sufficient funds to actually get people to reduce deforestation. In many PES programs that I know of, the payment levels are set to accommodate the fact that policymakers know that they are not paying only for the additional hectares. Rather, they are spreading the cost of compensating for the additionality across a much larger forest area. Therefore, the cost per hectare of forest looks low, while the cost per hectare of avoided deforestation is high. If credits are issued based upon more precise avoided deforestation measures, this will lower payments to project participants, and will risk undermining the entire process unless the price of credits is increased.

New response:

We agree that our previous changes did not go far enough in explicitly explaining the need for commensurate increases in credit prices to account for a decrease in credit production. We have now made this clear as follows:

“Across REDD+, our analysis shows that although claims of avoided deforestation have been overestimated, the scheme has still delivered real reductions in deforestation. The challenge now is to adjust expectations about the cost of protecting tropical forests and the price of the resulting carbon credits⁵⁴. Reducing project-level supply of carbon credits by using more robust counterfactual estimates will severely restrict forest conservation finance unless there is a commensurate increase in the price of credits⁵⁷.” (lines 453 - 457)

Reviewer 2 (Remarks to the Author):

Thank you for the opportunity to read the revised paper. I very much appreciate that most of my comments have been addressed. Previously, my main concerns focused on the manuscript's contribution and the use of empirical language/concepts. The authors have addressed both aspects adequately.

I only have a few minor editing comments and a small suggestion about the title.

Comment 1

The new title is slightly misleading: "Understanding the reasons" implies that the authors will analyze the political and economic reasons of over-crediting. E.g., certification institutions have an economic incentive to certify higher avoided deforestation figures in order to please the client. This manuscript analyses the mechanisms of over-crediting. I suggest reformulating the title to "Understanding the mechanisms for over-crediting in REDD+".

Response:

We agree with the reviewer that their suggestion is better and have updated the title accordingly.

Comment 2

Original comment:

Figure 1b shows there ratio $\frac{\text{Avoided deforestation credited by VERRA}}{\text{Avoided deforestation using quasi-experimental methods}}$. For overestimation, the ratio is < 1 , and for underestimation, the ratio is > 1 . I suggest flipping the definition of the ratio so that values > 1 reflect overestimation.

Original response:

We agree with the reviewer that this is confusing however because quasi-experimental methods tend to estimate values close to, at or even below zero, whereas certifying methods are all positive, the proposed approach would make it impossible to present all the projects as they would have very large or infinite value. Logging the data is also problematic due to the negative values. Therefore we have opted to retain the same presentation. However, we now presented over-crediting as suggested in the results:

"The mean additionality ratio (calculated as the mean quasi-experimental estimate of avoided deforestation / certified estimate, averaged across the 44 projects claiming to have avoided deforestation) of 0.27 (Fig. 2b) was significantly < 0 (one-tailed Mann-Whitney U against a group of 0s, $U = 1628$, $p < 0.001$). This suggests that on average projects were additional but had over-credited by a factor of 3.8."

New comment:

You could use an inverse asymptotic sine transformation. It has similar properties to the log-transformation, and it is defined for negative values. Imagine I use this figure in a lecture: It is complicated to explain to students that a point with a value of 0.16 on the y-axis means overcrediting, and they should calculate $1/0.16=6.25$, to understand that VERA calculated a 625% higher level of avoided deforestation than the quasi-experimental method.

New response:

We thank the reviewer for their comment and have substantially reworked Figure 2. We explored incorporating the suggested inverse sine transformation in the graph. This process prompted a re-examination of the underlying data, leading to the following adjustments in interpretation.

First, we reassessed the additionality ratio used in our initial analysis. Previously, we only calculated this ratio by dividing the mean quasi-experimental estimate by the certified estimate, a statistic similar to the “OAR” used by Probst et al. (2024). However, as the reviewer noted, this resulted in a confusing presentation in Figure 2, where smaller values indicated greater over-crediting. We have now inverted the ratio for the figure and in the supporting text as suggested. This adjustment, however, introduced an unintended complication: projects with negative quasi-experimental estimates remained below the zero line, with values closer to zero suggesting *less* over-crediting. Consequently, the interpretation of the scales became inconsistent depending on the sign of the value. Moreover, confidence intervals crossing the $y = 0$ line ceased to convey meaningful information.

To address this, we made the following revisions to the ratios plot in Figure 2b. We removed confidence intervals and highlighted negative values in red, which we positioned at the top of the plot. We also updated our statistical analysis to use bootstrapping to derive confidence intervals for the means. To this end we introduced an additional test for over-crediting, evaluating whether projects would remain over-credited even when using the most generous quasi-experimental estimate within their respective 95% confidence interval ranges.

In the process of exploring the negative values we also noticed that these had been handled inconsistently across the quasi-experimental methods. Projects assessed to have negative additionality or no significant additionality were assigned a value of zero by Guizar et al. (2024) and West et al. (2023). In our revision we now include these values as negatives, except in the calculation of the OAR. We also identified that we had been working with a version of the West et al. (2023) data from their pro-print but that was updated on publication. Finally, Guizar et al. (2024) has now been updated (it was an early pre-print but is now submitted to a journal and includes more sophisticated analysis). We have therefore integrated the most up to date data into our analysis. Together, these changes have influenced the global mean additionality ratio and our presentation of it. These changes are described in full below.

From Results:

“The majority of projects (35 of 43) did reduce deforestation (Figure 2). For the projects with multiple quasi-experimental estimates, 15 (40% of the 37 projects with calculable confidence

intervals) showed statistically significant additionality at the 95% confidence level. Just eight projects (shown in red in Figure 2) experienced more deforestation than their quasi-experimental controls.

Despite most projects reducing deforestation, over-crediting was widespread with certified estimates of avoided deforestation significantly greater than mean quasi-experimental estimates (one-tailed paired Wilcoxon signed-rank test; $V = 925$, $p < 0.001$). Over-crediting was statistically significant at the 95% confidence level for 29 projects (79% of the 37 with calculable confidence intervals; Fig. 2a). Consequently, the mean additionality ratio was 4.3 (95% bootstrapped CI: 2.9 to 7.3), indicating it offered about a fifth of the avoided deforestation reported by certified estimates. Across all the credits assessed, the global additionality ratio (the certified estimate divided by the mean quasi-experimental estimate) was 9.3 (95% bootstrapped CI: 4.3 to 23.6), indicating that a credit bought at random would offer only a tenth of the avoided deforestation claimed (Fig. 2b). However, it is important to note that over-crediting was not universal: the three projects from Madagascar, for example, have a mean additionality ratio of 0.9 meaning the certified and quasi-experimental estimates were similar (S8).” (lines 451 - 469)

Figure 2: Estimates of avoided deforestation from quasi-experimental approaches were frequently lower than certified estimates, indicating widespread over-crediting. (a) REDD+ projects' certified estimates of avoided deforestation against mean estimates from the five quasi-experimental methods, error bars show 95% confidence intervals. (b) The additionality ratio (the certified estimate divided by the mean quasi-experimental estimate) against certified estimates of avoided deforestation, the dashed grey line shows the global additionality ratio. In both panels y-axes are on a log scale. Dashed red lines show where certified and quasi-experimental estimates of avoided deforestation were equal. Points above and below the line were assessed to have over-credited and under-credited respectively. Darker colours indicate projects that were assessed by a greater number of quasi-experimental approaches (to a maximum of four because there was no overlap in the projects assessed by the two West et al. studies). The additionality ratio cannot be calculated for the eight red points as they experienced more deforestation than predicted by their quasi-experimental controls.

And from Methods:

“For each of the 43 projects, we compiled the certified estimates of annualised avoided deforestation in hectares, along with all available quasi-experimental estimates. For the 37 projects with more than one quasi-experimental estimate, we calculated the mean, standard deviation, coefficient of variation, standard error, and 95% confidence intervals. For the remaining six projects, which had only a single quasi-experimental estimate, this value was taken to be the mean.

We determined how many projects had mean quasi-experimental estimates below the certified estimate, as well as how many had upper 95% confidence intervals below the certified estimate. To test whether certified estimates were significantly greater than mean quasi-experimental estimates, we applied a one-tailed Wilcoxon signed-ranks test, due to the non-normality of the data. We also evaluated how many projects exhibited quasi-experimental avoided deforestation estimates that were significantly greater than zero, using the lower bound of the 95% confidence intervals.

We also assessed the over-crediting as the ratio of the certified to the mean quasi-experimental estimate of avoided deforestation. We refer to this as the additionality ratio, which expresses the number of times more credits produced through certification than suggested by quasi-experimental estimates. The mean additionality ratio was calculated for all projects that issued credits that also had positive mean quasi-experimental estimates. Projects with negative mean quasi-experimental estimates were excluded. The global additionality ratio, reflecting how many certified avoided deforestation units likely correspond to one quasi-experimentally validated unit across the REDD+ portfolio, we divided the total certified avoided deforestation by the total of the mean quasi-experimental estimates (including the negative values). Both the mean additionality ratio and the global additionality ratio were bootstrapped using 10,000 resamples with replacement to generate 95% confidence intervals.” (lines 632 - 658)

Comment 3

Line 51. "Unfortunately, over-crediting by REDD+ Projects is still not widely accepted." It is unclear who is not accepting and why everybody should accept. I recommend using a milder, scientifically impartial language.

Response:

We are grateful for this comment regarding more impartial language and have updated the start of the specific paragraph to the following:

"Without recognition of the scale of over-crediting, and an understanding of the underlying mechanisms, there is a risk that the same mistakes will be repeated in new generations of credits, seriously undermining the impact of nature-finance." (lines 57 - 59)

Hopefully this new text removes a sense of blame while simultaneously conveying the importance of our study as requested by reviewer 1.

Comment 4

The introduction should have a brief paragraph summarizing the results.

Response:

We appreciate the request for a paragraph highlighting the scientific contribution. In order to maintain the flow of the introduction as a whole, we have attempted to achieve this in a single paragraph that summarises both our scientific contribution and our results. As follows:

"Our study contributes to the literature examining how to infer causal effects from observational data^{27,28}, especially in conservation where randomised control trials are rare or impossible^{19,20}. Our results show clearly that despite variation in estimates from quasi-experimental methods, there is consistent evidence of over-crediting. This result cannot be explained by differences between the remote sensing layers used by quasi-experimental and certified assessments. Instead, we find that over-crediting was primarily driven by the selection of control areas with greater exposure to deforestation than projects and through unrealistic modelling of future deforestation rates." (lines 125 to 133)

Comment 5

The introduction should have a paragraph highlighting the scientific contribution: "We contribute to the literature on ..."

Response:

See response to comment 4.

Comment 6

Abbreviations are used but not explained in the main text (as far as I see). E.g. ACC, QEM.

Response:

We thank the reviewer for catching this! These have now been defined the first time they are used in both the main text and the supplementary materials.

Comment 7

Line 430. The concept of "dynamic baselines" is recommended, but not explained at any Point.

Response:

We are grateful to the reviewer for highlighting this. We have now removed the reference to dynamic baselines and use the following text instead:

"Formalising the use of quasi-experimental methods with ex post counterfactuals to account for dynamic changes in deforestation risk, would considerably reduce the risk of over-crediting in future^{53,54}." (lines 463 - 464)

Reviewer #3 (Remarks to the Author):

There is no reviewer 3

Reviewer #4 (Remarks to the Author):

The paper has the potential to make a significant contribution. However, while the manuscript has improved significantly in terms of the analysis and writing, it is still not ready for publication, in my opinion. There are 2 main remaining challenges: The quality of the writing and the lack of proper description and justification of statistical techniques.

Comment 1

Writing:

While the contributions of the manuscript and the description of the methods are much clearer, grammatical errors and overly complicated sentences still abound in the text.

Response:

We are very sorry this reviewer felt that this is the case. We have spotted and corrected a few grammatical errors and do not believe there are others. It is always challenging to get the balance right in highly-technical writing: what one person considers an overly complicated sentence may be considered by others as including crucial detail to allow a judgement to be made on the methods. We have worked very hard in this revision to make the language as clear as possible. Specifically we have ensured each paragraph starts with a clear topic sentence and have shortened many of the longer sentences. We also asked independent colleagues to read the paper and flag parts which they find difficult and have revised iteratively.

Comment 2

Does deforestation in the bespoke layer capture forest loss due to road construction? Do the Verra approaches consider reforestation as a way to offset forest loss? It is important to clarify what the original certification mentions consider “deforestation” as some forest loss is not deforestation.

Response:

All VERRA methods only assess deforestation using area-based forest vs non-forest classifications commonly between two points in time (although more time points can be used). All forms of forest loss within project areas are meant to be recorded in monitoring documents, including from road construction. However, short-term disturbance events may be missed, including from regrowth. We explain this in the discussion as follows:

... we found that the ACC detected similar or higher deforestation rates in project areas than certified estimates using bespoke remote sensing layers (Fig. 3 and S4). This may well have occurred because the ACC is a multi-temporal approach that considers all disturbance events occurring since the start of the time series⁵, whereas bespoke approaches typically only detect non-forest classes at two points in time, which may miss short-term disturbances³⁶. (lines 357 - 358)

Comment 3

What is the default setting for GDALDEM for slope calculations? Is it degrees? (line 514)

Response:

We thank the reviewer for highlighting the lack of clarity on our part regarding the slope variable. We have updated the section to now state the following:

“From the elevation layer we calculated slope using GDALDEM (v3.8.4) with default settings. This produced a slope variable consisting of integer values representing the mean slope in each raster cell ranging from 0-90°.” (lines 563 - 566)

Comment 4

I suggest including citations of the actual datasets used in the analyses (e.g. lines 503-506) and not referencing just the secondary studies that use these datasets

Response:

We thank the reviewer for spotting this omission. We have now included references to Mapbiomas and Global Forest Change. With the text updated as follows:

“Deforestation was assessed using publicly available remote sensing products for tracking changes in forest cover, specifically the corrected Mappiomas dataset used by West et al. (2020)⁶, the Global Forest Change (GFC) dataset used by West et al. (2023)⁸, and the Tropical Moist Forest Annual Change Collection (ACC)⁵ used by the Guizar-Coutiño et al.^{7,39} and PACT analyses. GFC reports deforestation as the year of gross forest loss.” (lines 520 - 524)

Comment 5

I don't know what the text on lines 625, 279-280, and 59-60 mean. For example, what are “areas used for spatial modeling”?

Response:

We have now edited the text to make our meaning clear.

Instead of:

We chose to focus on control areas used to measure deforestation rates and not areas used for spatial modelling. (line 625)

We now say:

For certified control areas, we focused on the ‘Reference Regions for Deforestation’ (RDD) used by certified methods as these were used to measure the rate of deforestation, rather than the ‘Reference Regions for Location’, which were used to measure how much of the deforestation was expected to happen in the project. (lines 703 - 705)

Instead of:

Yet the deforestation rates used in certified ex-ante modelling were produced from bespoke measurements of deforestation in certified control areas

We now say:

“This was difficult because we did not have access to the certified deforestation rates in control areas, so instead inferred them by multiplying the ACC control area deforestation rates by 0.58 (the median ratio of the bespoke to ACC rates in the 17 projects in this analysis).” (lines 291 - 294)

Instead of:

(1) their results lack consistency due to variations in model specification¹⁷, and (2) the globally available, remotely sensed forest cover datasets they rely on detect less deforestation than bespoke layers¹⁸.

We now say:

“The criticisms levelled at the independent studies are based on two key arguments: (1) their estimates of avoided deforestation are inconsistent due to variation in how analyses were conducted¹⁷, and (2) the publicly available forest cover datasets used by these studies detect less deforestation than the bespoke remote sensing classifications used by projects¹⁸.” (lines 61 - 65)

Comment 6

Why introduce the synthetic control method in lines 90-93? It seems out of place. Also, as outlined in my previous report, newer methods like synthetic difference-in-difference may outperform synthetic controls (Arkhangelskiy et al. 2021).

Response:

We agree with the reviewer that the statement on synthetic controls was a little out of place in isolation. We have now changed the text to improve our point that there are various quasi-experimental approaches available to, including adding a reference to Arkhangelskiy et al. 2021:

“A related approach is the synthetic control method that can replicate pre-project trends through the selection and weighting of untreated units^{8,23-25}. This has been further refined in the form of synthetic difference-in-differences to more effectively account for time-varying and unobserved confounders²⁶.” (lines 94 - 98)

Comment 7

What is bootstrapped in the PACT method (Table 1 and line 494)? How is the bootstrap done?

Response:

We have substantially reworked our explanation of the PACT methodology to add clarity. The reference to bootstrapping was due to PACT using 100 unique iterations of matching sets from which to calculate the mean additionality across. This is now explained with the following text:

“This process was repeated to produce 100 matched project-control sets, each set is thus a length equivalent to 10% of the project units set, and thus the total sample across all sets is ten times larger than the project units set.”

Ex post deforestation was measured for the project and control units by calculating changes in the undisturbed class since the start of the project. To calculate the area of undisturbed forest at each time point, we calculated the proportion of all units in the undisturbed class before multiplying by the project area. This was done across all 100 matched sets to produce estimates of project forest cover and counterfactual forest cover.

Averaging across these sets, we produced annualised mean forest cover in hectares. By subtracting the counterfactual forest cover by the project forest cover, annual cumulative additionality/avoided deforestation figures were computed. Total avoided deforestation during the project period was measured at the final year of the project's most recent monitoring period (as the additionality figures are cumulative).” (lines 613 - 627)

Comment 8

I would clarify that Fig 1a is similar to a quantile-quantile plot for log-transformed data. Currently, that figure and its legend are very difficult to read. I would clarify in the axes labels that the outcome is log-transformed.

Response:

In response to Reviewer 2 Comment 2, we have changed this figure and updated its analysis substantially. We believe that these changes have considerably improved the clarity of the figure. With regards to the scales on the y-axes whilst the units are not log transformed, the breaks on the axes are, we have made the following clarification in the description:

“In both panels the y-axis is on a log scale” (line 184)

Comment 9

Reword that you are showing 95% Confidence Intervals instead of “confidence intervals show 1.96 standard errors” (line 174-175)

Response:

We have made the suggested alteration.

Comment 10

The presence and location of spillovers depends on the extent of markets (e.g., see Pfaff & Robalino 2017 for a review and Miteva et al. 2017 for an example of a specific mechanism of spillovers as a function of markets). The text mentions that some of the projects are large (lines 526-527). Given the lack of underlying theory of change, the manuscript cannot model where spillovers are likely to occur. However, this point should be acknowledged as a limitation in the discussion.

Response:

We thank the reviewer for highlighting this and added the following to our discussion section on limitations:

“Finally, REDD+ projects could cause spillover effects within their vicinity, which would affect the deforestation observed within control units⁴¹⁻⁴³. While some studies have attempted to

overcome this issue by only matching to areas greater than some distance from project boundaries⁴⁴, this strategy may be insufficient because of the complex and hard to predict way in which spillovers can occur.” (lines 406 - 411)

Comment 11

Reword Q4 (line 256)

Response:

We agree with the reviewer that the original wording of the question title was unclear, and have reworded it as follows:

“Q4. After isolating the effects of forest cover layer and control area selection, how much of over-crediting can be explained by ex ante modelling?” (lines 270 - 271)

Comment 12

I am not sure what lines 285-286 and lines 343-344 mean—a critique of a specific ex ante method? A general critique of ex ante methods?

Response:

We agree that this wasn't completely clear. We have completely rewritten the results section for Q4 to explain more clearly our finding that a large proportion of over-crediting is likely due to the various *ex ante* modelling approaches taken:

Finally, having adjusted for control area selection and the use of different deforestation rate estimates, we attributed the remaining 78% of over-crediting to the various ex ante modelling approaches used in certified assessments.

In the discussion, instead of:

Predicting an ex ante counterfactual scenario is difficult. However, flexibility in the modelling approaches used to estimate this counterfactual likely played a major role in over-crediting.

We now say:

“Projecting a counterfactual scenario of deforestation, likely to occur in the absence of a REDD+ project ex ante, is inevitably fraught with uncertainties and appears to be a key driver of over-crediting. We suggest this is the predominant mechanism resulting in over-crediting (Fig. 5), after accounting for the effect of control area selection and different deforestation rate estimates.” (lines 371 - 375)

Comment 13

Fig 5 is very hard to read: Why are there bars that don't correspond to X axis labels?

Response:

We thank the reviewer for pushing us to look at this figure again. We have simplified the plot and labelled it more clearly. This involved removing the stacked bars, which could be seen as a source of confusion.

Figure 5. Isolating the contribution of different possible mechanisms by which over-crediting occurred. The avoided deforestation rates from the subset of 17 projects for which this analysis is possible projects produced using different combinations of quasi-experimental and certified estimates in project and control areas. The percentage of over-crediting attributable to each possible mechanism is revealed through sequential comparisons of the avoided deforestation estimated by the different combinations. ACC refers to the tropical moist forest annual change collection used by quasi-experimental (QE) PACT method, and bespoke* indicates the values inferred by multiplying the ACC values by 0.58 (the median ratio of the bespoke to ACC rates in the projects). Error bars show medians and interquartile ranges with significance indicated by ** $p < 0.01$.

Comment 14

I am not following the point on lines 334-335. If something is not statistically significant, the coefficient is 0. Unless the point the text is trying to make is that there is a lot of heterogeneity (hence the wide confidence intervals)?

Response:

We have made the changed the text, to clarify what we mean, so that instead of:

“Though this relationship lacked statistical significance (Fig. 4), when isolated from other explanations, elevated ACC deforestation within certified control areas contributed around 57% to total over-crediting”

We now state:

When measured using the ACC, certified control areas had higher rates of deforestation than their quasi-experimental counterparts (Fig.4), which likely contributed substantially to over-crediting (Fig. 5). (lines 367 - 369)

Comment 15

I would emphasize that the credibility of quasi-experimental designs depends on, first and foremost, the theory of change that helps us formulate hypotheses, select relevant to a specific context covariates, select a proper estimation approach, and explain observed patterns (lines 361-365). Also, the robustness of quasi-experimental designs can be assessed via sensitivity analyses (lines 363-364).

Response:

This is an important point, we have updated the manuscript to incorporate it. First, we now explicitly mention a theory of change in the introduction:

“All methods required control areas to be similar to project areas in terms of drivers of deforestation in their landscapes, which were selected according to their reported ability to predict deforestation alongside a landscape-level theory of change.” (lines 76 - 78)

We have also explicitly described our causal model of land use change in our methods section:

“Secondly, layers were compiled to select control units analogous to project units using a method informed by a causal model of land use change. This was developed from empirical evidence of the effect of local policies and regulations, economic pressures, historical changes in land cover, and environmental conditions on deforestation rates and conservation efforts.” (lines 556 - 559)

Regarding the sensitivity analysis, we have updated the limitations paragraph in the discussion section to both reflect how new advancements in the field of QEMs have produced results relating to their performance under sensitivity analysis and in a placebo landscape setting:

“However, simulations of landscapes where the rates of avoided deforestation are known or manipulated can be used to demonstrate the reliability of these approaches³⁷. For example, new evidence demonstrates that the PACT method can predict 83% of the variance in

tropical forest deforestation rates across such a set of untreated ‘placebo’ landscapes³⁸. Secondly, a core assumption of some of the quasi-experimental methods used is that all significant confounders (covariates associated both with where REDD+ projects are implemented and with deforestation) are controlled for²⁰. Given that quasi-experimental studies apply the same methods across multiple projects, they are not accounting for local drivers of deforestation which may also be associated with the placement of REDD+ projects. However, Guizar-Coutiño et. al. (2025) explore the potential influence of such hidden confounders and reveal they would need to have an effect size multiple times larger than the most impactful observed covariates to undermine this finding³⁹.” (lines 388 - 400)

Comment 16

What is a “modified method” (line 461)?

Response:

We have now addressed this lack of clarity by reformatting the section as follows:

“We also included results from a study under review by Guizar-Coutiño et al.³⁹, which implemented a novel framework to reanalyse the set of projects examined in Guizar-Coutiño et al.⁷ study. In this newer study 7.1 ha circular sample regions were used instead of 0.09 ha pixels to characterise forest loss observations. The causal impact of REDD+ projects on deforestation was examined by adjusting for observed confounders and pre-treatment deforestation trends, using a two-stage matching approach, commencing with propensity score matching via random forests, followed by propensity score subclassification on the matched datasets. The average treatment effect of the treated was examined using multiple model specifications and sensitivity to unobserved confounders was assessed using the sensemakr framework²⁹. The avoided deforestation estimates used in our study correspond to the doubly robust model specification reported in their study.” (lines 496 - 508)

Comment 17

Lines 494-496 & 627-635 need a citation to back up the approaches

Response:

We agree that references are necessary to justify the use of the bootstrapping approach used by PACT and the georeferencing approach used for digitising the certified control areas. For the former we reference the original Efron and Tibshirani (1985) paper as well as Austin and Small (2014), which talks about the use bootstrapping in propensity score matching:

“The PACT method used a bootstrapped one-to-one pixel (0.09 ha) matching approach to produce paired project and control units from which to estimate mean project and counterfactual outcomes^{64,65}.” (lines 541 - 543)

For the approach used in digitising the certified control areas we reference the best practice article on digitising maps produced by ESRI:

“Certified control area polygons were not publicly available as shapefiles so were digitised by georeferencing maps available in project design documents (from the Verra Registry) either by tracing the polygons by hand in QGIS (v3.26.3) or by a colour thresholding and automated polygonisation procedure in R, equivalent to the process described by the Environmental Systems Research Institute⁷⁰.” (lines 707 - 711)

Comment 18

Which correlation was used --Spearman vs. Pearson (line 566)?

Response:

We thank the reviewer for highlighting this discrepancy. In our initial analysis of Q1, we used a Pearson correlation to test if the ACC rate of deforestation within project areas was aligned with the Certified rate. However, this result has now been omitted from the manuscript in the formatted Q1 section.

Comment 19

Since F_0 is a share, lines 595-596 do not make sense as worded.

Response:

We have modified the text to provide more clarity.

“To determine F_0 we took the proportion of the ACC pixels that were in the undisturbed class for the yearly layer closest to the project start date and multiplied this by the area of the project, measured from the project polygon. Thus F_0 was the total area of undisturbed forest in hectares at the start of the project.” (lines 672 - 675)

Comment 20

The manuscript supports the range for, what is referred to as “standardized mean difference” by citing a recent paper in Conservation Biology (Ref. 22). Given the long history of the statistic in other disciplines, I would reference studies by statisticians or econometricians who propose that cut-off as a rule of thumb (e.g. Abadie & Imbens, 2011; Imbens et al. 2009 but also earlier studies). Also, line 551 references an article in the Guardian and not a statistics study.

Response:

Thanks. We agree that there are more appropriate references. We now refer to Rosenbaum and Rubins 1985 paper, as well as a Stuart’s 2010 review of matching methods for causal inference. We have done this on each mention, which corrects the erroneous reference to the Guardian article.

Comment 21

It is still not clear to me why the main text of the manuscript uses only “undisturbed” forests in the Vancutsem et al (2021) data and not all forest including forest regrowth (e.g.,

paragraph starting on line 593). Only a note in passing is made in the text that some analysis using all 4 forest categories can be found in the appendix (although I could not find that--Section S4 uses just deforestation and degradation). The main text does not clarify what the comparison indicates (lines 616-617).

Response:

Deciding how best to treat the data layers is, to some extent, a judgement call. We chose not to include degraded land in the definition of forest cover (instead focusing solely on undisturbed areas), as we were influenced by research from Holcomb et al 2024., which finds that degradation within the ACC corresponds with a significant loss of biomass which is relevant to REDD+. Expanding this analysis to regrowth we find that it has mean biomass lower than the degraded class. We have now added this analysis to the supplementary material in the form of S11 and further clarify our decision in the text.

We have updated the text to state the following:

“PPACT considered degradation of the undisturbed class alongside deforestation, because this has been shown to result in biomass reduction of >50% within 12 months⁶¹. Our own analysis of the above ground biomass densities for different ACC forest classes for numerous project areas also showed that regrowth landscapes were also far below the biomass of their undisturbed counterparts (see S11).” (lines 528 - 532)

Furthermore we have expanded the analysis presented in S4, which now also shows how including the regrowth class as forest cover lowers the ACC measured deforestation rates. Thus, because we classify both degradation and regrowth as forms of forest loss, our PACT-derived estimates of avoided deforestation are likely higher. This is due to the fact that our approach detects more deforestation in the counterfactuals, which has a greater impact on determining the magnitude of additionality (i.e., avoided deforestation) than measurements within the project area itself.

Comment 22

Lines 447-449 mention that whenever deforestation rates were not available, they were “totaled over the evaluation period”. However, later the methods present a compound formula for calculating the average deforestation over time (Eq. 2.2)—why were 2 different methods used?

Response:

We thank the reviewer for catching this mistake. In a prior version of the manuscript, the approach we used to extract annual deforestation rates was the mean of the year on year change. However, for some projects this information was absent, meaning that instead a compounded rate over the evaluation period, was used. Now we use a compounded rate for all projects and use cases, as detailed in Eq. 2.2. We have now modified the prior reference to the following so that it is less misleading.

“Where avoided deforestation was not reported, it was calculated by subtracting project deforestation from counterfactual deforestation. These figures were then combined with the

during of the evaluation period (which had a mean length of 6.4 years) to produce annualised compound avoided deforestation rates (Eq. 2.2).” (lines 480 - 484)

Comment 23

Statistics:

The credibility of counterfactual analyses depends on the underlying theory of change. Omitting key covariates (e.g., proximity to rivers, markets which may not be necessarily captured by the travel time to health centers etc) may drastically change the control units and, thus, results. For this reason, it is necessary to justify the baseline covariates used in the quasi-experimental design with some theory of change. The current version of the manuscript mentions only a handful, some of which may not refer to the baseline (period prior to the intervention) (lines 503-524).

Response:

We strongly agree with the reviewer’s comment. It is, of course, essential that the most appropriate covariates are included in order to select the most appropriate controls. Nevertheless, we have selected the covariates carefully to ensure they do properly reflect the underlying causal mechanisms that result in deforestation. We think it is disputable whether proximity to rivers is a better covariate than time to healthcare layer. This is because the travel to healthcare layer integrates over all known travel routes, including rivers, as well as terrain, land cover, and road quality to assess the travel time. We completely accept that healthcare centres may well not be the most appropriate proxy for economic pressures and that it would be better to use destinations that are economically important, such as processing facilities, distribution centres and markets, but, as far as we are aware, these are not generally available as a single accessibility layer. Economic pressure is also somewhat captured using our proportional cover layers which describe the trajectory of land use change in the surrounding area through time.

We now better explain our choice of covariates in the methods and acknowledge their limitations:

Secondly, layers were compiled to select control units analogous to project units using a method informed by a causal model of land use change. This was developed from covariates that encode the effect of local policies and regulations, economic pressures, historical changes in land cover, and environmental conditions on deforestation rates and conservation efforts. We used the following layers: International country borders produced by OpenStreetMap⁶⁴ to encompass national regulatory limits, Ecoregions provided by Resolve⁶⁵ to ensure biotic equivalence, and elevation provided by NASA’s Shuttle Radar Topography Mission⁶⁶ as a proxy for abiotic conditions. From the elevation layer we calculated slope using GDALDEM (v3.8.4) with default settings, which is an important proxy for accessibility and suitability for economic activities (this produced a slope variable consisting of integer values representing the mean slope in each raster cell ranging from 0-90°). Motorised accessibility to healthcare in 2019 provided by the Malaria Atlas Project⁶⁷ was used as a proxy for market exposure/economic pressure. This accessibility layer integrates all known travel routes, including rivers, as well as terrain, land cover, and road

quality to assess the travel time to population centres large enough to have a healthcare facility. Finally, we also computed the proportional cover of undisturbed and deforested classes from each year of the ACC time-series (1990-2021) to assess land use change trajectories through time in the surrounding area. Proportional cover was calculated by counting the number of pixels in each class and dividing by the total number of pixels within the 1 km radius neighbourhood around each pixel. It is important to note that accurate data regarding land value and accessibility to markets is not universally available, particular in frontier landscapes. As such we use spatial proxies such as the proportional cover and motorised accessibility to healthcare, recognising that these leave space for unobserved confounders. (lines 556 - 588)

We also agree that missing confounders could affect the results of our study but new work we have conducted, testing our models using placebo projects, finds they predict 83% of the variation in counterfactual outcomes over a 10 year period. And, we found that omitted confounders would have to have effects much larger than the most important covariates included in order to remove the signal of over-crediting. We have added this information in the discussion:

...new evidence demonstrates that the PACT method can predict 83% of the variance in tropical forest deforestation rates across such a set of untreated 'placebo' landscapes³⁸. Secondly, a core assumption of some of the quasi-experimental methods used is that all significant confounders (covariates associated both with where REDD+ projects are implemented and with deforestation) are controlled for²⁰. Given that quasi-experimental studies apply the same methods across multiple projects, they are not accounting for local drivers of deforestation which may also be associated with the placement of REDD+ projects. However, Guizar-Coutiño et. al. (2025) explore the potential influence of such hidden confounders and reveal they would need to have an effect size multiple times larger than the most impactful observed covariates to undermine this finding³⁹. (lines 390 - 400)

Comment 24

Why was it necessary to limit the analysis to only 100 matched pairs based on 10% of the data (lines 540-545)?

Response:

Our apologies for the lack of clarity in explaining the matching process. We did not limit the analysis to 100 matched pairs. We have updated the text below to make the method clear:

"This process was repeated to produce 100 matched project-control sets, where the number of points in each set was 10% of the total number of points sampled from within the projects.

Ex post deforestation was measured for the project and control units by calculating changes in the undisturbed class since the start of the project. To calculate the area of undisturbed forest at each time point, we calculated the proportion of all units in the undisturbed class before multiplying by the project area. This was done across all 100 matched sets to produce estimates of project forest cover and counterfactual forest cover.

Averaging across these sets, annualised mean figures are produced. Subtracting the control metric by the project metric, annual additionality/avoided deforestation figures are computed. Total avoided deforestation during the project period was measured as the difference between the figures for project start year and the final year of the project's most recent monitoring period.” (lines 613 - 627)

Comment 25

Why was a t-test used to assess differences in the distributions for the certified areas (line 654), whereas other assessments were based on the standardized mean difference? The issue is that the t-statistic is a function of the sample size: Just increasing the sample size can make a t-statistic statistically significant, ceteris paribus (Imbens & Wooldridge 2009). Conversely, a t-test based on a small # observations is likely to have insufficient power.

Response:

We thank the reviewer for this comment. We have updated the text to clarify that the t-tests were only used across the distribution of SMDs at the project level. This was to test if across the project units the SMDs were significantly different from zero. This means that the n for the t-tests was only 17.

Across the set of projects, we tested whether the sample population of standardised mean differences were different from zero for each of the characteristics using t-tests. (lines 734 - 735)

Response to reviewers

NCOMMS-24-45991-T: Learning lessons from over-crediting to ensure additionality in forest carbon credits

We would like to thank the reviewers for their extremely helpful and valued comments. Their suggestions have helped greatly strengthen the analysis and improved our exposition of its novelty and relevance. Below we provide a detailed, point by point response (no highlight) to the reviewers' comments (highlighted in purple).

Reviewer 1 (Remarks to the Author):

I am satisfied with the responses to most of my remaining comments.

Comment 1

Comment 1 round 3:

Regarding my previous statement on the extent of your contribution and your response, I believe that we simply have a disagreement of opinion. I think that most researchers believe that their work is "vital", which is why they continue to do it. The fact that your previously results are "still not widely accepted" is not surprising, as research can take time to be integrated into policy and private sector spheres. However, a frustration that people aren't paying sufficient attention to what you have said isn't necessarily a strong justification for publishing results that are very similar to previous ones.

That said, the science of what you have done here is sound, and there is sufficient detail for replication to occur. Your conclusions follow logically from the analysis and so you have met key benchmarks for publishable research.

Response 3 round 3:

The reviewer is correct that despite not being widely accepted by many involved in forest carbon markets, the evidence in the academic literature for over-crediting in REDD+ project is overwhelming. This is even more true now following a further study demonstrating over-crediting, recently published in Science (Tang et al. 2025). As a result, in this rewrite, while we present a synthesis of the evidence on the extent of over-crediting (now from six empirical studies, having updated our analysis to include the Tang et al. estimates), we have substantially de-emphasized over-crediting as our main message. Instead, the manuscript (MS) is now framed around the ongoing interest in forest carbon credits contributing to closing the forest finance gap and the lessons that can be taken to ensure avoided deforestation carbon credit represents the avoided deforestation claimed. We have done this through the following 9 changes:

1. In paragraph 1 of the introduction we set out the background for our work, highlighting that there is a need to understand why over-crediting occurred:

L49-58: *“There is substantial global interest in the potential for both compliance and voluntary markets to fund forest conservation despite widespread evidence that many first-generation REDD+ (Reducing Emissions from Deforestation and forest Degradation) projects have issued more credits than were justified by their impact on deforestation^{7–11}. The resulting scandal¹² contributed to more than US \$1.1Bn being wiped from the value of the voluntary carbon market in 2023 and a further contraction in 2024¹³. Understanding what these early projects did and did not achieve, and the mechanisms that resulted in over-crediting, is key to informing efforts to improve the integrity of forest carbon credits, which is an essential step if they are to contribute to closing the forest finance gap.”*

2. We situate our research within the conservation literature at the beginning of the second paragraph where we say:

L60-62: *“A major challenge in measuring the impacts of forest conservation projects is estimating counterfactual outcomes (what would have happened to deforestation in the absence of a project)^{14–18}.”*

3. Then we situate the need for our research within the existing REDD+ literature. We highlight that, although the evidence of over-crediting is clear, the reasons why it occurred are largely hypothetical, which creates a clear need to understand why it happened:

L72-83: *“The prominence of REDD+ has drawn scrutiny from several independent research groups that have applied these quasi-experimental methods to evaluate the impact of first-generation REDD+ projects on deforestation^{7–9,23–25}. Consistently, these studies have found evidence of over-crediting. Proposed explanations include the use of inappropriate reference areas, unrealistic ex ante modelling that exaggerated expected deforestation, and selective use of certification methodologies that inflated credit issuance^{7,9,11,23}. However, these hypothesised mechanisms have not been systematically tested¹⁰. Some industry insiders have rejected the evidence for over-crediting altogether^{26,27}, arguing that (1) quasi-experimental estimates of avoided deforestation are inconsistent across studies due to methodological differences²⁸, and (2) over-crediting is an artefact of using of publicly available global forest cover datasets that detect less deforestation than the bespoke remote sensing classifications used by projects²⁹.”*

4. Subsequently we contextualise the role for carbon markets. Crucially, we do not take a position on whether markets are or are not a good thing (we cite key literature which lays out critiques beyond the questions of additionality and over-crediting addressed by our paper). However, as long as avoided deforestation carbon credits are being traded (and there are many initiatives which are using these credits), it is crucial that one credit represents the avoided deforestation that is claimed:

L85-94: *“Carbon markets are, of course, not the only way tropical forest conservation can be funded^{30,31}. There are well-known problems associated with carbon offsetting³² and with the integrity of forest carbon credits beyond the issues of over-crediting and additionality^{33,34}. Nevertheless, carbon markets are likely to play a role in bridging the forest funding gap, at least in the short to medium term⁶. It is therefore critical to learn as much as possible from the first-generation of REDD+ projects, both their successes and failures, to inform the future*

development of methods for certifying credits. Given how harmful over-crediting is to the overall objective of carbon markets (reducing carbon in the atmosphere), understanding the mechanisms through which over-crediting occurred is vital if future approaches are to avoid repeating the failures of the past.”

- We have redrawn Figure 2 to emphasize that while almost all projects sold too many credits, the majority did indeed slow deforestation relative to a credible counterfactual. This is an important point which has received little attention to date.

- In the discussion we now have a new paragraph that highlights how our study changes the interpretation of over-crediting:

L339-347: “While some industry-linked commentators continue to deny that over-crediting in first-generation REDD+ projects was a problem^{26,27,29}, the scientific evidence is now clear. Our synthesis shows that across the portfolio of projects examined, almost 11 times more credits were issued than was justified. However, most of this excess was driven by a small number of projects that generated the largest volumes of credits (top right in Fig. 2b). After

excluding the 9 biggest issuers, over-crediting among the remaining projects was much lower, around 4.0 times, though still substantial. This pattern highlights the need for methodological advances that prevent egregious cases of over-crediting that disproportionately shape the overall integrity of REDD+."

7. Although the structure behind the discussion of the mechanisms has remained largely unchanged the text has been aligned with the new framing of the paper:

L367-370: "So why might over-crediting have occurred? One possible mechanism is that projects were situated in areas at low risk of deforestation, whereas reference areas faced greater threats^{7,43}. Indeed, there is good evidence that site-based and policy interventions tend to target areas of disproportionately low risk^{44,45}."

And: L374-377: "A second, potentially complementary mechanism arises from the flexibility in model choices available to projects for making ex ante predictions of counterfactual deforestation. Such prediction is inherently uncertain, especially when involving spatial or non-linear components^{46,47}."

8. We have made the relevance of our findings to JREDD+ clearer:

L417-436: "The first-generation REDD+ methodologies evaluated here are now being replaced by new jurisdictional approaches (JREDD+) for voluntary and compliance markets. JREDD+ methodologies (specifically VM0048 and ART TREES) still make ex ante counterfactual predictions but these are now based on the mean annual deforestation rate across the jurisdiction during a 5-6 year reference period^{54,55}. By eliminating project-selected reference areas, a major source of bias has been removed. Crucially, the analysis is performed by independent data providers with no direct financial stake in the number of credits issued, which is also a substantial improvement. However, reliance on ex ante predictions still carries risk: it assumes deforestation drivers remain constant between the historic and project periods, which is rarely true... One final consideration is that VM0048 permits nested projects with baselines produced using ex ante spatial modelling, which we have shown to be problematic."

9. Finally, we have structured our conclusory paragraph around the need to address methodological limitations if integrity in REDD+ is to be restored:

L454-465: While the first-generation of REDD+ projects achieved far less than claimed, many nonetheless reduced deforestation. Given the modest progress towards deploying engineered carbon capture and storage, there remains a role for forest-derived carbon credits on the path to net zero. The challenge is to ensure that claimed impacts reflect real outcomes. Removing methodological flexibility, particularly the use of project-selected reference areas and ex ante modelling, is a crucial first step, but embedding quasi-experimental evaluations would ensure issued credits represent truly additional reductions in deforestation. Either way, far fewer credits would be issued to projects using these approaches, meaning prices will need to rise to pay for the genuine cost of equitable projects with real climate benefits⁶⁴. Bridging the forest-finance gap will require abandoning the illusion of cheap carbon and paying the true cost of credible mitigation.

Together we believe these 9 changes make the manuscript much stronger and more clearly demonstrate its novel contribution.

Reviewer 2 (Remarks to the Author):

Thank you again for the opportunity to read this paper in its 2nd revised version. I think it is an excellent paper that has benefited immensely from the comments of all reviewers. I expect that the paper will have a significant impact on the readers of Nature Communications and be widely cited. First, I am also glad that my comments led the authors to revise their database, based on their complementary studies, and identified errors in the data.

Comment 1

Comment 1 round 3:

I remain sceptical about the contribution made here beyond what has already been shown in other publications by the authors (First comment of the first review). Nonetheless, I fully support this publication, but must insist that the authors should clearly state their scientific contribution.

In this context, there is a misunderstanding of my previous comment that followed up on comment 1 of the first review:

"The introduction should have a paragraph highlighting the scientific contribution: "We contribute to the literature on"

In response, the authors added this paragraph:

"Our study contributes to the literature examining how to infer causal effects from observational data^{27,28}, especially in conservation where randomised control trials are rare or impossible^{19,20}. Our results show clearly that despite variation in estimates from quasi-experimental methods, there is consistent evidence of overcrediting. This result cannot be explained by differences between the remote sensing layers used by quasi-experimental and certified assessments. Instead, we find that over-crediting was primarily driven by the selection of control areas with greater exposure to deforestation than projects and through unrealistic modelling of future deforestation rates."

What I expected was a paragraph that contextualizes the paper within a) the (forest) conservation research in general and b) the REDD+ crediting research in particular – explaining what this paper contributes (methodological, theoretical, or conceptually, ...) to our knowledge beyond what was shown in West et al. (2020), West et al. (2024), Guizar-Coutiño et al. (2022), Guizar-Coutiño et al. (2024), and Balmford et al. (2023).

Response 1 round 3:

This point is obviously extremely important. We have substantially reworked the title, abstract, introduction and discussion to make the contribution of the paper much clearer. We have de-emphasized the evidence for overcrediting as this is indeed, as pointed out by Reviewer 1, now widely accepted in the academic literature. Instead, we show that many

projects have effectively slowed deforestation despite over-crediting. We therefore focus on the mechanisms that resulted in over-crediting and the lessons that can be learned to inform the development of better methodologies for issuing credits. We argue that over-crediting must be resolved if carbon credits are to contribute to closing the forest finance gap.

To properly situate our research within the conservation literature, we have added the following text at the beginning of the second paragraph of the introduction:

L60-62: “A major challenge in measuring the impacts of forest conservation projects is estimating counterfactual outcomes (what would have happened to deforestation in the absence of a project)¹⁴⁻¹⁸.”

We have also added a new paragraph to the introduction that sets out the need for our research within the existing REDD+ literature. We highlight that, although the evidence of over-crediting is clear, the reasons why it occurred are largely hypothetical, which creates a clear need to understand why it happened:

L72-83: “The prominence of REDD+ has drawn scrutiny from several independent research groups that have applied these quasi-experimental methods to evaluate the impact of first-generation REDD+ projects on deforestation^{7-9,23-25}. Consistently, these studies have found evidence of over-crediting. Proposed explanations include the use of inappropriate reference areas, unrealistic ex ante modelling that exaggerated expected deforestation, and selective use of certification methodologies that inflated credit issuance^{7,9,11,23}. However, these hypothesised mechanisms have not been systematically tested¹⁰. Some industry insiders have rejected the evidence for over-crediting altogether^{26,27}, arguing that (1) quasi-experimental estimates of avoided deforestation are inconsistent across studies due to methodological differences²⁸, and (2) over-crediting is an artefact of using of publicly available global forest cover datasets that detect less deforestation than the bespoke remote sensing classifications used by projects²⁹.”

A full summary of the changes made to reframe our paper is presented in our response to Reviewer 1 above.

Reviewer #3 (Remarks to the Author):

There is no reviewer 3

Reviewer #4 (Remarks to the Author):

While the article has improved, I still have non-negligible concerns regarding the writing, and data, specifically pertaining to the differences in the bespoke and remote sensing datasets. More details are below.

Comment 1

Comment 1 round 3:

The contributions of this article are still not clearly stated in the Intro (lines 125-132). I think the text in the response documents was better.

Response 1 round 3:

We agree with the reviewer that our previous revision still did not properly state the novel contributions of study.

We have reworked the manuscript very substantially to make our novel contribution much clearer, including a reworked title, abstract, introduction and discussion. A full summary of the changes is presented in our response to Reviewer 2 above. We are grateful to all three reviewers for pushing us to make these changes and believe they make the manuscript much stronger as a result.

Comment 2

Comment 2 round 3:

It should be clarified in the text that the implicit underlying assumption is that markets (and property rights) are localized, so that the leakage would also be local (line 581, 595-597).

Response 2 round 3:

We agree with the reviewer that excluding local areas is necessary if we assume that local leakage is important. For that reason we have added the following discussion text:

L412-416: "Finally, REDD+ projects can cause spillover effects (leakage) that might influence deforestation rates in control units. While some studies have attempted to overcome spillovers driven by localised changes in markets by only matching to areas beyond a certain distance from project boundaries^{23,24}, this strategy may be insufficient because of the complex and unpredictable manner in which spillovers can occur⁵³."

As well as this text in the Methods:

L592-594: "This is important because, under the assumption that leakage arises through the reorganisation of local supply chains or markets, there would be concentrated interference at short distances from projects."

Comment 3

Comment 3 round 3:

The manuscript claims that because the data by Vancutsem et al (2021) detected higher forest loss inside project areas, it must also detect higher rates in control areas (line 355-358). I am not sure if this is true: If the forests inside the control polygons are allocated differently (e.g., more scattered or different tree species), it may not be as easy to forest loss. Some more support is needed to address this claim.

Response 3 round 3:

While we do not think it likely that classification errors are likely to differ between projects and reference areas, due to their proximity and supposed similarity, we have nonetheless added the following text to the discussion:

L364-366: "...classification errors are unlikely to differ systematically between project and reference areas unless they differ markedly in terms of species composition or fragmentation."

Comment 4

Comment 4 round 3:

How many projects had negative quasi-experimental estimates? (line 653).

Response 4 round 3:

These are the eight projects shown in figure 2. This is now clarified in the text, with a reference to the figure:

L139-141: "Eight projects (shown in red in Figure 2) experienced more deforestation than their quasi-experimental controls, although only in one case was this significant."

Comment 5

Comment round 3:

Include in the text whether the forest loss differences vanished if temporary forest loss ("degraded category) and regrowth forest were included (line 695). This statement is currently only in S4.

Response round 3:

In the discussion, we now say:

L360-363: "Expanding the forest cover definition to include degraded forest and counting only long-term transitions to non-forest produced lower deforestation rates (see S4 for comparisons including different forest cover definitions)."

Comment 6

Comment round 3:

Insert in the text some potential explanation for why the Madagascar projects resulted in more plausible carbon credits (lines 335-340)

Response round 3:

In our updated analysis incorporating the results of Tang et al. we find that the under-crediting projects are now from Peru as well as Madagascar. Therefore, we no longer specifically mention the Madagascar projects in the discussion.

Comment 7

Comment round 3:

I would use “forest loss” and not deforestation (e.g., line 354) as it is not clear why forest was lost (some forest loss is not deforestation).

Response round 3:

While we appreciate the subtlety of the reviewer’s comment, we have opted to stay with the original wording, which is consistent with the use of the word deforestation throughout. Deforestation and forest loss are arguably interchangeable terms: both confer a transition from forest to non-forest but with no certainty about the durability of the transition. However, we have added the term “(loss of undisturbed forest)” in the results section where we start assessing the mechanisms under question two. In the methods we also now make the definition clear:

L534-537: “The Guizar-Coutiño and PACT studies considered deforestation to be degradation (disturbances lasting <2.5 years) or deforestation (disturbances lasting >2.5 years) of the undisturbed class, even if regrowth subsequently occurred.”

Comment 8

Comment round 3:

In my experience, the Discussion section focuses on interpreting the results. I am not used to seeing results (e.g., figures) referenced there (e.g., as in lines 340-370)

Response round 3:

We accept the reviewers position that this is not typical and have removed the references to figures from the discussion.

Comment 9

Comment round 3:

The main text should briefly mention what alternative methods are available (line 380)

Response round 3:

We agree with the reviewer that the original text was not entirely clear. Therefore we have revised the text to:

L381-385: “For most projects, the methodology used for credit issuance significantly overestimated avoided deforestation, but alternative or differently parameterised certification methodologies were available which could have produced more credible estimates. For example, across four projects, VM0006 produced estimates not significantly different from quasi-experimental results (S6).”

Comment 10

Comment round 3:

Lines 381-384 contain significant claims that need to be substantiated with citations (e.g., is there actual evidence of “perverse incentives” and of “financial interests”? Also, whose—Verra’s?

Response round 3:

Project proponents benefit financially from the credits produced and they often lead methodology selection and the resulting development of design and monitoring documents. This is clearly stated on Verra’s website. Because Verra charges \$0.05 on each credit issued, they also benefit when more credits are issued (see here for a list of fees). We now reference both of these weblinks:

L385-388: “We conclude that methodological flexibility for project proponents and certification bodies, both of whom have financial incentives to produce greater numbers of credits^{48,49}, provided an additional mechanism contributing to over-crediting in first generation REDD+ projects”

Comment 11

Comment 11 round 3:

The credibility of quasi-experimental designs depends, first and foremost, on a rigorous theory of change that takes into consideration the local context. This point is acknowledged in lines 394-400. Ex post robustness checks and sensitivity analyses need to assess the credibility of quasi-experimental research designs. Nowadays there are also better methods than matching (e.g., the explosion of panel data estimators, some of which can be applied to small samples (see ref 26 in the text). These points should be clarified in the discussion in lines 385-412.

Response 11 round 3:

We agree with the reviewer that we still had not properly described the advances in quasi-experimental methods. We have now added a description of these approaches under the limitations in the discussion:

392- 395: “Quasi-experimental methods are advancing all the time and new methods exist which combine features of difference-in-differences and synthetic control, such as interactive fixed-effects, augmented synthetic controls, and synthetic difference-in-differences^{35,50,51}.”

Comment 12

Comment 12 round 3:

I am not sure how the 83% prediction of variance (line 391) supports robustness to unobserved confounders. Isn’t a 17% margin of error large?

Response 12 round 3:

We no longer make this point directly in our revised MS. Nevertheless we agree that 17% unexplained variance is still meaningful at the project-level. The point we were making is not that prediction of 83% of the variance means that our results are robust to unobserved confounders. However, it is a measure of the effect of unobserved confounders and random error, which would act to reduce the ratio of the explained variance to the total variance. We suggest that this is reassuring because it shows we are explaining a very large amount of the variance in deforestation over a 10 year period, which is pertinent to counterfactual estimation in REDD+. The new text now reads:

L395-397: "...simulated landscapes where deforestation rates are known have been used to demonstrate the reliability of the approaches used in the studies we synthesize^{52,53}."

Comment 13

Comment 13 round 3:

Given that potentially better statistical methods exist, why are they not used? (lines 442-444)

Response 13 round 3:

While we had acknowledged that quasi-experimental methods were perhaps not being adopted due to the uncertainties associated with the numbers of credits we have expanded this text to explain that these methods are being adopted but there are still some major barriers in REDD+:

L447-452: "These approaches are now being adopted by newer certification methodologies for afforestation, restoration and revegetation (ARR)⁶⁴ and improved forest management (IFM)⁶⁵. Uncertainties in credit yields due to the unpredictable nature of the counterfactual, along with the implications of over-crediting for existing projects would seem to be reasons such ex post approaches have not yet been applied but have a key role to play in ensuring the future integrity of forest carbon credits³⁷."

Comment 14

Comment 14 round 3:

What "multiple model specifications" (line 504-505)?

Response 14 round 3:

We now make clear what multiple model specifications means:

L511-514: "[Guizar-Coutiño et al.²⁵]... assess the average treatment effect of the treated using multiple model specifications with different algorithms, parameters and covariates. The avoided deforestation estimates used in our study correspond to the doubly robust model specification reported in this new study."

Comment 15

Comment 15 round 3:

I am not sure what “propensity score subclassification” (line 503) is. At the very least, a citation is needed.

Response 15 round 3:

The propensity score subclassification only pertains to Guizar-Coutino et al. (2025) where it is explained in detail. As below (comment 16), we have revised the text to make it clear. We also now point to the original reference to Imbens and Ruben (2015):

L507-511: “Guizar-Coutiño et al.²⁵ assess the causal impact of REDD+ projects on deforestation by adjusting for observed confounders and pre-treatment deforestation trends, using a two-stage matching approach, commencing with propensity score matching via random forests, followed by propensity score subclassification on the matched datasets⁶⁴.”

Comment 16

Comment 16 round 3:

Was any sensitivity analysis done for more than just the study by Guizar-Coutino et al. (2025) (line 397)? The text mentions the sensmakr framework (line 505) but no description appears in the results.

Response 16 round 3:

The sensmakr framework only applies to Guizar-Coutino et al. (2025). We have substantially edited this section to make the different methods used by the studies synthesized in this study clearer. We have different paragraphs covering the studies included, and the methods for Guizar-Coutino et al. (2025) and then PACT, as follows:

L500-519: “We compiled data from six quasi-experimental analyses which estimated ex post counterfactual and project outcomes for REDD+ projects (Table 1). We included work published by West et al.^{7,9} (two analyses), Guizar-Couniño et al.⁸ and Tang et al.²³.

We also included results from a study under review by Guizar-Coutiño et al.²⁵, which implemented a novel framework to reanalyse the set of projects examined in Guizar-Coutiño et al.⁸ study. In this new study, 7.1 ha circular sample regions were used instead of 0.09 ha pixels to characterise forest loss observations. Guizar-Coutiño et al.²⁵ assess the causal impact of REDD+ projects on deforestation by adjusting for observed confounders and pre-treatment deforestation trends, using a two-stage matching approach, commencing with propensity score matching via random forests, followed by propensity score subclassification on the matched datasets⁶⁷. They assess the average treatment effect of the treated using multiple model specifications, including different algorithms, parameters and covariates. The avoided deforestation estimates used in our study correspond to the doubly robust model specification reported in this new study.

Finally, we used the Permanent Additional Carbon Tonne (PACT) v2 method²⁴ (described below) to produce additional estimates and to explain the reasons for over-crediting. In each case, we only included projects for which certified estimates were also available.”

Comment 17

Comment 17 round 3:

The horizontal bar with two stars on Fig. 3b is confusing.

Response 17 round 3:

We have removed the horizontal bar.

Comment 18

Comment 18 round 3:

Issues related to grammar and readability (not a comprehensive list):

Response 18 round 3:

We thank the referee for identifying these errors. We have resolved the specific issues highlighted by the reviewer in the specific comments and responses immediately below and have taken care to improve grammar and readability throughout.

- Lines 645-646 as worded are hard to understand.

L658-660: "We also evaluated how many projects had avoided deforestation estimates that were significantly greater than zero, using the lower bound of the 95% confidence intervals across quasi-experimental studies."

- Lines 653-655 are likely missing a verb.

"The mean over-crediting ratio was calculated for all projects that had issued credits and also had positive mean quasi-experimental estimates."

- Line 663 is also confusing: What deforestation rates?

L665-667: "We calculated annual deforestation rates to make the quantities of deforestation measured by the certified assessments and the ACC layer (e.g. those produced by PACT) directly comparable."

- Active voice is better than passive voice (e.g., lines 715-720), but, at least, it did not obfuscate the meaning of sentences.

L717-719: "We compared the exposure to deforestation risks in the control, reference and project areas to explore how far the selection of reference areas accounted for differences between certified and quasi-experimental approaches."

- The readability sentences like the ones on lines 346-349 and 276-279 can be improved (unnecessarily wordy)

The first sentence has been simplified to: L355-357: *“However, we found no evidence that the widely used ACC layer systematically detected less deforestation. In fact, it detected similar or higher rates in project areas than certified estimates using bespoke layers”*

And the second sentence has been written in two parts as: L271-280: *“To dissect the relative contributions of the mechanisms identified in preceding analyses, we examined their effects in the subset of projects (n = 17) for which all required data were available... We then sequentially re-estimated avoided deforestation using different combinations of the deforestation rates drawn from the certified and quasi-experimental assessments to isolate the effect of each mechanism”*

- Awkward phrasing (line 355-357 and 461-462): For example, it is unclear what “this” refers to.

The first sentence has now been removed. The second sentence has been changed to: L458-461: *“Removing methodological flexibility, particularly the use of project-selected reference areas and ex ante modelling, is a crucial first step, but embedding quasi-experimental evaluations would ensure issued credits represent truly additional reductions in deforestation.”*

- Grammar issue in lines 474-476 (“however” is a conjunctive adverb that requires different punctuation)

L477-479: *“Certified project documents did not report deforestation consistently, however the ex post observed deforestation within the project and the ex ante modelled deforestation under the counterfactual scenario were available.”*

- Some sentences contain unnecessary repetitions: For example, “Replace” is used twice in the same sentence (line 413-415)

L418-419: *“The first-generation REDD+ methodologies evaluated here are now being replaced by new jurisdictional approaches (JREDD+) for voluntary and compliance markets.”*

Comment 19

Comment 19 round 3:

How was the bootstrapping done (lines 658, 162)? What is being resampled?

Response 19 round 3:

The bootstrapping was done by resampling the projects with replacement 10,000 times. This is now stated as follows:

L672-674: *“Projects were randomly sampled with replacement 10,000 times, with the results then used to derive mean values and 95% confidence intervals.”*

Comment 20

Comment 20 round 3:

The first 4 bars in Fig. 5 are overlapping. The dark blue, the orange, and the red bars also overlap. As displayed, they suggest that the differences are not statistically significant. For this reason, I am not convinced by the decomposition of the differences in the quasi-experimental vs. ex ante models (lines 281-301). The horizontal bar with stars also seems out of place.

Response 20 round 3:

We agree with the reviewer that the lack of significant effects in Fig. 5 makes the relative contribution of the different mechanisms unclear. The analysis is a synthesis from the smaller set of projects for which we have all the data we need, and as a result it lacks power. Nevertheless, the effects it suggests are matched by those in other analyses in the MS with larger sample sizes, where the effects are significant. In fig. 3b we show that more deforestation is measured using the ACC; in fig. 4d we show that control areas are exposed to significantly greater threats, and deforestation is elevated but not significantly (again the same size is small; $n = 17$); and in S6 we show that methodologies are selected which issue significantly more credits than expected. Figure 5 is not about re-presenting these findings but rather about synthesising the various lines of reasoning (which we can only do with a much smaller sample of projects) to dissect the approximate share of over-crediting associated with each; we believe this is an important contribution to make. That said, we have modified the text to recognise the lack of significance and the uncertainty that remains:

L271-275: "To dissect the relative contributions of the mechanisms identified in preceding analyses, we examined their effects in the subset of projects ($n = 17$) for which all required data were available. The smaller sample size meant that most effects within this subset were not statistically significant, but they were of a similar magnitude to those observed in the larger analyses of each mechanism."

Comment 21

Comment 21 round 3:

What do the bespoke layers consider as forest loss? Do logging roads and log landing pads count? It is still not clear to me from the text to what extent the bespoke layers and the remote sensing datasets are consistent in how they quantify forest loss.

Response 21 round 3:

We appreciate that this was still not clear from the main text alone. For that reason we have added the following statement at the beginning of the subsection on *Certified estimates of avoided deforestation* within the Methods:

L470-473: "These include the loss of all areas that had been forest for at least 10 years prior to the establishment of the project. Forest cover was classified using cloud free satellite images and bespoke remote sensing approaches (see S1 for details)."

S1 has more details on how certification methods measure deforestation and we hope the reference to it is helpful.

For further clarification, there is no distinction in how bespoke approaches treat deforestation based on its end use. So roads and logging pads are treated the same as deforestation for agricultural purposes.

Comment 22

Comment 22 round 3:

The ACC layer (Vancutsem et al. 2021) uses “degraded” as a label for forest that is temporarily lost (i.e., less than 2.5 years). By excluding this short-term forest loss, the analysis predictably underestimates the forest loss calculated from remote sensing data. However, S4 mentions that including the “degraded” forest decreased the rates of forest loss. This does not make sense to me: If only the numerator is increasing (e.g., #cells with forest loss)/total forest cells in a unit, the ratio should increase.

Response 22 round 3:

We suspect our original text has introduced a misunderstanding about how our analysis was implemented. In the ACC layer, there are 4 forest classes:

Class 1: undisturbed

Class 2: degraded

Class 3: deforested

Class 4: regrowth

In the main text we consider the transitions that correspond to the most meaningful loss of above ground biomass as demonstrated by Holcomb *et al.* (2024) and by us in S11: these are the transitions 1->2, 1->3 and 1->4. Therefore the denominator is 1_{t_0} and the numerator is $(1_{t_0} \cap 2_{t+i}) + (1_{t_0} \cap 3_{t+i}) + (1_{t_0} \cap 4_{t+i})$. In the supplementary information, we test sensitivity to inclusion of degradation in the forest cover definition but then only consider forest loss to be longer-term (>2.5 year) transitions from the undisturbed and degraded classes. Although transitions 2->3 are included this way, transitions 1->2, which are more frequent in our analysis, are excluded. At the same time the denominator also increases from 1_{t_0} to $(1 \cup 2)_{t_0}$. The overall result, as shown in S4 is that deforestation rates are decreased.

Of course, neither approach properly accounts for all the pertinent transitions 1->2, 1->3 and 2->3 as well as the regrowth transitions. To do this properly requires handling degradation, deforestation and regrowth rates separately as shown in Guizar-Coutiño *et al.* (2022; 2024). If we were to add the deforestation of degraded forest to the deforestation rates presented in the main text they would be higher still.

We have now clarified the transitions we consider as deforestation in the Methods section of the main text:

L534-543: “The Guizar-Coutiño and PACT studies considered deforestation to be degradation (disturbances lasting <2.5 years) or deforestation (disturbances lasting >2.5 years) of the undisturbed class, even if regrowth subsequently occurred. Degradation was included because it has been shown to result in biomass reduction of >50% within 12

months. Our analysis of above ground biomass densities for different ACC cover classes shows that both degraded and regrowth classes had <50% of the above ground biomass of the undisturbed class (see S11). Therefore, for the mechanistic component of our study, we followed the PACT methods by defining forest cover solely as the undisturbed class within the ACC dataset. We explore the differences that arise using different definitions of forest cover in S4 and S7.”

And in the results:

L195-198: “When we ran the analysis to include degraded forest in the ACC-measured forest area at project start, so that deforestation was only measured as long-term (>2.5 years) transitions to non-forest, deforestation rates were lower and not significantly different from certified values (S4).”

We have also added further clarity in S4 of the Supplementary Information:

“Changing the forest cover definition when using the ACC data to include the undisturbed and degraded forest classes reduced the ex post deforestation rates measured in project areas from 0.26%/year reported in figure 3 of the main text to 0.09%/year. The lower ACC deforestation rates result in there being no difference from the certified project deforestation rates (Wilcoxon paired signed-ranks, $V = 248$, $p = 0.1866$). This change in definition means that deforestation is considered to occur when there is a transition from either the undisturbed or degraded class to the deforested or regrowth classes. Including transitions to the regrowth class is probably not appropriate because the mean carbon density of the regrowth class is similar to that of the degraded class... If we only consider changes to the deforested class, and exclude transitions to the regrowth class, the ACC deforestation rate is even lower.”

To make this distinction clear, in the discussion we now say:

L360-363: “Expanding the forest cover definition to include degraded forest and counting only long-term transitions to non-forest produced lower deforestation rates (see S4 for comparisons including different forest cover definitions).”

Comment 23

Comment 23 round 3:

Further, the study focuses on the loss of “undisturbed” forest and does not allow for forest regrowth or the loss of forest that has been previously restored (lines 573-579, line 619, 672). It is not clear from the write-up if the exclusion of regrowth is another reason there are differences between the remote sensing and bespoke datasets (e.g., lines 350-354). S4 mentions that including regrowth lowers the deforestation rates. However, I am not clear if S4 also includes the loss of forest that has been previously regrown?

Response 23 round 3:

Apologies that this was still not clear. We have tried to make our description of the deforestation analysis in the main text and supplementary information clearer as described in our response to comment 22. In addition we have also added the following text to highlight that bespoke datasets may miss short-term disturbances and regrowth:

L357-360: "This may have occurred because the ACC is a multi-temporal approach that considers all disturbance events occurring since the start of the time series⁵ whereas bespoke approaches typically detect non-forest pixels at two points in time, which may miss short-term disturbances and regrowth events⁴¹."

In our analysis deforestation events could include forest that had previously regrown but only if they were undisturbed forest at the beginning of the project. Equally, in bespoke assessments, deforestation events could include forest that had previously regrown but only if they were classified as forest at the beginning of the project.

Comment 24

Comment 24 round 3:

The manuscript argues that using a 2019 layer for the travel time to health centers is an appropriate proxy for economic pressure/"market exposure" (lines 567-571). I am still not convinced: First, whether the variable proxies for market access depends on the drivers of deforestation in a given location: Commercial ports may not have a health center; timber is often transported down a river, where there may not be health centers. Logging roads may or may not be accessible for motorized vehicles throughout the year. Second, the endogeneity concern is significant: It is possible that roads and health centers appeared after deforestation took place or after the projects were implemented. Land cover, one of the components in the calculation, is also endogenous. For this reason, it is not clear to me why the distances to ports and rivers (both readily available) are not used, at least, to supplement the list of covariates. At minimum, some sensitivity analysis is needed for the endogenous travel time covariate.

Response 24 round 3:

The issue here is that we need travel times that are appropriate proxies for economic pressure. Euclidean distance is not necessarily a good proxy for travel time because there may be significant barriers that prevent travel. For example, steep topography or a lack of roads between the focal pixel and the target end point. To the best of our knowledge, travel times are only available to health clinics. This has obvious limitations, which we now set out:

L404-410: "A potentially important hidden confounder in our analysis is the economic value of land, which we proxied using accessibility (travel time to healthcare in 2019⁵⁴). This covariate has weaknesses that could result in the selection of inappropriate controls. While healthcare is often synonymous with certain size population centres, these places are not necessarily the same as processing or transportation hubs for timber or agriculture. Equally, most projects pre-date 2019, meaning transportation patterns are partly determined by project activities."

We agree with the reviewer that it would be better to include travel time to commercial ports or processing facilities. Accounting for seasonal variation due to changes in road or river

conditions would be a further improvement. And a time series of travel times that could be used to match locations based upon their pre-project accessibility, would be better still. We are aware of work underway to produce these layers as they are clearly important but, to the best of our knowledge, they are not yet ready for incorporating in our analysis.

We disagree that changes in pre-project land cover are endogenous, because they occur prior to project implementation. There may be situations where pre-project land cover changes reflect anticipation of project establishment, just as we describe for jurisdictional REDD+, but this should be minimised by considering a 10 year period and is an even more significant issue for synthetic control approaches that predominately match on historic changes in forest cover. For this reason we contend that trends in pre-project land cover are a useful proxy for economic pressure, especially in the absence of better data layers on accessibility.

This is not to say that matching studies should ignore these issues, but we feel the additional value of adding Euclidean distance to ports or rivers is highly unlikely to undermine our results and would be a substantial computational endeavour: we would need to re-match all projects, requiring approximately 50 days of processing time on our high performance computer (250 cores and 1TB memory). For this reason we haven't included these additional sensitivity analyses.

Response to reviewers

NCOMMS-24-45991-T: Learning lessons from over-crediting to ensure additionality in forest carbon credits

We would like to thank the reviewers for their continued attention to detail which has ensured our analysis is robust and accurately presented. Below we provide a detailed, point by point response (no highlight) to the reviewers' comments (highlighted in purple).

Reviewer #1 (Remarks to the Author):

Comment 1

I appreciate the effort that you made to reframe your results. My own opinion is that this dissection of sources of error is more interesting for a field journal than for a general interest outlet like Nature Communications. That said, assessments of fit are vague and subjective, and so I leave this up to the editors of this journal to decide if they feel that is the case.

We are very happy that the reviewer is satisfied with the changes we have made.

Reviewer #2 (Remarks to the Author):

There is no reviewer 2

Reviewer #3 (Remarks to the Author):

There is no reviewer 3

Reviewer #4 (Remarks to the Author):

The manuscript has improved substantially in terms of writing. Two remaining issues:

Comment 1

It is still not clear to what extent the bespoke and ACC layers are comparable (this is an issue I have been raising for a while now). The bespoke layers seem to account for the loss of regrown forests provided they had been in place for at least 10 years (lines 471-472). In contrast, the ACC layers used in the quasi-experimental designs exclude reforestation. Further, the ACC layers classify forest disturbance, regrowth, or loss based on the preceding 3 years of data (Vancutsem et al. 2021). They use a different methodology for the last three years of data. The ACCs layers do not have good data prior to 2000 for Africa and may not do a good job excluding tree plantations for some locations (see the description of the caveats in the Vancutsem et al. 2021 paper). The manuscripts reports that ACC data from 1990 to 2021 (last year of available data in ACC, I think) were used (lines 561-563).

To improve the paper, I think the loss of long-term reforestation (>10 years) should be included in the deforestation calculations to match the bespoke layers. The Vancutsem et al. paper reports estimates for long-term reforestation (10+ years) in their paper, so the data by the same authors exist.

We appreciate the reviewer's concern that there are fundamental differences between the ACC and the bespoke layers used in certification. Ultimately, these are two different approaches to tracking emissions from deforestation and both contain errors and idiosyncrasies. The main reason for including a direct comparison between the layers was to assess whether the ACC detected less deforestation than the bespoke layers, which could explain at least part of the over-crediting effect detected. Our results show this is not the case.

We agree with the reviewer that, according to Verra's eligibility criteria, forest older than 10 years should be included in our analysis. This is particularly important because disturbed forests and plantations may be expected to exhibit higher deforestation rates than undisturbed forests, with effects on additionality. Therefore, we reanalysed our data to test the effect of including regrowth pixels and found such pixels were extremely rare, occurring at a median rate of only 0.07% (max = 0.8%) at the start of projects. Consequently, including regrowth had a negligible effect on our results. We have nevertheless included this new analysis in the supplementary information. In S4 we have added a new analysis that presents the effect of measuring deforestation as loss of undisturbed or regrowth forest in project areas, where there is almost no effect:

Deforestation defined as loss of undisturbed or regrowth ACC classes

Measuring deforestation as transitions: (1) from the undisturbed class to any other class, or (2) from the regrowth class to the deforested class, did not change the median ex post deforestation rates measured in project areas when compared with measuring loss of undisturbed forest alone. The median ACC rate remained at 0.26% a year, as presented in figure 3.

In S7 we also present an equivalent analysis in control areas:

Measuring deforestation as transitions: (1) from undisturbed forest to any other class, or (2) from the regrowth class to the deforested class, had no effect on the median deforestation rate measured in quasi-experimental control areas, which remained at 0.41%/year as presented in figure 4. In certified control areas deforestation rates decreased from 0.69%/year in figure 4 to 0.68%/year.

In both new analyses the effects were small due to extremely few regrowth pixels at the start of projects.

We also present a comparable analysis in S4 and S7 where transitions from the degraded class to the deforested class are included in addition to transitions from the undisturbed class to any other class alone and in combination with transitions from the regrowth class to the deforested class. The degraded class was more common than the regrowth class but was still rare, comprising less than 3% of pixels at the start of projects. Consequently, deforestation rates also changed very little.

Despite the small changes in the projects and control areas, we also reanalysed the avoided deforestation rates from figure 5, which now appears in a new supplementary information section, S12:

The sensitivity of avoided deforestation rates to the ACC transitions considered

Measuring deforestation as transitions: (1) from the undisturbed class to any other class, or (2) from the regrowth class to the deforested class, had very little effect on the avoided deforestation rates used for isolating the effects of forest cover layer and reference area selection and ex ante modelling.

The avoided deforestation rates from the main analysis (solid lines) are compared with avoided deforestation rates that also include transitions from the regrowth class to the deforested class (dashed lines). There is no bar for deforestation rates including regrowth for certified control - certified project, as this does not make use of the ACC layers.

In the discussion we now say:

Lines 365-368: Including forest classified as regrowth at the project start had almost no effect because this class was rare within projects. The effect of including forest classified as degraded or regrowth by the ACC in deforestation calculations is presented in the supplementary information (S4, S7 and S12).

The data/methods section needs to insert more details about the time frames of the ACC data used as well as factors like tree plantations or projects in Africa could be a reason for the difference in the two data sources.

We agree that adding more detail on the nature of the ACC could be helpful for understanding the differences between the data sources. Therefore we have added the following text to the methods section:

Lines 545- 552: The ACC dataset classifies each 30 m Landsat pixel within the humid tropics as either undisturbed forest, water or non-forest in 1990. Then disturbance events are classified according to their duration, as either degradation (<2.5 years) or deforestation (>2.5 years). If forests subsequently recover, after a period of at least 3 years, this is classified as regrowth. Some disturbed forests and plantations are erroneously classified as undisturbed forest due to a lack of availability of a longer time series of observations and because their spectral characteristics are similar to those of natural forests. This is particularly problematic in Africa due to limited Landsat coverage before 2000.

Comment 2

While the text has been substantially improved, I am still struggling to understand some paragraphs.

We are glad the reviewer thinks there has been a substantial improvement in the text. We are sorry they still considered some sections were unclear at the last revision. We have made further edits recommended by this reviewer.

What is the difference between the over-crediting ratio (lines 663-668) and the global over-crediting ratio (lines 670-675)? Just the exclusion of the negative quasi-experimental estimates?

The over-crediting ratio is calculated for each project. From these we subsequently calculate the mean and global over-crediting ratios. We have added further clarification in this version:

Lines 682-693: We next assessed the over-crediting as the ratio of the certified to the mean quasi-experimental estimate of avoided deforestation. We refer to this as the over-crediting ratio, which expresses how many times more credits were issued through certification than suggested by quasi-experimental estimates. The mean over-crediting ratio was calculated as the mean of the over-crediting ratios of all projects that had issued credits and also had positive mean quasi-experimental estimates; projects with negative mean quasi-experimental estimates were excluded. The global over-crediting ratio was calculated by dividing the summed certified avoided deforestation by the summed mean quasi-experimental estimates (including negative values). The mean and global over-crediting ratios represent how many certified avoided deforestation units correspond to one quasi-experimentally validated unit for a project selected at random, and across the REDD+ portfolio as a whole, respectively.

Given that there are 17 projects with good enough data for comparisons, I am not following why “Projects were randomly sampled with replacement 10,000 times” (lines 673-675)

We have clarified our reasoning here. The data are non-normal which makes the standard error a poor estimate of the variation. We have made this clear in the revised text:

Lines 693-695: Because the data were non-normal, non-parametric 95% confidence intervals were produced for the mean and global over-crediting ratios by randomly sampling projects with replacement 10,000 times.

I would insert the original sample size in line 616

The sample density is described in the text above, so we have added the sample sizes there:

Lines 620-621: This resulted in a mean of 17,397 units across the projects (min = 687, max = 44,697)

Line 525 claims the annual avoided deforestation is just the cumulative estimate/#years whereas the methods describe compounding (lines 680-695)

Yes this is true and we appreciate the reviewer's attention to detail. Annual deforestation in hectares was calculated without compounding and presented in Fig. 2. However, for the remaining analyses where deforestation rates are compared we used compounding. For clarity we have revised the first statement so that it says:

*Lines 533-535: Annual avoided deforestation ~~rates were~~ **in hectares was** calculated by dividing cumulative avoided deforestation by the total number of years of the evaluation, to compare estimates produced over different time periods.*

And the second statement:

*Lines 700-702: We calculated **percentage** annual deforestation rates to make the quantities of deforestation measured by the certified assessments and the ACC layer (e.g. those produced by PACT) directly comparable.*

In many disciplines “mechanisms” connotes the process through which something happens. Here you are using “mechanism” instead of “explanation”. Consider revising the text.

The title of the original article was:

“Understanding the reasons for over-crediting in REDD”.

This was changed at the second round of review in response to reviewer 1's comment:

“This manuscript analyses the mechanisms of over-crediting. I suggest reformulating the title to “Understanding the mechanisms for over-crediting in REDD+”.”

We also adopted this language throughout the manuscript. While “explanations” would be a suitable alternative, because we are trying to understand the process through which over-crediting happened, we have opted to stay with reviewer 1's original suggestion for the title and have retained the wording throughout.

Spillovers are not “unpredictable” (line 416) but rather require modeling.

We agree with the reviewer that spillovers are not necessarily unpredictable therefore we have deleted that wording:

Lines 412-421: Finally, REDD+ projects can cause spillover effects (leakage) that might influence deforestation rates in control units. While some studies have attempted to overcome spillovers driven by localised changes in markets by only matching to areas beyond a certain distance from project boundaries^{23,24}, this strategy may be insufficient because of the complex ~~and~~ ~~unpredictable~~ manner in which spillovers can occur⁵⁵.

You may want to add the need for transparency (e.g., in determining reference areas) in generating credits to lines 370-375.

We agree with the reviewer that greater transparency would be beneficial, which is a point that has been lost during the revision process. We have opted to include the point later in the discussion where we highlight the benefits of quasi-experimental approaches:

Lines 450-453: Quasi-experimental crediting would not alter the timing of credit issuance, as even methods that use ex ante counterfactuals still rely on ex post measurements within projects to quantify additionality, while greatly improving transparency⁴².

I am not sure what “pooling credits at the scheme-level” means (line 443)

We have revised the text to make it clear what we mean:

Lines 447-459: Of course, there continue to be significant uncertainties in project-level estimates, but these can be ameliorated by buying credits from multiple projects ~~pooling credits at the scheme level~~ or reducing the number of credits issued to safeguard any claims being made^{36,63}.

If possible, it will be interesting to insert a short description of how the reference areas are selected for ex-ante modeling: Is it just the exposure to the drivers of deforestation within a pre-defined cutoff (as reported in Table 1) ? Is there a reason to believe the ex ante modeling assumes a different mechanism of change than the ex post quasi-experimental analysis?

We can see that it might still be difficult for a reader new to this subject to get an overview of the certified methods. For that reason we have revised the methods section to provide this overview and point to table 1, and the supplementary information (S1), where we describe certified methods in detail:

Lines 62-65: The first-generation of REDD+ projects could choose from several certification methodologies (Table 1a), each of which used measurements of historic deforestation in expert-selected ‘reference’ areas in conjunction with ex ante modelling approaches to predict ‘baseline’ deforestation (see S1 for further details)¹¹.

Lines 477-483: Reference areas were selected so that they had similar characteristics to projects in terms of forest cover and expected drivers of deforestation (see table 1). Forest cover was classified using cloud-free satellite images and bespoke remote sensing approaches. Ex ante estimates of counterfactual deforestation were estimated using a variety of different modelling approaches according to the methodology used. Further detail on certified project monitoring and counterfactual estimation is presented in S1.

Why are there two types of shapes (diamonds and circles) in Fig. 2b?

We have now revised Fig. 2b, so that it only has circles consistent with Fig. 2a.

Was the accessibility in the reference areas within 10-20% of the treated areas as suggested by Table 1? (lines 259-266)

This is an interesting question. We find that they are and have added the following text to the results, as suggested:

Lines 251-253: In fact, the most accessible parts of reference areas were more accessible than the most accessible areas within projects (+/- 20%) 82% of the time.

Response to reviewers

NCOMMS-24-45991-T: Learning lessons from over-crediting to ensure additionality in forest carbon credits

We are delighted the reviewers are finally happy with the manuscript. We have addressed their comments as outlined below.

Reviewer #1 (Remarks to the Author):

There is no review 1

Reviewer #2 (Remarks to the Author):

There is no reviewer 2

Reviewer #3 (Remarks to the Author):

There is no reviewer 3

Reviewer #4 (Remarks to the Author):

The manuscript has improved enough to warrant publication. However, there are still issues that would need to be addressed:

Comment 1

Citations are needed when you make statements based on previous work. For example, lines 545-552 are based on Vancutsem et al. 2021 (the authors of the ACC layer); the points summarized in these lines are featured in the original paper.).

We have added the reference at the beginning of the paragraph.

"The ACC dataset⁵ classifies each 30 m Landsat pixel within the humid tropics as either undisturbed forest, water or non-forest in 1990."

Comment 2

Fine to keep "mechanism" instead of "explanation", but add a sentence to define what you mean in the text.

We have added the following text to improve clarity:

"However, these hypothesised mechanisms **through which over-crediting occurred** have not been systematically tested¹⁰."

Comment 3

3. Please proofread for statements like "In fact, the most accessible parts of reference areas were more accessible than the most accessible areas within projects (+/- 20%) 82% of the time." (lines 251-253)--apart from the needless repetitions, the sentence takes a lot of time to process.

We have rewritten this paragraph to simplify the language to aid interpretation and remove repetition.

"The same pattern was observed across all 17 projects that provided mapped reference areas (Fig. 4c, S3). Compared with project areas, reference areas were more accessible (median standardised mean difference = -0.30, $t = -2.41$, d.f. = 16, $p = 0.028$). 82% of projects used reference which had locations more accessible than the most accessible locations within projects. Similarly, reference areas were less forested than projects at their start (median standardised mean difference = -0.42, $t = -3.10$, d.f. = 16, $p = 0.007$) as well as five and ten years beforehand (median standardised mean difference = -0.48, $t = -4.04$, d.f. = 16, $p = 0.003$; and -0.48, $t = -4.28$, d.f. = 5, $p = 0.008$, respectively). Historic deforestation rates were not significantly different. In contrast, none of the quasi-experimental control areas differed from the project areas in any of the observed confounders (Fig. 4c)."

Similarly we have carefully read through the entire manuscript and made a number of minor changes throughout, as well as adding two references to substantiate our point about the continued role for forest-derived credits at the end of the discussion:

"Given the high costs and modest progress towards deploying engineered carbon capture and storage, there remains a role for forest-derived carbon credits on the path to net zero^{66,67}."

These changes can be assessed in the included version of the manuscript with tracked changes.

NCOMMS-24-45991-T

“Independent statistical approaches address overcrediting problems in REDD+”

August 2024

Summary

Thank you for the opportunity to read this paper. I very much enjoyed reading it.

This work investigates the Carbon Credit certification methods for avoiding deforestation from REDD+ (Reducing Emissions from Deforestation and Degradation) projects. Carbon credits are an essential pillar of the agenda to reduce the impacts of deforestation on carbon emissions and climate change. The author’s work highlights the flaws of current methodologies in creating reliable estimates of avoided deforestation from REDD+ projects. Crediting institutions mostly estimated high levels of avoided deforestation. In contrast, using reliable assessment methods, the paper shows that REDD+ projects had much lower levels of avoided deforestation. As a result, REDD+ projects have been the base for issuing overly high carbon credits. The authors highlight that over-crediting stems (a) from a failure to create credible counterfactual scenarios and (b) from systematically choosing methods that generate higher levels of avoided deforestation. This research is especially important as upcoming reforms aim to expand REDD+ projects and the monetary compensation to jurisdictional entities. Maintaining the credibility of the certification system will be essential to secure the future funding of these market-based forest conservation instruments.

I appreciate the analysis of the overestimation of avoided deforestation in REDD+ projects. This is an excellent depiction of the potential flaws in estimating overly high levels of avoided deforestation. My primary concerns are about the novelty of the results, the author’s empirical reasoning, and the use of the empirical terminology. If the authors pursue these improvements and others they receive elsewhere, I believe the paper will make a very valuable contribution to the literature.

Major comments:

1. **Novelty:** The database for this paper is collected from different sources. The authors combine data from the crediting institution (i.e., VERRA) and from previous publications evaluating REDD+ projects using quasi-experimental methods published by West et al. (2020), West et al. (2024), Guizar-Coutiño et al. (2022), Guizar-Coutiño et al. (2024), and Balmford et al. (2023) (henceforth WGGB). The latter, Balmford et al. (2023), is an unpublished document describing the PACT method. Based on the manuscript, it is very difficult to understand if PACT or Balmford et al. (2023) are a source of data or if the authors are referring to a methodology. Is the paper conducting new impact evaluations or just using the results from Balmford et al. (2023) or even from (cf. Swinfield et al., 2024) – a recent publication of the lead authors? If so, what

is the genuine contribution of the paper? Has the paper “only” pooled estimates and re-used the information to display the results in a more comprehensive way?¹.

The authors should highlight their essential contribution better. It seems the essence lies in the two messages i) There is over-crediting, ii) Over-crediting can stem from a) selecting dissimilar control, b) mismeasuring deforestation in control areas, and c) choosing evaluation methods that rely on strong assumptions (see comment XXX). To avoid confusion, I suggest sticking with the PACT method. First, show Figure 3, then Figure 1, and finally, Figure 4. Figure 2 should be an Appendix table. It lacks power and is merely based on 4 REDD+ projects analyzed by West et al. (2024).²

2. **Empirical terminology:** The manuscript is filled with confusing terminology. I suggest carefully reviewing the manuscript using an econometric lens. Examples are

- i. line 61: “The key problem in quantifying additionality is to identify the counterfactual outcome”. It is impossible to “identify” the counterfactual. The counterfactual is never observable. One can only *estimate* the counterfactual. By estimating the counterfactual, one can *identify* the causal effect and “infer” causality.
- ii. Similarly, all figures use the label “counterfactual” whereas it should be labeled *control group*, *control area*, *comparison group*, etc.
- iii. The methods used by WWGGB are labeled “statistical methods”. This is a wrongful description because VERRA also uses a “statistical method”. It is just that WWGGB use *quasi-experimental* methods.

3. **Empirical reasoning:**

- i In general, the paper misses a careful discussion about empirical concepts and methods. I agree that WWGGB’s and the presented quasi-experimental methods (QEM) are superior, but the authors’ advocacy is flawed. The authors argue that QEMs presented are better because they create a “better” comparison group. This judgment is based on improved covariate balance after matching. Nonetheless, the empirical assumption of QEMs (they vary in detail) is that unrealized counterfactual outcomes can be projected by observable pre-treatment characteristics (and outcomes). Unobservable characteristics might still exist that drive the selection of both treatment and outcome simultaneously. This would then lead to an estimation bias (direction unclear) of any of the quasi-experimental methods used in WWGGB. VERRA’s methods, in essence, rely on similar assumptions. That is why VERRA could always argue that they have ‘context knowledge’ - a characteristic that is not controlled for in WWGGB. Therefore, the essence of the critique of

¹Figure 1 is based on all 47 REDD+ projects as it collects data from VERRA, West et al. (2020), West et al. (2024), Guizar-Coutiño et al. (2022), Guizar-Coutiño et al. (2024), and Balmford et al. (2023) (henceforth WWGGB). Figure 2 is limited to 4 projects because West et al. (2024) as data on it. Figure 3 is restricted to 16 projects where boundaries of VERRA’s control/ reference area are available but then again uses PACT/Balmford et al. (2023) to assess the projects.

²Figure 2 shows min/max bounds instead of confidence intervals of each estimate. I wonder if VERRA’s estimates lie within the confidence intervals of the quasi-experimental estimates.

VERRA cannot hinge on a “better” covariate balance among observable characteristics.³ The argument must be made on the details of the underlying assumptions of VERRA’s methods and the quasi-experimental methods. Clearly, VERRA uses very strong assumptions: no leakage, no unobserved time-varying confounders (linear projection), no cross-sectional time-constant confounders, population as the main driver of deforestation, etc. In comparison, matching and panel regression use less stringed empirical assumptions: Selection on observable, no unit-and time-varying confounders, i.e., conditional parallel trends, etc.

- ii Furthermore, I suggest adding a few sentences on the rapidly developing quasi-experimental methods literature. WWGGB’s methods could be deemed outdated. For example, synthetic control methods are evolving, and new estimators rely on fewer assumptions while producing better covariate balances (cf. Hazlett and Xu, 2018). Nonetheless, I don’t think the authors should implement any of the newer estimators.

Minor comments

1. Figure 1b shows there ratio $\frac{\text{Avoided deforestation credited by VERRA}}{\text{Avoided deforestation using quasi-experimental methods}}$. For overestimation, the ratio is < 1 , and for underestimation, the ratio is > 1 . I suggest flipping the definition of the ratio so that values > 1 reflect overestimation.
2. Instead of using min/max values for the lines in Figure 1, I suggest using 90%/95% confidence intervals. For points based on multiple estimates, you can calculate the joint standard error and plot the joint confidence interval. This way, it would be easier to assess if the quasi-experimental methods significantly differ from VERRA’s estimates. Alternatively, I would plot all the estimates separately instead of making a joint Plot of all estimates of WWGGB or only focus on estimates from PACT to remain consistent with Figures 3 and 4.
3. Again, I suggest excluding the whole section on the “difference in variation” of the estimates (cf. Figure 2). On reason: Variance depends on the number of point estimates. WWGGB has more point estimates than VERRA, so the variance of this group tends to be lower. Further, the depiction of the probability in line 196 is a bit far-fetched.⁴ The whole thing lacks power – only 4 projects.
4. line 53: I don’t understand the point in “Second, the carbon ...”
5. line 547: The link to download the data does not work.

³It is easy to counter-argue that the list of covariates used in PACT is incomplete and does not reflect the regional socio-economic differences and deforestation pressures in, e.g., Peru (Figure 3).

⁴Also, 0.009 should be compared to 0.5 (the probability of a random selection).

References

- Balmford, Andrew, David Coomes, James Hartup et al.** 2023. “PACT Tropical Moist Forest Accreditation Methodology.” [10.33774/coe-2023-g584d-v2](https://doi.org/10.33774/coe-2023-g584d-v2).
- Guizar-Coutiño, Alejandro, David Coomes, Tom Swinfield, and Julia P G Jones.** 2024. “Sensitivity of estimates of the effectiveness of REDD+ projects to matching specifications and moving from pixels to polygons as the unit of analysis.” *bioRxiv*. [10.1101/2024.05.22.595326](https://doi.org/10.1101/2024.05.22.595326).
- Guizar-Coutiño, Alejandro, Julia P. G. Jones, Andrew Balmford, Rachel Carmenta, and David A. Coomes.** 2022. “A global evaluation of the effectiveness of voluntary REDD+ projects at reducing deforestation and degradation in the moist tropics.” *Conservation Biology* 36 (6): . [10.1111/cobi.13970](https://doi.org/10.1111/cobi.13970).
- Hazlett, Chad, and Yiqing Xu.** 2018. “Trajectory Balancing: A General Reweighting Approach to Causal Inference With Time-Series Cross-Sectional Data.” *SSRN Electronic Journal*. [10.2139/ssrn.3214231](https://doi.org/10.2139/ssrn.3214231).
- Swinfield, Tom, Siddarth Shrikanth, Joseph W. Bull, Anil Madhavapeddy, and Sophus O. S. E. zu Ermgassen.** 2024. “Nature-based credit markets at a crossroads.” *Nature Sustainability*. [10.1038/s41893-024-01403-w](https://doi.org/10.1038/s41893-024-01403-w).
- West, Thales A. P., Jan Börner, Erin O. Sills, and Andreas Kontoleon.** 2020. “Overstated carbon emission reductions from voluntary REDD+ projects in the Brazilian Amazon.” *Proceedings of the National Academy of Sciences* 117 (39): 24188–24194. [10.1073/pnas.2004334117](https://doi.org/10.1073/pnas.2004334117).
- West, Thales A.P., Barbara Bomfim, and Barbara K. Haya.** 2024. “Methodological issues with deforestation baselines compromise the integrity of carbon offsets from REDD+.” *Global Environmental Change* 87 102863. [10.1016/j.gloenvcha.2024.102863](https://doi.org/10.1016/j.gloenvcha.2024.102863).

Editorial Note: Attachment from Reviewer 1 for Version 1

Thank you for your thorough responses to my comments. I feel quite satisfied with the main methodological responses and I think the paper has been improved by restructuring. I am not quite at ease with the changes made in response to my comments on policy framing. I will present these one by one. My new comments are *italicized*.

My original comment: a. Some constellation of the authors in this paper have already established that there is an issue with over certification on forest carbon markets. **It would be useful to understand what the first result of this paper adds there that is new.**

Your original response: As the reviewer says, a series of papers indicated over-crediting but there has been substantial push-back arguing that quasi-experimental methods are unreliable because they generate too varied estimates and use remote sensing layers that detect too little deforestation. For the first time, this paper examines these claims in detail by applying several different quasi-experimental methods to a large sample of projects and finds that:

1. While quasi-experimental estimates do indeed vary, they do so far less than certified estimates
2. The global remote sensing layers used by quasi-experimental methods do not detect less deforestation than bespoke layers used in certification
3. There is evidence that certified control areas were exposed to greater threat of deforestation than projects
4. There is also evidence to suggest flexibility in *ex ante* modelling caused elevated claims of avoided deforestation
5. These issues together resulted in over-crediting, underscoring the fundamental unsuitability of current certification methods compared with quasi-experimental methods.

We have made this much clearer in our revision through the introduction of figure 1, which summarises these questions. The manuscript is then structured so that the questions are posed one-by-one and answered in the results.

My new comment: *In some sense, many of these points appear to be refinements of work that you have already published. It seems to me that a dissection of these mechanisms is a more field-relevant focus, rather than an analysis that brings us information that should be widely distributed to the readership of Nature Communications.*

My original comment: b. The main constructive recommendation of the paper is that ex post statistical approaches be used in carbon crediting. As I was thinking about this, I got confused about what is actually being done versus what is proposed. Are the authors suggesting withholding payments until assessment of avoided deforestation can occur ex post? This seems like an impossible proposition, both at the broader institutional level and at the level of producer participation in the event that projects involve individual land users foregoing deforestation, which they must do for there to be additionality. Recent research suggests that successfully implementing PES in low-income settings may require ex ante payments to actually induce wide-scale participation (Izquierdo-Tort et al. 2024). It therefore seems more reasonable to suggest a way to consistently apply matching or the construction of synthetic controls ex ante using data that is generally available and covariates that are known to be correlated with both selection into

treatment and deforestation outcomes. I think that is what is being suggested here. Counterfactual deforestation could then be predicted for reference areas, and then once time has passed actual deforestation rates would be used in assessment. I think what I would like to see is clarification of the policy proposition.

Your original response: We appreciate the reviewer's confusion here. Current practice is that the *ex ante* counterfactual scenario is produced before the project is implemented but credits are issued once an *ex post* assessment of project stock has been completed. The problem that we are raising is that the *ex post* assessment is limited to the project deforestation, while the counterfactual is not revised to accommodate the most up-to-date information. The reviewer is of course right that these projects have to raise investment in order to finance the project, prior to the sale of credits, but we have argued in Swinfield *et al.* (2024) that the credits should not be issued without also revising the counterfactual in the *ex post* assessment. We now make this clear at several points.

In the introduction:

"These ex ante counterfactual estimates were compared with measurements of deforestation in project areas to determine a certified metric of avoided deforestation with which to issue credits."

In the discussion:

"A potential solution is to move to only issuing credits ex post based on assessments which integrate new developments in quasi-experimental methods. "

AND

"For example, nature-based restoration projects typically assume that all increases in carbon or biodiversity above a pre-project level are additional⁴⁴. However, this is a strong assumption which should be validated by comparing project stocks with the average stocks in control areas selected using quasi-experimental methods⁴⁵."

My new comment: *I think that you might add a sentence after your first discussion sentence that says something like, "This does not change the timing of credit issuance, but rather mandates that the counterfactual be taken into account."*

My original comment: c. Two implementation issues should be recognized in the discussion. i. The first of these is that perfect prediction of the location of future deforestation is impossible. Deforestation is a rare event statistically, and so targeting programs such that only future deforested land is enrolled does not make sense, regardless of how much data you have with which to make the prediction. Even trying to approach this kind of targeting in a REDD+ program would require taking money away from actual incentives and funneling it into bureaucratic processes intended to support targeting, which would be wasteful. This means that you have to pay for enrollment of larger areas of land than are likely to be deforested.

Your original response: We agree with the reviewer that predicting the exact location of deforestation is very difficult. There are inevitable trade-offs between costs of targeting and additionality. For that reason we think that moves towards jurisdictional programmes will be

positive but the inclusion of quasi-experimental approaches would be a further improvement. We now say this in the discussion:

“Under JREDD+ methods such as VM0048 and ART TREES, counterfactual deforestation rates are set as the mean annual deforestation rate over a recent historic period (usually 5-6 years)¹². Our results suggest that this more straightforward and inflexible approach to estimating a counterfactual could significantly reduce over-crediting risks (Fig. 5). Further JREDD+ is designed to enable integration into Article 6 of the Paris agreement allowing jurisdictions to raise finance through the trade of International Transfers of Mitigation Outcomes³⁷. Under these methodologies, independent data providers, with no direct financial stake in the number of credits issued produce the assessment of deforestation over the historic period. This is a positive step which should reduce the perverse incentives which resulted in over-crediting in earlier project-level REDD+ methodologies.

Unfortunately, the movement to JREDD+ does not entirely remove the risk of perverse incentives leading to over-crediting. The methods for JREDD+ assume that the drivers of deforestation remain constant between the historic period and the project period. This assumption will often not hold as forest conservation or trade policies, major infrastructure projects³⁸ or extreme climatic events³⁹ will all vary over time and can impact deforestation. In fact, ex post measures of deforestation in untreated jurisdictions have been shown to differ by as much as 100% from ex ante predictions based on historic averages^{33,40}. If projects were to enrol in JREDD+ schemes randomly without regard for the likely change in future deforestation trends, this may not result in systematic bias. However, evidence tells us that proponents will opt into crediting schemes when it is to their advantage^{41,42}, with projects being preferentially developed in jurisdictions with declining deforestation pressures. A potential solution is to move to only issuing credits ex post based on assessments which integrate new developments in quasi-experimental methods. Ideally, different

quasi-experimental results should be considered, from methods such as the Permanent Additional Carbon Tonne (PACT) method used in our study⁴³ or advances in synthetic controls that require fewer assumptions²⁶. Although, quasi-experimental methods are still susceptible to unobservable, time-varying confounders, the scale of these effects can be tested and reported³¹. Therefore, until there is clear consensus on how best to measure impacts, a precautionary approach should be taken, with conservative estimates of avoided deforestation produced to safeguard the claims being made¹⁹.”

My new response: *I am happy with this discussion, but this suggests that a very policy-relevant analysis would try to evaluate the improvement in assessment by comparing the JREDD counterfactual to your more precise approach.*

My original comment: ii. This does not in itself undermine the principle of this paper. However, it is worth considering that the global community often says that avoided deforestation and, more recently, forest restoration, is “cheap.” The perceived cheapness of the intervention may be at least partly driven by the fact that when we calculate REDD+ transfers per hectare using current estimates of deforestation, they are very low. However, we might find that if we use current per hectare rates spread over a much lower, precisely estimated amount of avoided deforestation, we may not be able to induce any participation in these types of programs, because they will not sufficiently compensate producers for their opportunity cost. Environmental non-profit

organizations and rich country governments often engage in wishful thinking regarding the true costs of nature-based solutions. It is important that the research community does not fall prey to this same mistake.

Your original response: We agree with the reviewer that there is a widespread belief that REDD+ credits are the cheap option. We have now made clear in the discussion that our finding reveals that conserving nature is likely to cost more than expected:

“Indeed, despite inflated supply obscuring their real value, nature-based credits remain a critical tool in the fight against climate collapse and mass biodiversity loss.”

My new comment: *I think you could say this more clearly. There is a real risk that if someone decided to issue credits based upon your methodology, there would not be sufficient funds to actually get people to reduce deforestation. In many PES programs that I know of, the payment levels are set to accommodate the fact that policymakers know that they are not paying only for the additional hectares. Rather, they are spreading the cost of compensating for the additionality across a much larger forest area. Therefore, the cost per hectare of forest looks low, while the cost per hectare of avoided deforestation is high. If credits are issued based upon more precise avoided deforestation measures, this will lower payments to project participants, and will risk undermining the entire process unless the price of credits is increased.*

NCOMMS-24-45991-T

“Understanding the reasons for over-crediting in REDD+”

August 2024

Summary

Thank you for the opportunity to read the revised paper. I very much appreciate that most of my comments have been addressed. Previously, my main concerns focused on the manuscript’s contribution and the use of empirical language/concepts. The authors have addressed both aspects adequately.

I only have a few minor editing comments and a small suggestion about the title.

Minor comments

1. The new title is slightly misleading: “Understanding the *reasons*” implies that the authors will analyze the political and economic reasons of over-crediting. E.g., certification institutions have an economic incentive to certify higher avoided deforestation figures in order to please the client. This manuscript analyses the *mechanisms* of over-crediting. I suggest reformulating the title to “Understanding the mechanisms for over-crediting in REDD+”.

2. **Original comment:** Figure 1b shows there ratio $\frac{\text{Avoided deforestation credited by VERRA}}{\text{Avoided deforestation using quasi-experimental methods}}$. For overestimation, the ratio is < 1 , and for underestimation, the ratio is > 1 . I suggest flipping the definition of the ratio so that values > 1 reflect overestimation.

Authors response: We agree with the reviewer that this is confusing however because quasi-experimental methods tend to estimate values close to, at or even below zero, whereas certifying methods are all positive, the proposed approach would make it impossible to present all the projects as they would have very large or infinite value. Logging the data is also problematic due to the negative values. Therefore we have opted to retain the same presentation. However, we now presented over-crediting as suggested in the results:

“The mean additionality ratio (calculated as the mean quasi-experimental estimate of avoided deforestation / certified estimate, averaged across the 44 projects claiming to have avoided deforestation) of 0.27 (Fig. 2b) was significantly $\neq 0$ (one-tailed Mann-Whitney U against a group of 0s, $U = 1628$, $p \leq 0.001$). This suggests that on average projects were additional but had over-credited by a factor of 3.8.”

New comment: You could use an inverse asymptotic sine transformation. It has similar properties to the log-transformation, and it is defined for negative values. Imagine I use this figure in a lecture: It is complicated to explain to students that a point with a value of 0.16 on the y-axis means overcrediting, and they should calculate $1/0.16=6.25$,

to understand that VERA calculated a 625% higher level of avoided deforestation than the quasi-experimental method.

3. Line 51. "Unfortunately, over-crediting by REDD+ Projects is still not widely accepted." It is unclear who is not accepting and why everybody should accept. I recommend using a milder, scientifically impartial language.
4. The introduction should have a brief paragraph summarizing the results.
5. The introduction should have a paragraph highlighting the scientific contribution: "We contribute to the literature on ..."
6. Abbreviations are used but not explained in the main text (as far as I see). E.g. ACC, QEM.
7. L430. The concept of "dynamic baselines" is recommended, but not explained at any point.

NCOMMS-24-45991-T

“Understanding the mechanisms for over-crediting in REDD+”

September 2025 – Round 3

Thank you again for the opportunity to read this paper in its 2nd revised version. I think it is an excellent paper that has benefited immensely from the comments of all reviewers. I expect that the paper will have a significant impact on the readers of Nature Communications and be widely cited.

First, I am also glad that my comments led the authors to revise their database, based on their complementary studies, and identified errors in the data.

I remain sceptical about the contribution made here beyond what has already been shown in other publications by the authors (First comment of the first review). Nonetheless, I fully support this publication, but must insist that the authors should clearly state their scientific contribution.

In this context, there is a misunderstanding of my previous comment that followed up on comment 1 of the first review:

“5. The introduction should have a paragraph highlighting the scientific contribution: ”We contribute to the literature on ...”

In response, the authors added this paragraph:

Our study contributes to the literature examining how to infer causal effects from observational data^{27,28}, especially in conservation where randomised control trials are rare or impossible^{19,20}. Our results show clearly that despite variation in estimates from quasi-experimental methods, there is consistent evidence of over-crediting. This result cannot be explained by differences between the remote sensing layers used by quasi-experimental and certified assessments. Instead, we find that over-crediting was primarily driven by the selection of control areas with greater exposure to deforestation than projects and through unrealistic modelling of future deforestation rates.

What I expected was a paragraph that contextualizes the paper within a) the (forest) conservation research in general and b) the REDD+ crediting research in particular – explaining what this paper contributes (methodological, theoretical, or conceptually, ...) to our knowledge beyond what was shown in West et al. (2020), West et al. (2024), Guizar-Coutiño et al. (2022), Guizar-Coutiño et al. (2024), and Balmford et al. (2023).

References

- Balmford, Andrew, David Coomes, James Hartup et al. 2023. “PACT Tropical Moist Forest Accreditation Methodology.” [10.33774/coe-2023-g584d-v2](https://doi.org/10.33774/coe-2023-g584d-v2).
- Guizar-Coutiño, Alejandro, David Coomes, Tom Swinfield, and Julia P G Jones. 2024. “Sensitivity of estimates of the effectiveness of REDD+ projects to matching specifications and moving from pixels to polygons as the unit of analysis.” *bioRxiv*. [10.1101/2024.05.22.595326](https://doi.org/10.1101/2024.05.22.595326).
- Guizar-Coutiño, Alejandro, Julia P. G. Jones, Andrew Balmford, Rachel Carmenta, and David A. Coomes. 2022. “A global evaluation of the effectiveness of voluntary REDD+ projects at reducing deforestation and degradation in the moist tropics.” *Conservation Biology* 36 (6): . [10.1111/cobi.13970](https://doi.org/10.1111/cobi.13970).
- West, Thales A. P., Jan Börner, Erin O. Sills, and Andreas Kontoleon. 2020. “Overstated carbon emission reductions from voluntary REDD+ projects in the Brazilian Amazon.” *Proceedings of the National Academy of Sciences* 117 (39): 24188–24194. [10.1073/pnas.2004334117](https://doi.org/10.1073/pnas.2004334117).
- West, Thales A.P., Barbara Bomfim, and Barbara K. Haya. 2024. “Methodological issues with deforestation baselines compromise the integrity of carbon offsets from REDD+.” *Global Environmental Change* 87 102863. [10.1016/j.gloenvcha.2024.102863](https://doi.org/10.1016/j.gloenvcha.2024.102863).